# 3D-DLP: Self-supervised 3D Object-centric Scene Representation Learning

**Ellina Zhang** [1]   **Madhavan Iyengar** [1]   **Amir Zadeh** [2]   **Chuan Li** [2]   **David Held** [1]   **Deepak Pathak** [1]   **Tal Daniel** [1]

## Abstract

We introduce 3D-DLP, a self-supervised object-centric representation learning model that decomposes scene-level RGB-D or voxel observations into a set of 3D latent particles. Building on the Deep Latent Particles (DLP) framework, each particle encodes disentangled attributes, including 3D keypoint position, bounding box dimensions, and appearance features, and represents a distinct entity in the scene. The model learns interpretable per-particle segmentation maps through an end-to-end self-supervised reconstruction objective. We demonstrate on both simulated and real-world datasets that the learned latent space is interpretable and controllable: by manipulating particle positions and decoding, we can generate novel scene configurations. Furthermore, we show that leveraging these compact 3D latent particles for downstream robotic manipulation improves performance over baselines that either lack explicit 3D information or rely on memory-intensive dense 3D inputs without object-centric structure. Code and videos are available at https://eubooks3003.github.io/3d-dlp/.

## 1. Introduction

3D representations are increasingly vital for robotic decision-making (Shridhar et al., 2022; Goyal et al., 2023), especially for manipulation where understanding scene geometry is crucial. Unlike 2D projections, explicit 3D representations preserve spatial relationships for contact reasoning and faithfully capture true geometry; while voxelization alone does not recover unobserved regions, a structured 3D grid provides a more uniform substrate for learning under partial observability, particularly when fused across multiple views.

However, 3D sensor data–RGB-D images, point clouds, voxels–is noisy, sparse, and high-dimensional. Voxel methods scale cubically with resolution; point clouds struggle with variable density and occlusions; dense 3D features often exceed memory budgets for real-time control.

Object-centric 3D representations offer a promising alternative by decomposing scenes into semantic entities. Supervised methods like GROOT (Zhu et al., 2023) show that explicit 3D object representations improve generalization over holistic scene models but require costly annotations that limit real-world scalability.

Self-supervised 2D object-centric methods have excelled at complex multi-object manipulation (Haramati et al., 2024; Zadaianchuk et al., 2022; Qi et al., 2025; Haramati et al., 2026), proving factorized representations are indeed valuable for downstream control. Yet they cannot recover occluded regions or model precise 3D geometry essential for contact-rich tasks. Prior 3D object-centric approaches—including patch-based point-cloud methods (Wang et al., 2022) and neural-rendering variants (Luo et al., 2025; Smith et al., 2023; Stelzner et al., 2021; Zhao et al., 2024; Zhang et al., 2025)—typically operate on colorless or synthetic data or rely on rendering-side inverse problems, or they do not yield practical low-dimensional representations suitable for downstream policy learning.

We introduce 3D Deep Latent Particles (3D-DLP), extending efficient particle-based object-centric representations (Daniel & Tamar, 2022; Daniel et al., 2026) to directly process real-world RGB-D and voxel inputs. Our approach handles *colored* 3D observations directly, with a compact particle representation that scales to real data. We demonstrate self-supervised 3D object discovery, explicit latent editing (e.g., moving objects), and improved robotic manipulation–providing the *first* practical bridge from self-supervised 3D scene decomposition to downstream control.

Our contributions are as follows: (1) we introduce, to the best of our knowledge, the first self-supervised object-centric scene representation that operates on colored 3D voxels, and present a unified framework covering RGB-D, occupancy-voxel, and RGB-voxel inputs; (2) we identify two methodological components required to make 3D-DLP work on dense voxel scenes—an appearance-aware K-means keypoint prior and a chroma reconstruction loss—and

[1]Carnegie Mellon University [2]Lambda AI. Correspondence to: Ellina Zhang <erzhang@andrew.cmu.edu>.

*Proceedings of the 43rd International Conference on Machine Learning*, Seoul, South Korea. PMLR 306, 2026. Copyright 2026 by the author(s).

validate both via ablations; (3) we show that the learned latents are controllable and interpretable by editing particle position and scale; and (4) we adapt an entity-centric diffusion-based policy (Qi et al., 2025) to show that 3D-DLP particles yield consistent gains over matched 2D-particle and voxel-only baselines on 12 `MimicGen` and 10 language-conditioned `RLBench` tasks.

## 2. Related Work

We position our approach at the intersection of two research areas: self-supervised object-centric representations, which enable efficient scene decomposition but have been limited to 2D inputs, and policy learning from 3D observations, which leverages geometric information but typically operates on dense, unstructured representations.

**Self-supervised Object-centric Representations.** Several recent approaches have been proposed to learn a self-supervised object-centric decomposition of visual inputs, such as images or videos. Patch-based approaches (Lin et al., 2020; Crawford & Pineau, 2019; Stanić & Schmidhuber, 2019) propose object latents from each patch, whereas slot-based approaches (Locatello et al., 2020; Burgess et al., 2019; Engelcke et al., 2020; Greff et al., 2019) iteratively assign pixels to a pre-defined number of slots via spatial attention, and particle-based approaches (Daniel & Tamar, 2022; 2024) represent object latents based on learned keypoints. Crucially, these methods are restricted to 2D RGB inputs and do not account for 3D observations, such as RGB-D (RGB with depth) or voxels. An early extension to 3D is SPAIR3D (Wang et al., 2022), which adapts the patch-based SPAIR (Crawford & Pineau, 2019) to point clouds; however, it relies on a memory-intensive iterative pipeline that limits scalability to synthetic datasets and operates in *colorless* settings, and—being patch-based rather than particle-based—is methodologically distinct from our approach. To the best of our knowledge, our work provides the *first demonstration* of self-supervised 3D object-centric colored scene decomposition operating directly on 3D observations (RGB-D and voxels), bridging this gap by extending the particle-based Deep Latent Particles (DLP, (Daniel & Tamar, 2024)) framework to directly process RGB-D and voxel inputs with a compact, flexible representation that improves robotic manipulation performance.

More recent work has broadened the scope of 3D object-centric learning, several using neural-based rendering (Luo et al., 2025; Smith et al., 2023; Stelzner et al., 2021; Zhao et al., 2024; Zhang et al., 2025). Specifically, uOCF (Luo et al., 2025) learns object-centric neural fields from image observations via inverse rendering; DynaVol-S (Zhao et al., 2024) studies dynamic scene decomposition through object-centric voxelization and neural rendering; and GrabS (Zhang et al., 2025) tackles unsupervised 3D object segmentation in

point clouds using object-centric priors and embodied querying. These works are complementary to ours but differ in both representation and learning setup: in contrast to neural-field or rendering-based approaches, we extend the DLP framework itself to explicit 3D observations with a direct reconstruction objective in 3D, preserving DLP's particle structure while avoiding dependence on inverse rendering and continuous scene querying.

**Policy Learning from 3D Observations.** Several recent policy learning methods utilize 3D observations, such as RGB-D, point clouds or voxels, to improve spatial reasoning and contact-sensitive manipulation. PerAct (Shridhar et al., 2022; Grotz et al., 2024) fuses voxels with language conditioning, while RVT (Goyal et al., 2023; 2024) renders multiple views from point clouds to learn 3D action heatmaps in a pose-agnostic manner. In contrast, our method first learns a 3D object-centric latent representation from either RGB-D or voxels, which is then integrated into a diffusion-based policy (Qi et al., 2025) akin to recent policies that employ diffusion over 3D inputs (Ze et al., 2024; Liu et al., 2025). While 3D object-centric representations have recently shown great promise for policy learning, as demonstrated by GROOT (Zhu et al., 2023), such methods typically rely on pre-trained supervised segmentation and feature extractor models. Our proposed method, by contrast, is fully self-supervised and does not require any annotations or auxiliary models.

## 3. Background

Our method learns self-supervised 3D object-centric representations by extending the Deep Latent Particles (DLP) framework to RGB-D and voxel inputs. In the following, we briefly review the DLP model, which forms the basis of our approach.

**Deep Latent Particles (DLP).** DLP (Daniel & Tamar, 2022; 2024) is a particle-based self-supervised object-centric model trained as a variational autoencoder (VAE (Kingma & Welling, 2014)), where the latent space is structured as a set of particles that represent entities in the input scene. Given an RGB image $I \in \mathbb{R}^{3 \times H \times W}$, DLP encodes it into a set of $M$ foreground particles $\{z_{\text{fg}}^m\}_{m=1}^{M}$ and a single background particle $z_{\text{bg}}$. Each foreground particle factorizes into disentangled stochastic attributes

$$z_{\text{fg}} = [z_p, z_s, z_c, z_t, z_f] \in \mathbb{R}^{6+d_{\text{obj}}},$$

where the position $z_p \sim \mathcal{N}(\mu_p, \sigma_p^2) \in \mathbb{R}^2$ encodes 2D keypoint coordinates, the scale $z_s \sim \mathcal{N}(\mu_s, \sigma_s^2) \in \mathbb{R}^2$ models bounding-box dimensions, the composition order $z_c \sim \mathcal{N}(\mu_c, \sigma_c^2) \in \mathbb{R}$ specifies the stitching order and thus local occlusion relations, the transparency $z_t \sim \text{Beta}(a, b) \in [0, 1]$ controls per-particle presence, and the visual features $z_f \sim \mathcal{N}(\mu_f, \sigma_f^2) \in \mathbb{R}^{d_{\text{obj}}}$ capture the appearance of the local

region around the particle. The background is represented by a single latent

$$z_{\text{bg}} \sim \mathcal{N}(\mu_{\text{bg}}, \sigma_{\text{bg}}^2) \in \mathbb{R}^{d_{\text{bg}}},$$

which is fixed at the image center and encodes global background features. DLP first extracts keypoint proposals from image patches using a spatial-softmax operation (SSM (Jakab et al., 2018; Finn et al., 2016)) applied to learned feature maps; these keypoints are then turned into particles by predicting the aforementioned attributes from local features around each keypoint. In the decoder, each particle is mapped to a spatial appearance map, and all maps are composited on a canvas according to the particle position, scale, composition order, and transparency to reconstruct the input image. DLP uses spatial transformers (STN (Jaderberg et al., 2015)) for differentiable glimpse extraction and placement. We note that our implementation builds on the latest DLP revision (DLPv3 (Daniel et al., 2026)), which we refer to simply as DLP throughout the paper.

## 4. 3D Deep Latent Particles (3D-DLP)

We aim to learn self-supervised, object-centric representations of 3D scenes that are both compact and structured, supporting two key capabilities: (1) *scene decomposition*–disentangling individual objects from background clutter, and (2) *decision-making*–providing low-dimensional state representations suitable for downstream control policies. Given a 3D observation $\mathbf{x}$ (RGB-D image, occupancy voxels, or RGB voxels), our family of models, **3D-DLP**, infers $M$ foreground latent *particles* $\{z_{\text{fg}}^m\}_{m=1}^M$ –each encoding explicit 3D spatial location plus geometric and visual attributes, together with a single background latent $z_{\text{bg}}$.

We introduce three variants adapted to different 3D sensing modalities: **3D-DLP-D** processes RGB-D images $\mathbf{x} \in \mathbb{R}^{4 \times H \times W}$ (Appendix A.1); **3D-DLP-V** handles occupancy voxel grids $\mathbf{x} \in \{0,1\}^{1 \times D \times H \times W}$ (Appendix A.2); and **3D-DLP-VC** tackles the most challenging case of colored RGB voxel grids $\mathbf{x} \in [0,1]^{3 \times D \times H \times W}$.

All variants share a common three-stage architecture: a **Prior** proposes keypoint locations, the **Encoder** infers particle attributes and appearance latents, and the **Decoder** renders and composites particles to reconstruct the input. While 3D-DLP-D naturally extends 2D DLP (Daniel et al., 2026) by adding depth-channel support, 3D-DLP-V and 3D-DLP-VC introduce novel frameworks for learning object-centric structure directly from 3D volumetric data. Due to space constraints, we focus our main-text description on **3D-DLP-VC**–the colored voxel model representing our most general contribution–with full details for other variants deferred to the appendices. Figure 1 illustrates the architecture of 3D-DLP-VC.

**From point clouds to voxels.** 3D observations are often represented as RGB point clouds $\mathcal{P}^{\text{rgb}} = \{(\mathbf{q}_i, \mathbf{c}_i)\}_{i=1}^N$, where $\mathbf{q}_i \in \mathbb{R}^3$ are 3D coordinates $(z, y, x)$, $\mathbf{c}_i \in [0,1]^3$ are RGB colors, and $N$ varies per scene. While point clouds preserve fine geometric detail, their variable cardinality (different number of points per scene) and lack of a canonical grid make it difficult to (1) batch examples efficiently, (2) exploit translation-equivariant convolutional architectures (typically require regularly spaced grids rather than scattered points), and (3) instantiate the DLP pipeline of proposing keypoints, extracting local crops, and inferring particle attributes, which relies on spatially indexed feature maps and differentiable cropping.

**Voxelization.** To address these challenges, we *voxelize* the point cloud into a regular grid $\mathbf{x} \in [0,1]^{3 \times D \times H \times W}$, where each voxel stores an aggregated color of the points falling into it (details in Appendix A.2). This converts an irregular point set into a dense 3D tensor–a direct analogue of a 2D image where each cell has fixed spatial meaning. Voxels thus provide a natural substrate for extending 2D DLP to 3D: we replace 2D with 3D convolutions to obtain feature volumes, extract $M$ particle-centered crops via a 3D spatial transformer, and decode canonical cubic particle patches that are placed back into the global grid.

Next, we describe how the DLP components are adapted to account for these 3D considerations.

**Prior.** The prior proposes initial keypoint locations for latent particles. Unlike 2D DLP's spatial softmax (SSM) based keypoint prior, which fails on sparse, discontinuous voxel grids (as we verify in our ablation analysis, Sec. 5.1), we introduce an *appearance-aware* K-means (Hartigan & Wong, 1979) prior. For each occupied voxel $\mathbf{u}$, we form joint appearance-geometry features:

$$\mathbf{f}(\mathbf{u}) = \left[ \phi(\mathbf{c}(\mathbf{u})); \mathbf{p}(\mathbf{u}) \right] \in \mathbb{R}^6,$$

where $\mathbf{c}(\mathbf{u}) \in [0,1]^3$ is voxel color converted to perceptually-uniform CIELAB space $\phi(\mathbf{c}(\mathbf{u})) = [L^*, a^*, b^*]$–ensuring Euclidean distances reflect visual similarity better than RGB (Iizuka et al., 2016)–and $\mathbf{p}(\mathbf{u}) \in [-1,1]^3$ is normalized 3D position.

We perform clustering in this *joint appearance-geometry* space via weighted K-means using lightness weights $w(\mathbf{u}) = L^*(\mathbf{u})$, biasing toward visually informative surface regions. Each cluster $\mathcal{C}_k$ yields geometric-only cluster centers:

$$\bar{\mathbf{z}}_p^k = \frac{\sum_{\mathbf{u} \in \mathcal{C}_k} w(\mathbf{u}) \, \mathbf{p}(\mathbf{u})}{\sum_{\mathbf{u} \in \mathcal{C}_k} w(\mathbf{u})}.$$

This *key methodological contribution* produces particle centers that naturally align with object surfaces and color boundaries–far more effectively than geometry-only clustering (e.g., using only voxel occupancy features)–enabling

robust object discovery in colored 3D voxel scenes. Further details on lightness weighting and K-means initialization are in Appendix A.3.1.

**Encoder.** The encoder models the approximate posterior $q(z|\mathbf{x})$, inferring particle attributes and appearance latents from input voxels $\mathbf{x} \in [0,1]^{3 \times D \times H \times W}$. It closely follows 2D DLP (Daniel et al., 2026) with these key 3D adaptations: (1) 2D CNNs → 3D CNNs, (2) bilinear → trilinear STN sampling (Jaderberg et al., 2015), (3) 2D position and scale attributes → 3D vectors $(z, y, x)$, (4) no composition-order $z_c$ (occlusions handled by 3D rendering), and (5) SSM variance → intra-cluster covariance for keypoint selection (Appendix A.2.1).

*Particle attributes.* From the K-means proposals $\{\bar{\mathbf{z}}_p\}$, a 3D CNN attribute encoder with STN-extracted glimpses predicts per-particle offsets $\Delta \mathbf{z}_p^m$, scales $\mathbf{z}_s^m$, and transparencies $z_t^m$. Final positions are $\mathbf{z}_p^m = \bar{\mathbf{z}}_p^m + \Delta \mathbf{z}_p^m$. Following DLP (Daniel & Tamar, 2024), we select the top-$M$ most confident particles combining proposal and offset uncertainties.

*Appearance encoding.* For each selected $\mathbf{z}_p^m$, an STN extracts a local RGB volume glimpse (in original RGB space, unlike the prior's CIELAB). A CNN produces appearance latents $\mathbf{z}_f^m$. For particles with $z_t^m > 0$, we mask their regions from $\mathbf{x}$ and encode the residual with another CNN as background features $\mathbf{z}_{\text{bg}}$ (extended encoder details in Appendix A.3.2).

**Decoder.** The decoder defines $p(\mathbf{x}|z)$ and composites particles into a full-resolution reconstruction $\hat{\mathbf{x}} \in [0,1]^{3 \times D \times H \times W}$.

*Particle decoding.* Each foreground particle $m$ first decodes its appearance latent $\mathbf{z}_f^m$ via 3D CNN to a canonical cubic RGBA patch of size $P^3$:

$$\mathbf{z}_f^m \mapsto (\tilde{\alpha}_m, \tilde{\mathbf{c}}_m) \in [0,1]^{1 \times P^3} \times [0,1]^{3 \times P^3},$$

where $\tilde{\alpha}_m \in [0,1]^{P^3}$ is the alpha (opacity) channel and $\tilde{\mathbf{c}}_m \in [0,1]^{3 \times P^3}$ gives RGB color ($3 \times P^3$). Spatial attributes $(\mathbf{z}_p^m, \mathbf{z}_s^m)$ then place this patch onto the $D \times H \times W$ global grid using 3D spatial transformer (STN) with trilinear sampling, yielding per-voxel alpha $\alpha_m(\mathbf{u})$ and RGB color $\mathbf{c}_m(\mathbf{u})$. Particles are gated by their transparency scalar $z_t^m$: $\bar{\alpha}_m(\mathbf{u}) = z_t^m \cdot \alpha_m(\mathbf{u})$. The background latent $\mathbf{z}_{\text{bg}}$ decodes directly to full-resolution background RGB $\mathbf{c}^{\text{bg}}(\mathbf{u}) \in [0,1]^{3 \times D \times H \times W}$.

*Volumetric compositing.* To render realistic composite scenes from individual object renderings and a background, we perform volumetric compositing using foreground object density fields. The foreground weights for each object $m$ at pixel $\mathbf{u}$ are the normalized densities

$$w_m(\mathbf{u}) = \frac{\bar{\alpha}_m(\mathbf{u})}{\sum_j \bar{\alpha}_j(\mathbf{u}) + \varepsilon}, \quad \varepsilon = 10^{-9},$$

representing each object's contribution to the final pixel color $\mathbf{c}^{\text{obj}}(\mathbf{u}) = \sum_m w_m(\mathbf{u})\mathbf{c}_m(\mathbf{u})$. The background mask is $m^{\text{bg}}(\mathbf{u}) = 1 - \min(1, \sum_m \bar{\alpha}_m(\mathbf{u}))$, yielding the final rendered pixel:

$$\hat{\mathbf{x}}(\mathbf{u}) = m^{\text{bg}}(\mathbf{u})\mathbf{c}^{\text{bg}}(\mathbf{u}) + (1 - m^{\text{bg}}(\mathbf{u}))\mathbf{c}^{\text{obj}}(\mathbf{u}).$$

We provide further details on the stitching process in Appendix A.3.3.

**Loss.** Similarly to DLP, 3D-DLP is trained as a variational autoencoder (VAE) by maximizing an evidence lower bound (ELBO). The objective decomposes into:

$$\mathcal{L} = \mathcal{L}_{\text{rec}} + \beta_{\text{KL}} \mathcal{L}_{\text{KL}} + \beta_{\text{obj}} \mathcal{L}_{\text{obj}},$$

where $\mathcal{L}_{\text{rec}}$ is the reconstruction loss (detailed below), $\mathcal{L}_{\text{KL}}$ is the KL-divergence loss for particle latents w.r.t. fixed priors (identical to DLP) and detailed in Appendix A.2.4, and $\mathcal{L}_{\text{obj}} = \left(\sum_{m=1}^{M} \mathbf{z}_t^m\right)^2$ encourages sparse particle usage.

*Reconstruction loss.* We combine MSE with a chroma loss (Habermann et al., 2021) applied only on occupied (non-empty) voxels:

$$\mathcal{L}_{\text{rec}} = \|\hat{\mathbf{x}} - \mathbf{x}\|_2^2 + \lambda_{\text{chroma}} \sum_{\mathbf{u}} m(\mathbf{u})\|\hat{\mathbf{C}}(\mathbf{u}) - \mathbf{C}(\mathbf{u})\|_2^2,$$

with $\lambda_{\text{chroma}} = 500$. The occupancy mask $m(\mathbf{u}) \in \{0,1\}$ is the ground-truth occupancy channel - the chroma term is therefore evaluated only where color is defined.

*Chroma loss.* Chroma loss (Habermann et al., 2021) separates color into luminance $Y(\mathbf{u})$ (channel mean) and chrominance $\mathbf{C}(\mathbf{u}) = \mathbf{x}(\mathbf{u}) - Y(\mathbf{u})[1,1,1]^\top$ (color residual), penalizing only chrominance error on occupied voxels. This *prevents gray collapse*: MSE alone can match brightness with gray colors (zero chrominance) as illustrated in Figure 5, but chroma loss forces true hue/saturation fidelity, as confirmed by our ablations (Sec. 5.1). We provide detailed loss definitions in Appendix A.3.4.

**Implementation details.** We implement 3D-DLP in Py-Torch (Paszke et al., 2017), optimizing with Adam (Kingma & Ba, 2015) with a fixed learning rate of $8 \times 10^{-5}$. Models use $64^3$ voxel resolution (depth×height×width) and a dataset-dependent number of particles. K-means runs for 5 iterations with 64 proposals. Training uses batch size 32 on a single Nvidia GH200 GPU (~48 hours). Full hyperparameters are in Appendix G. Code is available at https://github.com/Eubooks3003/3d-dlp.

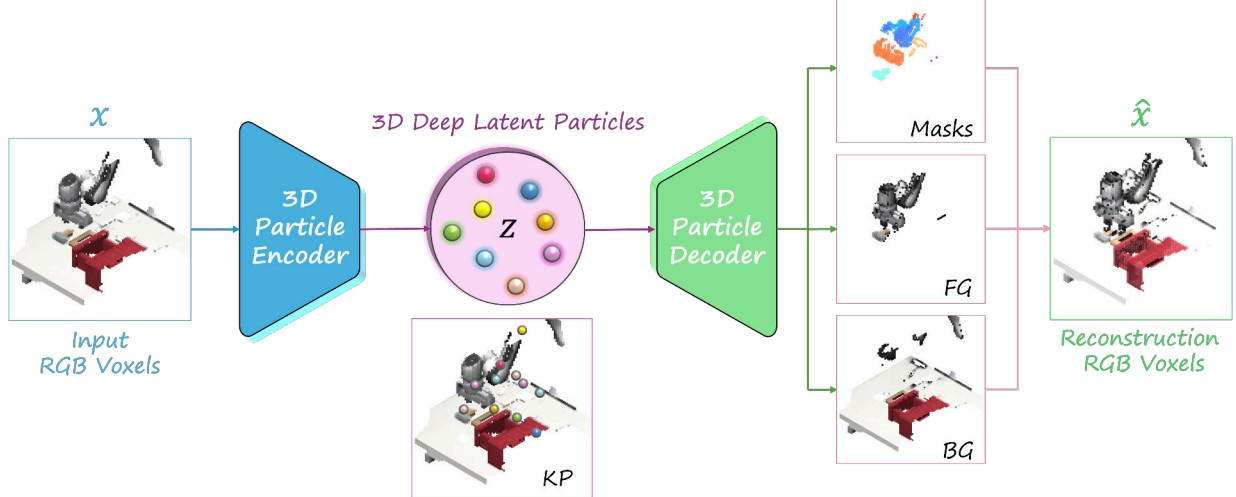

*Figure 1.* **3D-DLP-VC architecture for RGB voxels.** An input RGB voxel grid $x$ is encoded into $M$ latent particles $z$, each containing 3D keypoint positions, bounding-boxes (scale) and appearance features. The decoder renders per-particle foreground objects (FG), segmentation masks, and background (BG) volumes, then composites them into the final reconstruction $\hat{x}$.

## 5. Experiments

We design our experimental suite to address four key questions: (1) How does the visual reconstruction quality of our object-centric approach compare to non-object-centric baselines? (2) Are the learned latent representations interpretable and controllable—can we modify latent particles to generate meaningful scene variations? (3) Which core components contribute to the model's performance? and (4) Do 3D object-centric representations improve performance in downstream robotic manipulation tasks? We accordingly organize our experiments into two subsections: §5.1 addresses questions (1)–(3) via self-supervised scene decomposition, reconstruction, and ablations, and §5.2 addresses question (4) via imitation learning on `MimicGen` and `RLBench`.

### 5.1. Self-supervised 3D Object Discovery and Scene Reconstruction

We evaluate our self-supervised 3D-DLP models on synthetic and real-world datasets, assessing both object discovery and reconstruction quality. We compare reconstruction fidelity against non-object-centric baselines, visualize discovered keypoints and segmentation masks, demonstrate latent space controllability through particle modifications, and conduct ablation studies on our core components.

**Datasets.** We evaluate our approach on four datasets: two synthetic Blender (Community, 2018) corpora- `GenericShapes`, containing simple geometric meshes such as spheres and cylinders, and `ShapeNetScenes`, comprising scenes with 2–5 randomly sampled ShapeNet (Chang et al., 2015) objects; one robotic simulation dataset—`MimicGen` (Mandlekar et al.,

2023); and one real-world benchmark—the `UW RGB-D Scenes Dataset v2 (RGB-D-SD-v2)` (Newcombe et al., 2015). The synthetic datasets (`GenericShapes`, `ShapeNetScenes`) each contain approximately 40,000 scenes; `MimicGen` is aggregated from 50 demonstration trajectories per task (∼180k frames total); and `RGB-D-SD-v2` provides 14 real reconstructions augmented via tabletop rearrangement. All datasets are split into train/val/test with an $[0.8, 0.1, 0.1]$ ratio. We provide additional details on dataset construction, voxelization, and data augmentations in Appendix B.

**Baselines.** To the best of our knowledge, 3D-DLP is the first method to perform self-supervised, object-centric scene decomposition directly from RGB voxels. Prior 3D object-centric work operates in different settings—e.g., colorless point clouds (Wang et al., 2022) or neural-field rendering (Luo et al., 2025). Scene-level voxel reconstruction is also rarely studied (Lee et al., 2023). We thus compare against two non-object-centric autoencoding baselines: a deterministic autoencoder (AE) and a variational autoencoder (VAE (Kingma & Welling, 2014)). Both use the same reconstruction loss and comparable encoder-decoder architectures as 3D-DLP. For VAE, we tune $\beta_{\mathrm{KL}}$ for optimal performance. We implement identical baselines across all input modalities: RGB-D, occupancy voxels, and RGB voxels. For RGB-D inputs, we additionally compare against slot-based object-centric methods SAVi (Kipf et al., 2022) and SLATE (Singh et al., 2022), adapted to RGB-D; 3D-DLP-D substantially outperforms both quantitatively (PSNR/SSIM/LPIPS) and qualitatively (slot under-utilization), with metrics and per-task decomposition visualizations in Appendix D.1.

**Metrics.** For RGB-D reconstruction, we report standard

image metrics: PSNR, SSIM (Wang et al., 2004), and LPIPS (Zhang et al., 2018). For voxel grids, we report intersection-over-union (IoU (Mescheder et al., 2019)), which measures per-voxel occupancy overlap, which is particularly important for multi-object scenes where accurate object boundaries enable proper separation from background and other objects. For RGB voxels, we additionally report masked PSNR, computed only over ground-truth occupied voxels. This excludes the large proportion of empty voxels, ensuring the metric reflects true surface reconstruction quality rather than background memorization.

**Object discovery.** Figures 1 and 2 demonstrate that 3D-DLP-VC discovers semantic keypoints and bounding boxes, and generates foreground and background masks without any supervision, enabling high-fidelity scene reconstruction. These results provide compelling evidence of a truly disentangled, object-centric latent representation that decomposes complex scenes into semantically meaningful entities. Additional visualizations for all modality variants are provided in Appendix D.

**Scene reconstruction.** Table 1 shows that 3D-DLP-VC substantially outperforms non-object-centric autoencoding baselines in Masked PSNR, while remaining competitive with the deterministic AE on IoU. The same trend holds for 3D-DLP-V on occupancy voxels (Appendix Table 11), where our object-centric approach consistently leads or matches baselines. We attribute the modest IoU gap to our VAE-based approach, which samples noisy latents via the reparameterization trick (Kingma & Welling, 2014) unlike the deterministic AE. This stochasticity trades minor reconstruction crispness for a disentangled, semantically structured latent space of explicit object entities. Figure 3 provides qualitative comparisons across datasets. On the less diverse RGB-D-SD-v2 dataset, non-object-centric methods produce competitive reconstructions. However, as data diversity increases (more object types, locations, colors) in the generated synthetic datasets, our particle-based inductive bias yields noticeably sharper, more faithful reconstructions. Appendix D presents results for all modality variants, confirming that 3D object-centric representations consistently enable superior reconstruction quality.

**Latent controllability.** We demonstrate the controllability and interpretability of our 3D latent particles by directly modifying their disentangled attributes and observing the effects on reconstructed scenes.

In Figure 4, perturbing particle 3D keypoints moves corresponding objects, while scaling attributes resize them, confirming the latent particles encode semantic, editable 3D object properties. These results validate 3D-DLP representations for downstream multi-object reasoning tasks. Additional examples appear in Appendix D.

| Dataset | Masked PSNR ↑ | IoU ↑ |
|---|---|---|
| GenericShapes | | |
| AE | $9.64 \pm 0.50$ | $0.279 \pm 0.05$ |
| VAE | $9.47 \pm 0.70$ | $0.011 \pm 0.01$ |
| 3D-DLP-VC (Ours) | $\mathbf{10.68 \pm 0.86}$ | $\mathbf{0.276 \pm 0.001}$ |
| RGB-D-SD-v2 | | |
| AE | $19.07 \pm 0.50$ | $\mathbf{0.771 \pm 0.06}$ |
| VAE | $15.63 \pm 1.04$ | $0.361 \pm 0.04$ |
| 3D-DLP-VC (Ours) | $\mathbf{21.57 \pm 0.78}$ | $0.731 \pm 0.04$ |
| MimicGen | | |
| AE | $11.39 \pm 0.79$ | $0.582 \pm 0.03$ |
| VAE | $4.35 \pm 2.09$ | $0.244 \pm 0.15$ |
| 3D-DLP-VC (Ours) | $\mathbf{24.41 \pm 0.26}$ | $\mathbf{0.910 \pm 0.01}$ |

*Table 1.* **RGB voxel reconstruction.** We report Masked PSNR (higher is better) and IoU (higher is better). 3D-DLP-VC substantially outperforms non-object-centric baselines (AE, VAE)

| Particles ($M$) | Masked PSNR ↑ | IoU ↑ |
|---|---|---|
| 8 | $21.30 \pm 0.29$ | $0.72 \pm 0.00$ |
| 16 | $22.48 \pm 0.31$ | $0.810 \pm 0.01$ |
| **24 (default)** | $\mathbf{24.41 \pm 0.26}$ | $\mathbf{0.910 \pm 0.00}$ |
| 40 | $23.93 \pm 0.35$ | $0.890 \pm 0.00$ |

*Table 2.* Ablation on number of particles $M$ for 3D-DLP-VC on MimicGen. Performance peaks at $M=24$; adding more particles does not improve reconstruction.

**Ablation study.** We conduct modality-specific ablations across RGB-D, occupancy voxels, and RGB voxels. In the main text, we focus on 3D-DLP-VC (RGB voxels) results in Table 3, deferring full results, such as loss type (MSE vs. BCE) for occupancy voxels, to Appendix C. All ablations use the MimicGen (stack task) dataset with 50 epochs of training. For 3D-DLP-VC, we ablate our K-means keypoint proposals by replacing them with spatial softmax (SSM) proposals (Daniel & Tamar, 2024): "SSM Raw" applies SSM directly to sparse voxels, while "SSM" uses learned 3D heatmap features. We also ablate our chroma loss (Habermann et al., 2021), using only MSE ("No Chroma Loss"). Results show K-means substantially outperforms SSM on sparse voxel volumes, while chroma loss significantly improves color fidelity (Figure 5 visualizes gray collapse without it).

We additionally ablate the number of particles $M$ (Table 2). Since each particle is low-dimensional, the main requirement is enough particles to cover the objects in the scene. Increasing from 16 to 24 improves reconstruction, but 40 particles yields no further gain. In practice, the model naturally ignores redundant particles: with $M=24$, only $\sim 15$ particles are active on average (transparency $z_t \approx 1$), though more complex scenes can require substantially more.

### 5.2. Imitation Learning with 3D Latent Particles

Our 3D object-centric decomposition discovers explicit, disentangled scene entities as compact 3D latent representa-

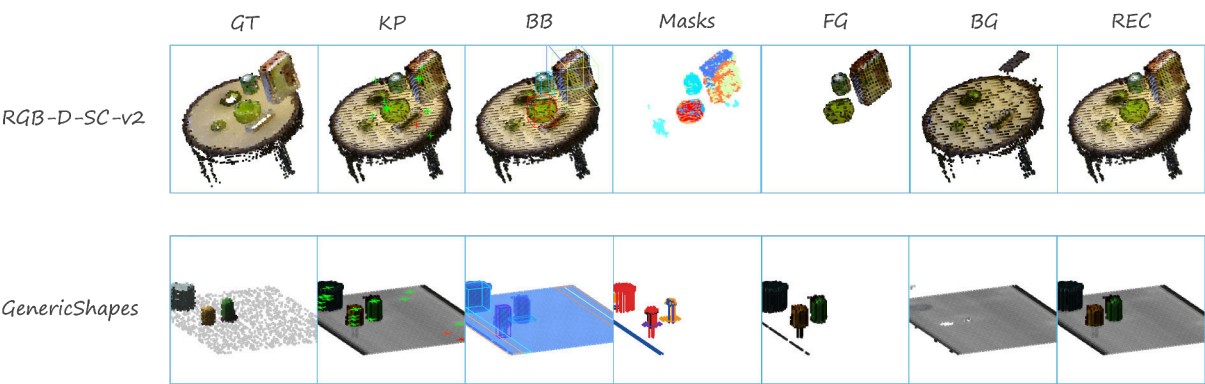

*Figure 2.* **3D-DLP-VC Scene Decomposition.** From input RGB voxels, 3D-DLP-VC infers latent particles with explicit attributes (keypoints, scales) and produces object/background masks entirely without supervision, compositing the input scene.

| Setting | Masked PSNR ↑ | IoU ↑ |
|---|---|---|
| **Keypoint Proposal** | | |
| SSM Raw | $13.52 \pm 1.28$ | $0.509 \pm 0.10$ |
| SSM | $15.06 \pm 1.67$ | $0.570 \pm 0.05$ |
| No Chroma Loss | $19.37 \pm 0.30$ | $0.785 \pm 0.03$ |
| **Full model** | $\mathbf{20.15 \pm 0.34}$ | $\mathbf{0.806} \pm 0.05$ |

*Table 3.* Ablations on RGB voxel reconstruction. "SSM Raw" and "SSM" replace the K-means keypoint proposals with spatial softmax applied to raw voxels and learned heatmaps, respectively, while "No Chroma Loss" uses only voxel-wise MSE reconstruction. The full model achieves the best reconstruction quality.

tions suitable for downstream tasks. Prior work demonstrates that 2D object-centric representations improve policy learning from images (Haramati et al., 2024; Qi et al., 2025). We evaluate whether the same benefit transfers, and is amplified, when the particles are truly 3D, by plugging 3D-DLP-VC tokens into a diffusion-based policy (Qi et al., 2025) and comparing to matched 2D and voxel baselines on two benchmarks.

**3D EC-Diffuser** We extend EC-Diffuser (Qi et al., 2025)—an entity-centric diffusion policy that jointly denoises future actions and particle states using a permutation-equivariant transformer—with several modifications, applied to all baselines that use EC-Diffuser: (i) a proprioceptive token $\mathbf{p}_t \in \mathbb{R}^{10}$ (end-effector position, 6D rotation (Zhou et al., 2019), and gripper scalar) that is denoised jointly with actions and particles; and (ii) support for language conditioning (e.g., for RLBench) via a frozen CLIP (Radford et al., 2021) language-token pathway. Full details are provided in Appendix E.

**MimicGen setup.** We use 12 multi-object, long-horizon tasks from MimicGen (Mandlekar et al., 2023), training a separate policy per task on 200 D0 demonstrations. Observations are fused from two static third-person cameras (agentview and sideview); no eye-in-hand input is

used by any method in our comparison. Point clouds are voxelized into a $64^3$ RGB grid (Appendix B), from which 3D-DLP-VC extracts particle tokens that are fed to the diffusion policy. All methods in this comparison—ours and the particle-based baselines—share the adapted EC-Diffuser backbone described above, including the proprioceptive token, so that the representation is the only variable. We evaluate with 50 rollouts per task across 3 random seeds and report mean ± std success rate.

**RLBench setup.** We additionally evaluate on 10 tasks from RLBench (James et al., 2020) drawn from the PerACT (Shridhar et al., 2022) subset, where each task is specified by a natural-language instruction and trained with 100 demonstrations. We follow the PerACT evaluation protocol of 25 rollouts per task. Policies use CLIP-based language embeddings.

**Baselines.** Baselines are designed to isolate the *representation* under a fixed policy backbone (EC-Diffuser with our adaptations); eye-in-hand RGB is disabled throughout to match our observation budget. *(i) 2D-DLP single-view + EC-Diffuser*: original EC-Diffuser representation, tokens from agentview only—isolates the 2D→3D lift. *(ii) 2D-DLP multi-view + EC-Diffuser*: per-camera tokens from both static cameras concatenated—controls for multi-view information alone. *(iii) EquiDiff (voxel-only)* (Wang et al., 2025): SE(3)-equivariant diffusion on $64^3$ RGB voxels—a strong dense-voxel reference. On RLBench we additionally report published PerACT (Shridhar et al., 2022) numbers as a language-conditioned voxel reference (not matched-compute). The rationale for omitting EC-Diffuser on raw voxels is that its full-attention architecture is computationally prohibitive for high-dimensional voxel inputs under our resource constraints; we provide further discussion in Appendix F.

**MimicGen results.** Table 4 reports per-task success rates. 3D-DLP-VC + EC-Diffuser achieves the highest

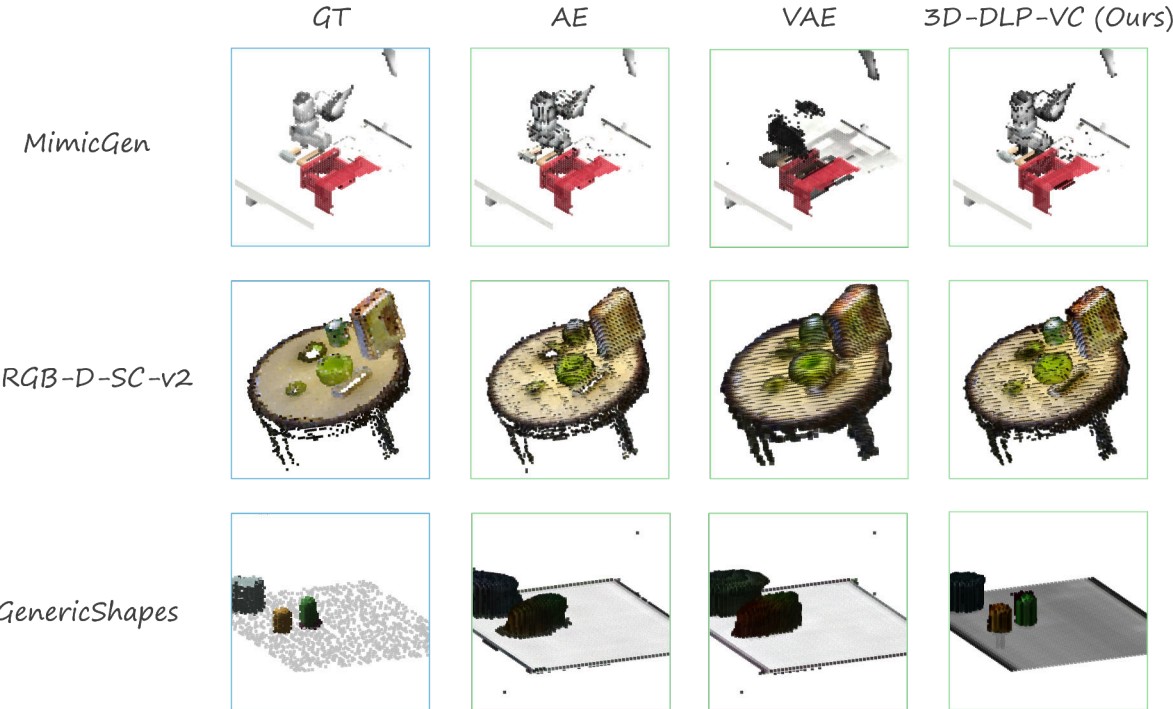

*Figure 3.* **RGB Voxel Reconstruction Comparison.** 3D-DLP-VC vs. non-object-centric baselines (AE: deterministic autoencoder; VAE: variational autoencoder) on input RGB voxels across the various datasets.

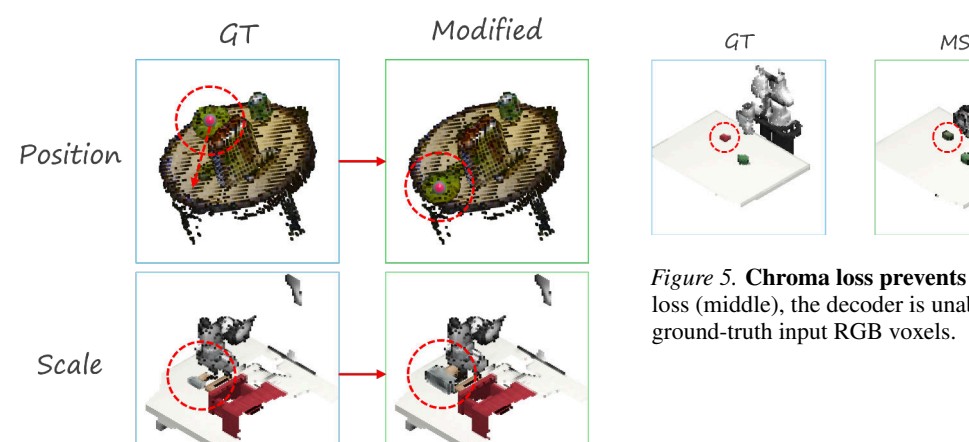

*Figure 4.* **Latent space controllability.** Modifying individual particle attributes–3D position (top) and scale (bottom)–directly translates to intuitive scene changes: translation and resizing.

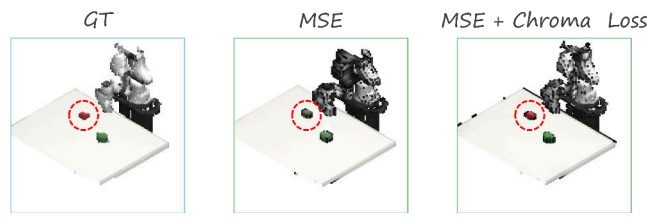

*Figure 5.* **Chroma loss prevents gray collapse.** Without chroma loss (middle), the decoder is unable to generate colors faithful to ground-truth input RGB voxels.

mean success rate (48.1% vs. 30.8% / 34.1% for 2D-DLP single/multi-view and 47.3% for EquiDiff voxel-only), winning 6 of 12 tasks (Stack, Stack Three, Hammer Cleanup, Mug Cleanup, Three Piece Assembly, and Square). Failure modes reveal limitations of the representation: on Coffee Preparation, the coffee cup is not cleanly isolated in the learned decomposition, capping downstream control. Decoded predicted-particle visualizations of the policy's imagined plans are in Appendix F.

**RLBench results.** Table 5 reports per-task success rates on the 10-task PerACT subset. Among the three matched-compute methods, 3D-DLP wins **9** of **10** tasks with margins of 19–48 absolute points on the four largest wins, losing only on Close Jar where the small axis-aligned target favors higher-resolution per-view 2D-DLP tokens. Against PerACT—a language-conditioned $64^3$ voxel policy that is not a matched-compute comparison—3D-DLP surpasses it on **7** of **10** tasks. PerACT's three wins (Close Jar, Stack Blocks, Push Buttons) play to its inductive bias: dense voxel coverage for small/precise targets, and a keypose-classification head for multi-step discrete actions.

| Method | Stack | Stack Three | Nut Asmbl. | Coffee |
|---|---|---|---|---|
| **3D-DLP (Ours)** | **94.6 ± 0.9** | **70.0 ± 1.6** | 6.0 ± 1.6 | 36.0 ± 1.6 |
| 2D-DLP single-view | 70.0 ± 4.9 | 18.0 ± 5.9 | 10.0 ± 3.3 | 72.7 ± 3.4 |
| 2D-DLP multi-view | 78.0 ± 2.8 | 14.7 ± 6.2 | 8.7 ± 2.5 | **82.0 ± 2.8** |
| EquiDiff (Voxel Only) | 82.0 ± 0.0 | 12.7 ± 5.7 | **10.7 ± 2.1** | 70.7 ± 3.4 |

| Method | Pick Place | Coffee Prep. | Hammer Cl. | Mug Cl. |
|---|---|---|---|---|
| **3D-DLP (Ours)** | 0.0 ± 0.0 | 0.0 ± 0.0 | **94.6 ± 0.9** | **64.0 ± 4.3** |
| 2D-DLP single-view | 0.0 ± 0.0 | 0.0 ± 0.0 | 55.3 ± 2.5 | 33.3 ± 5.3 |
| 2D-DLP multi-view | 4.7 ± 2.5 | 0.0 ± 0.0 | 66.7 ± 2.5 | 34.7 ± 7.5 |
| EquiDiff (Voxel Only) | **12.7 ± 2.2** | **34.0 ± 4.9** | 92.6 ± 3.3 | 34.0 ± 2.8 |

| Method | Kitchen | Three Pc. Asmbl. | Threading | Square |
|---|---|---|---|---|
| **3D-DLP (Ours)** | 86.7 ± 3.4 | **38.0 ± 4.0** | 36.0 ± 1.6 | **51.3 ± 0.9** |
| 2D-DLP single-view | 0.0 ± 0.0 | 33.3 ± 0.9 | 36.0 ± 1.6 | 41.3 ± 4.1 |
| 2D-DLP multi-view | 0.0 ± 0.0 | 29.3 ± 7.7 | **45.3 ± 6.8** | 45.3 ± 10.5 |
| EquiDiff (Voxel Only) | **95.0 ± 2.5** | 31.3 ± 3.7 | 42.0 ± 0.0 | 50.0 ± 2.8 |

*Table 4.* Imitation learning on 12 `MimicGen` tasks. 2D vs. 3D representations under the same policy (EC-Diffuser). All methods use 200 D0 demonstrations per task and do *not* use eye-in-hand input. We report mean ± std success rates (%) over 3 seeds of 50 rollouts. Bold indicates the best per task. 3D-DLP wins 6 of 12 tasks with the highest mean success rate (48.1%).

| Method | Close Jar | Open Drawer | Sweep to Dustpan | Meat Off Grill | Turn Tap |
|---|---|---|---|---|---|
| **3D-DLP (Ours)** | 16.0 ± 3.3 | **90.0 ± 1.6** | **100.0 ± 0.0** | **93.3 ± 1.9** | **94.7 ± 1.9** |
| 2D-DLP single-view | 30.7 ± 3.8 | 86.7 ± 6.8 | 97.3 ± 3.8 | 92.0 ± 3.3 | 69.3 ± 6.8 |
| 2D-DLP multi-view | 29.3 ± 3.8 | 88.0 ± 3.3 | 96.0 ± 3.3 | 89.3 ± 5.0 | 77.3 ± 1.9 |
| PerAct ($64^3$ voxels, multi-view) | **55.2 ± 4.7** | 88.0 ± 5.7 | 52.0 ± 0.0 | 70.4 ± 2.0 | 88.0 ± 4.4 |

| Method | Slide Block | Put in Drawer | Drag Stick | Push Buttons | Stack Blocks |
|---|---|---|---|---|---|
| **3D-DLP (Ours)** | **93.3 ± 3.8** | **97.3 ± 1.9** | **93.3 ± 1.9** | 57.3 ± 3.8 | 10.0 ± 0.0 |
| 2D-DLP single-view | 89.3 ± 1.9 | 94.7 ± 1.9 | 86.7 ± 1.9 | 18.7 ± 1.9 | 1.3 ± 1.9 |
| 2D-DLP multi-view | 81.3 ± 3.8 | 93.3 ± 1.9 | 84.0 ± 9.8 | 28.0 ± 6.5 | 5.3 ± 5.0 |
| PerAct ($64^3$ voxels, multi-view) | 74.0 ± 13.0 | 51.2 ± 4.7 | 89.6 ± 4.1 | **92.8 ± 3.0** | **26.4 ± 3.2** |

*Table 5.* Imitation learning on 10 `RLBench` tasks. We report 3D-DLP, 2D-DLP single-view, 2D-DLP multi-view, and PerACT (a language-conditioned $64^3$ voxel policy; not a matched-compute comparison). Values are success rate (%) reported as mean ± std. Bold marks the best per task across all four methods: 3D-DLP wins 7 of 10 tasks, PerACT wins the remaining 3 (Close Jar, Push Buttons, Stack Blocks).

## 6. Conclusion

In this work, we introduced 3D Deep Latent Particles (3D-DLP), a principled approach for learning self-supervised, object-centric representations directly from 3D observations. We demonstrated superior reconstructions on synthetic and real-world datasets, intuitive scene editing through explicit particle attribute modifications, and significant gains in complex multi-object robotic manipulation when integrating 3D-DLP representations with diffusion-based policies, establishing their value for decision-making. Extending 3D-DLP to dynamics and world modeling (Daniel et al., 2026) presents promising future work.

**Limitations.** Our voxelization approach incurs higher memory demands than point clouds; learning directly from raw point clouds remains future work. Like 2D DLP, 3D-DLP excels on datasets with recurring object types and static backgrounds, but scaling to highly dynamic, diverse real-world scenes with novel objects and cluttered, moving backgrounds presents important challenges for future research. Additionally, while our third-person 3D representations compete impressively without eye-in-hand cameras, integrating in-hand observations could further boost fine-grained manipulation performance.

## Acknowledgments

This material is based upon work supported by ONR MURI N00014-24-1-2748.

## Impact Statement

This paper advances representation learning for robotics through a compact, object-centric 3D latent state, with potential to improve reliability and generalization in robot learning systems. We foresee no direct negative societal consequences beyond the standard considerations for safely deploying learned robotic policies.

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

# A. 3D Deep Latent Particles (3D-DLP) – Extended Method Details

We aim to learn a self-supervised, object-centric representation of 3D scenes that is both compact and structured, supporting two key capabilities: (1) *scene decomposition*: disentangling objects from background, and (2) *decision-making*: providing a low-dimensional state representation for downstream policies. Given a 3D observation $\mathbf{x}$ (an RGB-D image, occupancy voxel grid, or RGB voxel grid), our model infers a set of $M$ foreground latent *particles* $\{z_{\text{fg}}^m\}_{m=1}^M$ and a single background latent $z_{\text{bg}}$. Each foreground particle corresponds to a localized entity in the scene and encodes both an explicit 3D spatial location and learnable geometric and visual attributes.

**Input modalities.** We consider three structured 3D sensing modalities. **RGB-D images** are represented as $\mathbf{x} \in \mathbb{R}^{4 \times H \times W}$ with channels $(R, G, B, D)$, where $H$ and $W$ denote image height and width, and $D$ denotes depth. **Occupancy voxels** are represented as $\mathbf{x} \in \{0, 1\}^{1 \times D_v \times H_v \times W_v}$, derived from geometry-only point clouds, where each voxel indicates binary occupancy. **RGB voxels** are represented as $\mathbf{x} \in [0, 1]^{3 \times D_v \times H_v \times W_v}$, derived from colored point clouds with RGB color values per occupied voxel. For voxel grids, $(D_v, H_v, W_v)$ denote the resolution along the depth, height, and width axes, respectively.

**Latent particle parameterization.**  Following Section 3, each foreground particle $\mathbf{z}_{\text{fg}}$ is parameterized as

$$\mathbf{z}_{\text{fg}} = [\mathbf{z}_p, \mathbf{z}_s, \mathbf{z}_c, \mathbf{z}_t, \mathbf{z}_f] \in \mathbb{R}^{d_p + d_s + 2 + d_{\text{obj}}},$$

where $d_p = d_s = 2$ for RGB-D images and $d_p = d_s = 3$ for voxel inputs. The position $\mathbf{z}_p \sim \mathcal{N}(\mu_p, \sigma_p^2) \in \mathbb{R}^{d_p}$ encodes spatial keypoint coordinates, and the scale $\mathbf{z}_s \sim \mathcal{N}(\mu_s, \sigma_s^2) \in \mathbb{R}^{d_s}$ encodes bounding-box dimensions. The composition order $\mathbf{z}_c \sim \mathcal{N}(\mu_c, \sigma_c^2) \in \mathbb{R}$ encodes the rendering order for resolving particle overlap, the transparency $\mathbf{z}_t \sim \text{Beta}(a, b) \in [0, 1]$ controls particle transparency, and the visual features $\mathbf{z}_f \sim \mathcal{N}(\mu_f, \sigma_f^2) \in \mathbb{R}^{d_{\text{obj}}}$ encode local appearance. The background is represented by a single latent

$$\mathbf{z}_{\text{bg}} \sim \mathcal{N}(\mu_{\text{bg}}, \sigma_{\text{bg}}^2) \in \mathbb{R}^{d_{\text{bg}}},$$

which is spatially anchored at the center and encodes global background appearance.

**Model components.** All 3D-DLP variants share a common three-stage pipeline, with modality-specific implementations for images and voxels.

*Prior (keypoint proposals).* Given a raw 3D observation, the prior proposes $M$ candidate particle locations. For RGB-D inputs, we use a learned keypoint module based on a spatial softmax (SSM (Jakab et al., 2018)), applied to convolutional feature maps to obtain 2D keypoint coordinates. For voxel inputs, we instead compute $M$ anchor locations using K-means clustering (Hartigan & Wong, 1979) over occupied voxels, which serve as 3D keypoint proposals.

*Encoder (particle latents).* Given the proposed keypoint locations $\{\bar{\mathbf{z}}_p^m\}_{m=1}^M$, an encoder refines them and predicts full particle latents. We first extract a canonical local neighborhood around each proposal using a differentiable spatial transformer (STN (Jaderberg et al., 2015)), then encode it into stochastic attributes. Concretely, the encoder predicts a *stochastic offset* $\Delta \mathbf{z}_p^m$ and forms the final position

$$\mathbf{z}_p^m = \bar{\mathbf{z}}_p^m + \Delta \mathbf{z}_p^m,$$

together with the remaining attributes: scale $\mathbf{z}_s^m$, transparency $\mathbf{z}_t^m$, visual appearance features $\mathbf{z}_f^m$, and a composition-order latent $\mathbf{z}_c^m$ that parameterizes the rendering order of overlapping particles.

*Decoder (glimpse composition).* The decoder maps each particle latent to a spatial *glimpse* (a local appearance map) in the observation space. All glimpses are then spatially transformed and composited on a global canvas according to their positions, scales, composition orders, and transparencies, together with the decoded background latent, to reconstruct the input observation.

In the following, we describe each modality variant.

## A.1. 3D-DLP-D: 3D Deep Latent Particles from RGB-D

We now describe how DLP is extended to RGB-D observations $\mathbf{x} \in \mathbb{R}^{4 \times H \times W}$, where the first three channels encode RGB color values and the fourth channel contains a depth map $D$ representing the distance from the camera's center of projection to each corresponding point in the 3D scene.

### A.1.1. PRIOR

Following DLP (Daniel & Tamar, 2024), the prior module proposes keypoint positions in a learnable manner by applying a patch-wise convolutional neural network (CNN) to the input, followed by a spatial softmax (SSM (Jakab et al., 2018)) layer. For 3D-DLP-D, the CNN input is extended from three to four channels to incorporate the depth channel.

Given an RGB-D frame $\mathbf{x}_t \in \mathbb{R}^{4 \times H \times W}$, we partition the image into a regular grid of $P \times P = N_{\text{patch}}$ non-overlapping patches and apply a small CNN encoder independently to each patch. For each patch, the CNN produces a single-channel feature map, and the SSM converts it into a 2D keypoint proposal (mean) $\bar{\mathbf{z}}_p$ and an associated uncertainty (covariance) in patch-local coordinates. These patch-local proposals are then transformed back to global image coordinates, yielding a pool of $N_{\text{patch}}$ candidate keypoints. In the encoder stage (Section A.1.2), a stochastic offset is predicted for each proposal, and the combined uncertainty from the keypoint proposals and the offset is used to select the top $M$ particles.

### A.1.2. ENCODER

We now describe the particle encoder, which defines the approximate posterior in 3D-DLP-D.

**Particles attributes encoding.** Given the $N_{\text{patch}}$ keypoint proposals from the prior, a CNN-based attribute encoder with four input channels takes local glimpses around these proposals, extracted using a spatial transformer network (STN (Jaderberg et al., 2015)), and predicts the latent attributes defined in Section A. Concretely, for each proposal $\bar{\mathbf{z}}_p$ the encoder outputs a stochastic offset $\Delta \mathbf{z}_p$, scale $\mathbf{z}_s$, and transparency $\mathbf{z}_t$, and forms the final particle position $\mathbf{z}_p = \bar{\mathbf{z}}_p + \Delta \mathbf{z}_p$. Following the particle selection procedure used in DLP (Daniel & Tamar, 2024), we combine the uncertainties from the proposal and the offset to obtain a confidence measure over keypoints and select $M$ keypoints $\{\mathbf{z}_{\text{fg}}^m\}_{m=1}^M$ as the final positions for the foreground particles. All attributes are learned jointly from both RGB and depth channels.

**Particles appearance encoding.** For each selected particle position $\mathbf{z}_p^m$, we apply an STN to extract an RGB-D glimpse and encode appearance separately for color and depth. Specifically, two small CNN encoders produce initial RGB features $\bar{\mathbf{z}}_f^{m,\text{rgb}}$ and depth features $\bar{\mathbf{z}}_f^{m,\text{depth}}$. Encoding RGB and depth in separate branches encourages an additional degree of disentanglement between photometric and geometric cues, which we find beneficial in our ablations (Section C). In parallel, for particles with non-zero transparency $\mathbf{z}_t^m > 0$, we mask out their surrounding regions in the input and feed the remaining pixels to background encoders, yielding initial background RGB and depth features $\bar{\mathbf{z}}_{\text{bg}}^{\text{rgb}}$ and $\bar{\mathbf{z}}_{\text{bg}}^{\text{depth}}$. At this stage, all appearance latents are deterministic.

**Composition order and interaction features encoding.** As in DLP (Daniel & Tamar, 2024), we model interactions between particles and the background using an attention-based interaction encoder. This module takes all the learned attributes and the deterministic particle and background appearance features and outputs (i) a stochastic composition order variable $\mathbf{z}_c$ for each particle, controlling the rendering order for overlapping particles, and (ii) a stochastic modulation $\Delta \mathbf{z}_f$ of the appearance features, producing final visual latents $\mathbf{z}_f = \bar{\mathbf{z}}_f + \Delta \mathbf{z}_f$ (and similarly for $\mathbf{z}_{\text{bg}}$). By allowing particles to exchange information with one another and with the background, the interaction encoder helps resolve overlaps and refine occlusion boundaries, leading to cleaner and more coherent reconstructions (Daniel & Tamar, 2024).

### A.1.3. DECODER

In the following, we detail the decoder which defines the likelihood in 3D-DLP.

**Particle decoder.** Each foreground particle is decoded independently into local RGB-D glimpses. Two small upsampling CNNs map the RGB-related features $\mathbf{z}_f^{m,\text{rgb}}$ to an RGBA glimpse $\tilde{x}_m^{p,\text{rgba}} \in \mathbb{R}^{4 \times P \times P}$ and the depth-related features $\mathbf{z}_f^{m,\text{depth}}$ to a depth glimpse $\tilde{x}_m^{p,\text{depth}} \in \mathbb{R}^{1 \times P \times P}$, representing the reconstructed appearance of particle $m$ in canonical patch coordinates. The RGB channels model color, the alpha channel provides a soft segmentation mask, and the depth channel gives per-pixel distance from the camera. Following the stitching mechanism in DLP, the composition order $\mathbf{z}_c$ and transparency $\mathbf{z}_t$ modulate the alpha mask, jointly determining the effective visibility and compositing order of each particle. The spatial attributes $(\mathbf{z}_p, \mathbf{z}_s)$ specify the particle's position and scale in the full image and are applied to both RGB and depth glimpses via a spatial transformer network (STN) to place them into a full-resolution RGB-D foreground canvas $\hat{x}_{\text{fg}}$.

**Background decoder.** The background latent $\mathbf{z}_{\text{bg}}$ is decoded with two separate upsampling CNNs for RGB and depth, producing a full-resolution RGB-D background image $\hat{x}_{\text{bg}}$.

**Reconstruction compositing.** The final reconstructed RGB-D image is obtained by alpha compositing the foreground and

background:

$$\hat{x} = \alpha \odot \hat{x}_{\text{fg}} + (1 - \alpha) \odot \hat{x}_{\text{bg}},$$

where $\alpha$ denotes the effective soft mask resulting from the particle-wise compositing process. For a more detailed description of the stitching procedure, we refer the reader to Daniel & Tamar (2024).

### A.1.4. Loss

Following DLP (Daniel & Tamar, 2024), 3D-DLP-D is trained as a variational autoencoder (VAE) by maximizing an evidence lower bound (ELBO) on RGB-D observations. We employ the same loss as in 2D RGB DLP, augmented with a reconstruction loss for the depth map and KL regularization for the depth appearance latents. For a single RGB-D frame $x = (x^{\text{rgb}}, x^{\text{depth}}) \in \mathbb{R}^{4 \times H \times W}$, the objective decomposes into an RGB-D reconstruction term, KL-divergence terms between inferred posteriors and fixed priors, and a sparsity regularizer:

$$\mathcal{L}_{\text{rgbd}} = \beta_{\text{rec}} \, \mathcal{L}_{\text{rec}}^{\text{rgbd}} + \beta_{\text{KL}} \, \mathcal{L}_{\text{KL}}^{\text{rgbd}} + \beta_{\text{obj}} \, \mathcal{L}_{\text{obj}}, \tag{1}$$

where $\beta_{\text{rec}}, \beta_{\text{KL}}$, and $\beta_{\text{obj}}$ are scalar weights (we use $\beta_{\text{rec}} = 1$ and typically set $\beta_{\text{KL}} = \beta_{\text{obj}}$).

**RGB-D reconstruction loss $\mathcal{L}_{\text{rec}}^{\text{rgbd}}$.** Let $\hat{x} = (\hat{x}^{\text{rgb}}, \hat{x}^{\text{depth}})$ denote the reconstructed RGB-D frame produced by compositing decoded particles and background (Section A.1.3). We use channel-wise mean squared error (MSE) over all pixels:

$$\mathcal{L}_{\text{rec}}^{\text{rgbd}} = \mathcal{L}_{\text{rgb}}\big(x^{\text{rgb}}, \hat{x}^{\text{rgb}}\big) + \mathcal{L}_D\big(x^{\text{depth}}, \hat{x}^{\text{depth}}\big), \tag{2}$$

where $\mathcal{L}_{\text{rgb}}$ is an MSE loss on the RGB channels and $\mathcal{L}_D$ is an MSE loss on the depth channel.

**KL-divergence loss $\mathcal{L}_{\text{KL}}^{\text{rgbd}}$.** Each foreground particle $m$ has posteriors over (i) a position offset $q(\Delta \mathbf{z}_p^m)$, (ii) a scale $q(\mathbf{z}_s^m)$, (iii) a composition-order variable $q(\mathbf{z}_c^m)$, (iv) a transparency variable $q(\mathbf{z}_t^m)$, and (v) appearance latents $q(\mathbf{z}_f^{m,\text{rgb}})$ and $q(\mathbf{z}_f^{m,\text{depth}})$ for RGB and depth, respectively. In addition, the background has an appearance posterior $q(\mathbf{z}_{\text{bg}})$. We place fixed priors on all latents—Gaussian priors for continuous variables and a Beta prior for transparency—and compute a *masked* KL so that inactive particles (with small $\mathbf{z}_t^m$) are not heavily penalized. The total KL can be written as

$$
\begin{aligned}
\mathcal{L}_{\text{KL}}^{\text{rgbd}} = {} & \mathcal{L}_{\text{KL}}^{\text{kp}} + \sum_{m=1}^{M} \mathbf{z}_t^m \, \text{KL}\big(q(\Delta \mathbf{z}_p^m) \,\|\, p(\Delta \mathbf{z}_p)\big) + \sum_{m=1}^{M} \mathbf{z}_t^m \, \text{KL}\big(q(\mathbf{z}_s^m) \,\|\, p(\mathbf{z}_s)\big) \\
& + \sum_{m=1}^{M} \mathbf{z}_t^m \, \text{KL}\big(q(\mathbf{z}_c^m) \,\|\, p(\mathbf{z}_c)\big) + \sum_{m=1}^{M} \text{KL}\big(q(\mathbf{z}_t^m) \,\|\, p(\mathbf{z}_t)\big) \\
& + \beta_f \sum_{m=1}^{M} \mathbf{z}_t^m \, \text{KL}\big(q(\mathbf{z}_f^{m,\text{rgb}}) \,\|\, p(\mathbf{z}_f^{\text{rgb}})\big) + \beta_f \sum_{m=1}^{M} \mathbf{z}_t^m \, \text{KL}\big(q(\mathbf{z}_f^{m,\text{depth}}) \,\|\, p(\mathbf{z}_f^{\text{depth}})\big) \\
& + \beta_f \text{KL}\big(q(\mathbf{z}_{\text{bg}}) \,\|\, p(\mathbf{z}_{\text{bg}})\big),
\end{aligned}
\tag{3}
$$

where $\mathcal{L}_{\text{KL}}^{\text{kp}}$ denotes the KL between the keypoint proposals and their prior (as in DLP), $\beta_f \leq 1$ is a weighting coefficient applied only to appearance feature KLs (as in DLP), and all priors are diagonal Gaussians or Beta distributions with fixed hyperparameters.

**Active-particle regularization $\mathcal{L}_{\text{obj}}$.** As in DLP-style models, we discourage solutions where many particles remain active by penalizing the total particle mass:

$$\mathcal{L}_{\text{obj}} = \left( \sum_{m=1}^{M} \mathbf{z}_t^m \right)^2. \tag{4}$$

This encourages sparsity in particle usage while still allowing the model to activate more particles when needed for complex scenes. The complete list of hyperparameters and prior settings is provided in Section G.

## A.2. 3D-DLP-V: 3D Deep Latent Particles from Occupancy Voxels

In 3D settings, observations are often represented as point clouds $\mathcal{P} = \{\mathbf{q}_i\}_{i=1}^{N}$, where each $\mathbf{q}_i \in \mathbb{R}^3$ denotes 3D coordinates $(z, y, x)$ and $N$, the number of points, varies across scenes, leading to *variable cardinality* of the point set. While point clouds preserve fine geometric detail, this variable cardinality and the absence of a canonical grid-based tensor layout make it non-trivial to (1) batch examples efficiently; (2) apply translation-equivariant convolutional architectures, and (3) directly instantiate the DLP-style pipeline of proposing keypoints, extracting local crops, and inferring particle attributes, which relies on spatially indexed feature maps and differentiable cropping.

To address these challenges, we adopt *voxelization*, arranging the point cloud into an occupancy grid $\mathbf{x} \in \{0, 1\}^{1 \times D \times H \times W}$, where each voxel is marked as occupied if at least one point falls inside it. This converts an irregular 3D point cloud into a dense, structured 3D tensor, a direct 3D analogue of a 2D image, such that each cell has a fixed spatial meaning. Voxels therefore provide a natural representation for extending 2D DLP to 3D: we replace 2D convolutions with 3D convolutions to obtain spatial feature volumes, extract $M$ particle-centered 3D crops with a 3D spatial transformer, and decode canonical cubic particle patches that are placed back into the global grid. In summary, voxelization sacrifices a small amount of geometric fidelity in exchange for a stable, grid-aligned representation that makes 3D keypoint and particle inference, as well as differentiable per-particle rendering, considerably simpler and more computationally efficient.

Crucially, because the latent space is explicitly 3D, we no longer need the composition-order variable $z_c$ used in 2D DLP and 3D-DLP-D. In 2D projections, $z_c$ approximates occlusion relations via stitching order; with true 3D coordinates and volumetric rendering, occlusions are naturally resolved during compositing, simplifying the particle latent space and eliminating a source of inductive bias.

**Voxel grid construction.** We define an *axis-aligned bounding box* (AABB) workspace in 3D, specified by its minimum and maximum corners $\mathbf{p}_{\min}, \mathbf{p}_{\max} \in \mathbb{R}^3$. This volume is discretized into a regular grid of size $D \times H \times W$, and we denote voxel indices by $\mathbf{u} = (u_z, u_y, u_x)$, where $u_z \in \{0, \dots, D-1\}$, $u_y \in \{0, \dots, H-1\}$, and $u_x \in \{0, \dots, W-1\}$.

A 3D point $\mathbf{q} \in \mathbb{R}^3$ is mapped to a voxel index by linearly normalizing it into the AABB and then discretizing:

$$\phi(\mathbf{q}) = \left\lfloor \left( \frac{\mathbf{q} - \mathbf{p}_{\min}}{\mathbf{p}_{\max} - \mathbf{p}_{\min}} \right) \odot (D-1, H-1, W-1) \right\rfloor .$$

For example, if $\mathbf{q}$ lies exactly at the center of the AABB, then the normalized term is $(0.5, 0.5, 0.5)$ and the corresponding voxel index is approximately the center of the grid, i.e., $\phi(\mathbf{q}) \approx (\lfloor (D-1)/2 \rfloor, \lfloor (H-1)/2 \rfloor, \lfloor (W-1)/2 \rfloor)$.

The binary occupancy grid $\mathbf{x} \in \{0, 1\}^{1 \times D \times H \times W}$ is then defined as

$$\mathbf{x}(\mathbf{u}) \;=\; \mathbb{I}\big[\, |\{i \;:\; \phi(\mathbf{q}_i) = \mathbf{u}\}| > 0 \,\big], \tag{5}$$

where $\mathbb{I}[\cdot]$ is the indicator function, returning 1 if its argument is true and 0 otherwise. The set

$$\{i \;:\; \phi(\mathbf{q}_i) = \mathbf{u}\}$$

collects all point indices whose voxel index equals $\mathbf{u}$, and its cardinality $|\cdot|$ counts how many points fall into voxel $\mathbf{u}$. Thus $\mathbf{x}(\mathbf{u}) = 1$ if at least one point from the point cloud is mapped to voxel $\mathbf{u}$, and $\mathbf{x}(\mathbf{u}) = 0$ otherwise.

Next, we describe how the DLP components are adapted to account for the aforementioned 3D considerations.

### A.2.1. PRIOR

Voxel grids are typically sparse (most entries are zero) and discontinuous (with sharp occupied/empty boundaries). In this regime, learning stable keypoint heatmaps early in training can be unreliable, so directly reusing the SSM-based keypoint prior from 2D-DLP and 3D-DLP-D (Section A.1) with 3D CNNs may yield very few salient detections. This forces a small set of keypoints to cover large portions of the scene and often leads to poor reconstructions. Instead, we adopt a simple geometry-driven prior based on K-means clustering (Hartigan & Wong, 1979) over occupied voxels.

We first map each occupied voxel index $\mathbf{u} = (u_z, u_y, u_x)$ to a normalized 3D coordinate $\mathbf{p}(\mathbf{u}) \in [-1, 1]^3$ using the same AABB workspace as in the voxelization step (Section A.2). Let

$$\mathcal{X} \;=\; \{\mathbf{p}(\mathbf{u}) \;:\; \mathbf{x}(\mathbf{u}) = 1\} \subset \mathbb{R}^3$$

denote the set of normalized coordinates of all occupied voxels. We then run K-means on $\mathcal{X}$ to obtain $K$ cluster centers $\{\bar{\mathbf{z}}_p^k\}_{k=1}^K$, which serve as keypoint proposals for particle positions.

**K-means initialization.** To avoid poor local minima and encourage coverage of distinct spatial regions, we use a K-means++-style (Arthur & Vassilvitskii, 2006) seeding scheme. We first choose the initial center uniformly at random from $\mathcal{X}$: $\bar{\mathbf{z}}_p^1 \sim \text{Uniform}(\mathcal{X})$.

Then, for $k = 2, \ldots, K$, we sample the next center from $\mathcal{X}$ with probability proportional to its squared distance to the nearest already chosen center:

$$\Pr\left(\bar{\mathbf{z}}_p^k = \mathbf{x} \in \mathcal{X}\right) \;\propto\; \min_{j<k} \left\|\mathbf{x} - \bar{\mathbf{z}}_p^j\right\|_2^2.$$

After initialization, we run a small, fixed number of iterations (we use $N_{\text{iter}} = 5$ in all experiments) to refine the centers. This makes the prior inexpensive and stable in practice, while still providing well-spread proposals. We provide a Pytorch-style code of this process in Figure 6.

**K-means cluster covariance.** In DLP, each keypoint proposal produced by the spatial softmax (SSM) module is associated with a covariance matrix derived from the corresponding heatmap, and this covariance is later combined with the position-offset variance to select the $M$ posterior particles (Daniel & Tamar, 2024). In 3D-DLP-V, we replace the heatmap-based covariance with an *intra-cluster covariance* computed directly from the occupied voxels assigned to each K-means cluster, so that more spatially compact clusters receive lower uncertainty.

Concretely, running K-means over the set of occupied coordinates yields clusters $\{\mathcal{I}_k\}_{k=1}^K$ and their centers $\{\boldsymbol{\mu}_k\}_{k=1}^K$, where

$$\boldsymbol{\mu}_k = \frac{1}{|\mathcal{I}_k|} \sum_{i \in \mathcal{I}_k} \mathbf{q}_i, \quad \mathbf{q}_i \in \mathbb{R}^3$$

are the normalized 3D coordinates of voxels in cluster $k$. We then define the empirical covariance of cluster $k$ as

$$\boldsymbol{\Sigma}_k = \frac{1}{|\mathcal{I}_k|} \sum_{i \in \mathcal{I}_k} (\mathbf{q}_i - \boldsymbol{\mu}_k)(\mathbf{q}_i - \boldsymbol{\mu}_k)^\top \;+\; \lambda I_3,$$

with a small term $\lambda I_3$ added for numerical stability when clusters contain few points. If a cluster is empty, we fall back to a default isotropic covariance. The resulting pairs $\{\boldsymbol{\mu}_k, \boldsymbol{\Sigma}_k\}_{k=1}^K$ provide both the keypoint proposals and their spatial uncertainty, directly replacing the SSM-derived means and covariances used in the 2D DLP variants

### A.2.2. ENCODER

In the occupancy voxel setting, the encoder, which models the approximate posterior, closely follows the 3D-DLP-D encoding pipeline (Section A.1.2) for an input volume $\mathbf{x} \in \{0, 1\}^{1 \times D \times H \times W}$, with the following key adaptations. First, all 2D convolutional networks are replaced by 3D CNNs operating on voxel grids. Second, the spatial transformer network (STN) uses trilinear sampling instead of bilinear sampling to extract local 3D glimpses in a differentiable manner. Third, the position $z_p$ and scale $z_s$ attributes become 3D vectors, parameterizing $(z, y, x)$ instead of 2D image coordinates. Fourth, as noted in Section A.2, the composition-order variable $z_c$ is no longer needed since 3D coordinates naturally resolve occlusions during volumetric rendering. Fifth, when selecting the $M$ posterior keypoints from the $K$ K-means proposals, the SSM-derived variance used in 2D DLP is replaced by the intra-cluster covariance introduced in Section A.2.1. Finally, the appearance features $z_f$ and background features $z_{\text{bg}}$ encode latent occupancy patterns only, as no RGB or depth channels are present in this modality. The encoder architecture is illustrated in Figure 8.

### A.2.3. DECODER

We now describe the decoder, which defines the likelihood in 3D-DLP-V.

**Particle decoder.** Each foreground particle is decoded independently into a local *canonical* occupancy patch. A 3D upsampling CNN maps the particle feature latent $\mathbf{z}_f^m$ to a cubic patch of occupancy logits $\tilde{\ell}_m \in \mathbb{R}^{1 \times P \times P \times P}$, where the patch resolution $P$ is chosen as a fixed fraction of the global voxel resolution, trading off local detail and compute. The logits are converted to per-voxel occupancy probabilities in canonical coordinates via $\tilde{\pi}_m = \sigma(\tilde{\ell}_m)$, where $\sigma(\cdot)$ denotes the sigmoid function. The spatial attributes $(\mathbf{z}_p^m, \mathbf{z}_s^m)$ specify the particle's 3D position and scale and are applied with a 3D spatial transformer using trilinear sampling to place each canonical patch into the global voxel grid, yielding placed logits $\ell_m(\mathbf{u})$ and probabilities $\pi_m(\mathbf{u}) = \sigma(\ell_m(\mathbf{u}))$ at voxel index $\mathbf{u}$.

```python
def kmeans(X, K, iters=5, tol=1e-4):
    # X: (N, 3) tensor of normalized 3D points
    device = X.device
    N = X.shape[0]

    # K-means++ initialization
    i0 = torch.randint(0, N, (1,), device=device)
    C = X[i0].clone() # first center
    while C.shape[0] < K:
        d2 = torch.cdist(X, C).pow(2).min(dim=1).values
        probs = (d2 + 1e-12) / (d2.sum() + 1e-12)
        i = torch.multinomial(probs, 1) # sample new center
        C = torch.cat([C, X[i]], dim=0)

    # K-means iterations
    for _ in range(iters):
        d2 = torch.cdist(X, C).pow(2)
        A = d2.argmin(dim=1) # assignments
        Cn = torch.stack([
            X[A == k].mean(dim=0) if (A == k).any() else C[k]
            for k in range(K)
        ], dim=0)
        shift = (Cn - C).norm(dim=1).mean()
        C = Cn
        if shift < tol:
            break

    # final assignments
    A = torch.cdist(X, C).pow(2).argmin(dim=1)
    return C, A
```

*Figure 6.* PyTorch-style implementation of the K-means prior used to obtain voxel-based keypoint proposals.

**Background decoder.** The background latent $\mathbf{z}_{\mathrm{bg}}$ is decoded by a separate 3D upsampling CNN into a full-resolution background occupancy field $\pi^{\mathrm{bg}}(\mathbf{u}) \in [0, 1]$.

**Reconstruction compositing.** Unlike RGB or RGB-D, occupancy is a Bernoulli field (occupied vs. empty) where a natural way to aggregate particles is via a probabilistic union. We gate each particle by its transparency variable $\mathbf{z}_t^m \in [0, 1]$ and combine the placed particle probabilities as

$$\pi^{\mathrm{obj}}(\mathbf{u}) = 1 - \prod_{m=1}^{M} \left(1 - z_t^m \pi_m(\mathbf{u})\right), \tag{6}$$

which corresponds to a *noisy-OR* over particles. For numerical stability, the product is evaluated in log-space. Defining $q_m(\mathbf{u}) = 1 - \mathbf{z}_t^m \pi_m(\mathbf{u}) \in (0, 1]$, we compute

$$\prod_{m=1}^{M} q_m(\mathbf{u}) = \exp\left(\sum_{m=1}^{M} \log q_m(\mathbf{u})\right) = \exp\left(\sum_{m=1}^{M} \log\left(1 - \mathbf{z}_t^m \pi_m(\mathbf{u})\right)\right). \tag{7}$$

The foreground objects are then combined with the decoded background via

$$\pi^{\mathrm{rec}}(\mathbf{u}) = 1 - \left(1 - \pi^{\mathrm{bg}}(\mathbf{u})\right)\left(1 - \pi^{\mathrm{obj}}(\mathbf{u})\right), \tag{8}$$

which treats foreground and background as independent Bernoulli sources for occupancy at each voxel. The noisy-OR compositing and background union can be implemented in a few lines of PyTorch, as shown in Figure 7

As occupancy probabilities are highly imbalanced (most voxels are empty), we initialize the bias of the final occupancy-logit layer to $\mathrm{logit}(p_0)$ for a small prior occupancy probability $p_0$ (we use $p_0 = 0.05$ in all experiments), which stabilizes optimization in the early training stages. The reconstruction process is illustrated in Figure 9.

```python
def decode_objects_occupancy(z_p, z_feat, z_scale):
    # z_p: [B,N,3] particle positions
    # z_feat: [B,N,D_f] particle features
    # returns occ_prob_per_obj: [B,N,1,D,H,W]
    patches = particle_dec(z_feat) # [B*N,1,Ps,Ps,Ps] logits
    B, N = z_p.shape[:2]
    patches = patches.view(B, N, 1, Ps, Ps, Ps)
    patches_t = translate_patches(z_p, patches, z_scale) # STN
    # patches_t: [B,N,1,D,H,W]
    occ_logits = patches_t
    occ_prob = torch.sigmoid(occ_logits)
    return occ_logits, occ_prob

def composite_occupancy(occ_prob, z_t, eps=1e-8):
    # occ_prob: [B,N,1,D,H,W] in [0,1]
    # z_t: [B,N] in [0,1], e.g. z_t[:,m] = z_t^m
    gate = z_t[:, :, None, None, None, None] # [B,N,1,1,1,1]
    p_k = torch.clamp(gate * occ_prob, 0.0, 1.0)
    log_1m = torch.log(torch.clamp(1.0 - p_k, min=eps))
    # p_obj(u) = 1 - prod((1 - z_t^m * p_m(u)))
    return 1.0 - torch.exp(log_1m.sum(dim=1, keepdim=True)) # [B,1,D,H,W]

def composite_with_background(p_obj, bg_logits):
    # p_obj: [B,1,D,H,W], bg_logits: [B,1,D,H,W]
    p_bg = torch.sigmoid(bg_logits)
    # p_rec(u) = 1 - (1 - p_bg(u)) * (1 - p_obj(u))
    p_rec = 1.0 - (1.0 - p_bg) * (1.0 - p_obj)
    return p_rec, p_bg
```

*Figure 7.* PyTorch-style implementation of voxel occupancy compositing.

### A.2.4. LOSS

Similarly to DLP (Daniel & Tamar, 2024), 3D-DLP-V is trained as a VAE by maximizing an evidence lower bound (ELBO), which we modify for the 3D setting as described next. For occupancy volumes, the likelihood is Bernoulli at each voxel, and the objective decomposes into a reconstruction term and KL-divergence terms for the inferred particle latents:

$$\mathcal{L}_{\text{occ}} = \beta_{\text{rec}} \, \mathcal{L}_{\text{rec}}^{\text{occ}} + \beta_{\text{KL}} \, \mathcal{L}_{\text{KL}}^{\text{occ}} + \beta_{\text{obj}} \, \mathcal{L}_{\text{obj}}, \tag{9}$$

where $\mathcal{L}_{\text{rec}}^{\text{occ}}$ is the occupancy reconstruction loss, $\mathcal{L}_{\text{KL}}^{\text{occ}}$ is the KL-divergence of the latent particles (similar to the RGB-D model in Sec. A.1.4 but without the composition-order term), and $\mathcal{L}_{\text{obj}}$ is the same active particle regularization term as in DLP with no changes. We set $\beta_{\text{rec}} = 1$ and $\beta_{\text{KL}} = \beta_{\text{obj}}$.

**Occupancy reconstruction loss $\mathcal{L}_{\text{rec}}^{\text{occ}}$.** Let $x(\mathbf{u}) \in \{0, 1\}$ denote the ground-truth occupancy at voxel index $\mathbf{u}$ and let $\pi^{\text{rec}}(\mathbf{u}) \in [0, 1]$ be the reconstructed occupancy probability obtained by compositing background and particles (Sec. A.2.3). To address the strong class imbalance (most voxels are empty), we optimize a *positive-class weighted* Bernoulli negative log-likelihood. In practice we use the numerically stable BCE-with-logits form: define logits $\ell^{\text{rec}}(\mathbf{u}) = \text{logit}(\pi^{\text{rec}}(\mathbf{u}))$ and

$$\mathcal{L}_{\text{wbce}} = \sum_{\mathbf{u}} \Big( - \alpha \, x(\mathbf{u}) \log \sigma(\ell^{\text{rec}}(\mathbf{u})) - (1 - x(\mathbf{u})) \log(1 - \sigma(\ell^{\text{rec}}(\mathbf{u}))) \Big), \tag{10}$$

where $\sigma(\cdot)$ is the sigmoid and $\alpha > 1$ upweights occupied voxels. We set $\alpha$ adaptively by computing the fraction of occupied voxels over the entire batch: $f = \mathbb{E}_{\mathbf{u}}[x(\mathbf{u})]$ as $\alpha = (1 - f)/f$, which we empirically found to result in better reconstructions.

We also add a soft Dice term (Sudre et al., 2017) on probabilities to directly encourage overlap between predicted and occupied sets:

$$\mathcal{L}_{\text{dice}} = 1 - \frac{2\langle \pi^{\text{rec}}, x \rangle + \epsilon}{\|\pi^{\text{rec}}\|_1 + \|x\|_1 + \epsilon}. \tag{11}$$

The total reconstruction loss is

$$\mathcal{L}_{\text{rec}}^{\text{occ}} = \mathcal{L}_{\text{wbce}} + \lambda_{\text{dice}} \mathcal{L}_{\text{dice}}, \tag{12}$$

with $\lambda_{\text{dice}} = 0.2$ and $\epsilon = 1 \times 10^{-6}$ in all experiments.

**KL-divergence loss $\mathcal{L}_{\text{KL}}^{\text{occ}}$.** The KL-divergence for the latent particles follows the same structure as the RGB-D model (Sec. A.1.4), with two changes: (1) no composition-order variable $z_c^m$ (thus no corresponding KL term), and (2) the prior keypoints $\mathcal{L}_{\text{KL}}^{\text{kp}}$ originate from K-means proposals rather than spatial-softmax (SSM):

$$
\begin{aligned}
\mathcal{L}_{\text{KL}}^{\text{occ}} = \mathcal{L}_{\text{KL}}^{\text{kp}} &+ \sum_{m=1}^{M} \mathbf{z}_t^m \, \text{KL}\big(q(\Delta \mathbf{z}_p^m) \,\|\, p(\Delta \mathbf{z}_p)\big) + \sum_{m=1}^{M} \mathbf{z}_t^m \, \text{KL}\big(q(\mathbf{z}_s^m) \,\|\, p(\mathbf{z}_s)\big) \\
&+ \sum_{m=1}^{M} \text{KL}\big(q(\mathbf{z}_t^m) \,\|\, p(\mathbf{z}_t)\big) \\
&+ \beta_f \sum_{m=1}^{M} \mathbf{z}_t^m \, \text{KL}\big(q(\mathbf{z}_f^m) \,\|\, p(\mathbf{z}_f)\big) + \beta_f \text{KL}\big(q(\mathbf{z}_{\text{bg}}) \,\|\, p(\mathbf{z}_{\text{bg}})\big),
\end{aligned}
\tag{13}
$$

where $\mathcal{L}_{\text{KL}}^{\text{kp}}$ denotes the KL between the keypoint proposals and their prior (as in DLP), $\beta_f \leq 1$ weights appearance feature KLs (as in DLP), and all priors are diagonal Gaussians or Beta distributions with fixed hyperparameters, reported in Section G.

## A.3. 3D-DLP-VC: 3D Deep Latent Particles from RGB Voxels

We now extend our framework to support color channels in the explicit 3D setting. Formally, let the RGB point cloud be

$$
\mathcal{P}^{\text{rgb}} = \{(\mathbf{q}_i, \mathbf{c}_i)\}_{i=1}^{N},
$$

where $\mathbf{q}_i \in \mathbb{R}^3$ denotes a 3D point and $\mathbf{c}_i \in [0,1]^3$ its RGB color. We discretize the workspace into a voxel grid $\mathbf{x} \in [0,1]^{3 \times D \times H \times W}$, where each voxel stores an aggregated color from the points that fall into it. Using the same axis-aligned bounding box and discretization map $\phi(\cdot)$ as in the occupancy case (Section A.2), we define, for each voxel $\mathbf{u}$ and color channel $c \in \{R, G, B\}$,

$$
\mathbf{x}^{(c)}(\mathbf{u}) = \begin{cases} \dfrac{1}{|\mathcal{I}(\mathbf{u})|} \displaystyle\sum_{i \in \mathcal{I}(\mathbf{u})} \mathbf{c}_i^{(c)} & \text{if } |\mathcal{I}(\mathbf{u})| > 0, \\ 0 & \text{otherwise,} \end{cases}
\tag{14}
$$

where $\mathcal{I}(\mathbf{u}) = \{ i : \phi(\mathbf{q}_i) = \mathbf{u} \}$ indexes all points whose coordinates are mapped to voxel $\mathbf{u}$ and $\mathbf{c}_i^{(c)}$ denotes the $c$-th color channel of point $i$. Intuitively, this assigns to each voxel the average RGB value of all points that fall inside it, and leaves voxels with no points black (zero color).

### A.3.1. PRIOR

For RGB voxels, we also use K-means clustering for keypoint proposals due to voxel sparsity. However, clustering solely on geometry can merge visually distinct nearby objects or over-allocate keypoints to large homogeneous surfaces. We therefore incorporate appearance information when proposing anchors.

Each occupied voxel $\mathbf{u}$ provides a color $\mathbf{c}(\mathbf{u}) \in [0,1]^3$ and normalized coordinate $\mathbf{p}(\mathbf{u}) \in [-1,1]^3$. We convert color to CIELAB space $\phi(\mathbf{c}(\mathbf{u})) = [L^*, a^*, b^*]$, which is perceptually uniform so Euclidean distances better reflect visual similarity than in RGB (Iizuka et al., 2016). We form a joint appearance-geometry feature

$$
\mathbf{f}(\mathbf{u}) = \big[\, \phi(\mathbf{c}(\mathbf{u})); \; \mathbf{p}(\mathbf{u}) \,\big] \in \mathbb{R}^6
$$

and *whiten* it across all candidate voxels by standardizing each of the 6 feature dimensions $j \in \{1, \ldots, 6\}$:

$$
\tilde{f}_j(\mathbf{u}) = \frac{f_j(\mathbf{u}) - \mu_j}{\sigma_j + \varepsilon},
$$

where $\mu_j = \mathbb{E}_{\mathbf{u}}[f_j(\mathbf{u})]$ and $\sigma_j = \text{Std}_{\mathbf{u}}[f_j(\mathbf{u})]$ are the per-dimension mean and standard deviation over occupied voxels. $\varepsilon = 1 \times 10^{-9}$ is added for numerical stability. This ensures color and position contribute equally to the clustering distance.

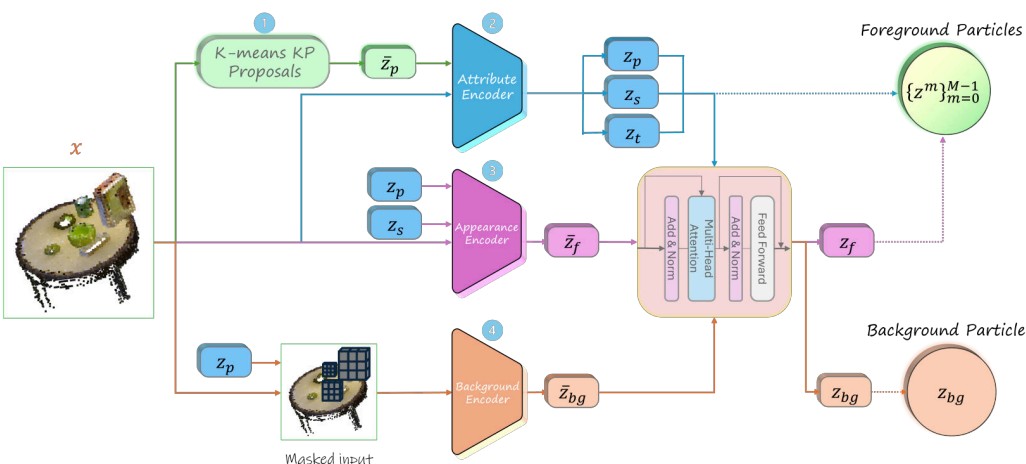

*Figure 8.* **3D-DLP-VC encoder architecture.** **(1)** K-means proposals $\bar{\mathbf{z}}_p$ are extracted from input voxels $x$. **(2)** An appearance encoder uses STN glimpses around proposals to predict refined positions $\mathbf{z}_p$, scales $\mathbf{z}_s$, and transparencies $\mathbf{z}_t$. **(3)** A second STN extracts initial appearance features $\bar{\mathbf{z}}_f$ from final particle crops. **(4)** In parallel, $\mathbf{z}_p$ mask the input for background encoder to produce $\bar{\mathbf{z}}_{\text{bg}}$. **(5)** An interaction encoder processes all attributes/features to output final $\mathbf{z}_f$ and $\mathbf{z}_{\text{bg}}$.

To bias proposals toward visually informative surface regions, we compute a nonnegative weight from lightness:

$$w(\mathbf{u}) = \max\big(L^*(\mathbf{u}), 0\big), \qquad \Pr(\mathbf{u}) = \frac{w(\mathbf{u})}{\sum_{\mathbf{u}' \in \Omega_{N_{\text{keep}}}} w(\mathbf{u}')}, \tag{15}$$

where $\Omega_{N_{\text{keep}}}$ contains the top-$N_{\text{keep}}$ occupied voxels ranked by $L^*$. We sample $n_{\text{samp}}$ voxels from this set according to $\Pr(\mathbf{u})$ and run K-means on their whitened features $\tilde{\mathbf{f}}(\mathbf{u})$. In all experiments, we set $N_{\text{keep}} = 4000$ and $n_{\text{samp}} = 2048$. Each resulting cluster $\mathcal{C}_k$ is converted to a geometric anchor via the weighted mean of coordinates:

$$\bar{\mathbf{z}}_p^k = \frac{\sum_{\mathbf{u} \in \mathcal{C}_k} w(\mathbf{u})\, \mathbf{p}(\mathbf{u})}{\sum_{\mathbf{u} \in \mathcal{C}_k} w(\mathbf{u})}.$$

This appearance-aware initialization produces particle centers that better align with object surfaces and boundaries in RGB voxel scenes, yielding more effective keypoint proposals than geometry-only clustering.

### A.3.2. ENCODER

The encoder for RGB voxels closely follows the occupancy voxel pipeline (Section A.2.2) with two adaptations. First, all 3D CNNs take 3 input channels instead of 1 to process the RGB voxel volume $\mathbf{x} \in [0,1]^{3 \times D \times H \times W}$. Second, the keypoint proposals are provided by the appearance-aware RGB voxel prior (Section A.3.1) rather than the geometry-only prior used for occupancy voxels. Note that here the encoded color channels remain in RGB space (unlike the CIELAB conversion used in the prior module), as the decoder directly generates RGB voxels. The encoder architecture is illustrated in Figure 8.

### A.3.3. DECODER

We now describe the decoder, which defines the likelihood for RGB voxel observations in 3D-DLP-VC. The architecture mirrors the occupancy voxel decoder but is adapted to generate RGB color channels.

**Particle decoder.** Each foreground particle $m$ is decoded independently into a canonical cubic RGBA patch. A 3D upsampling CNN maps the particle appearance latent $\mathbf{z}_f^m$ to an opacity field and RGB field

$$(\tilde{\alpha}_m, \tilde{\mathbf{c}}_m) \in [0,1]^{1 \times P \times P \times P} \times [0,1]^{3 \times P \times P \times P},$$

where $\tilde{\alpha}_m$ serves as a soft segmentation mask and $\tilde{\mathbf{c}}_m$ encodes local color in canonical coordinates. The spatial attributes $(\mathbf{z}_p^m, \mathbf{z}_s^m)$ specify the particle's 3D position and scale in the global grid and are applied via a 3D spatial transformer with

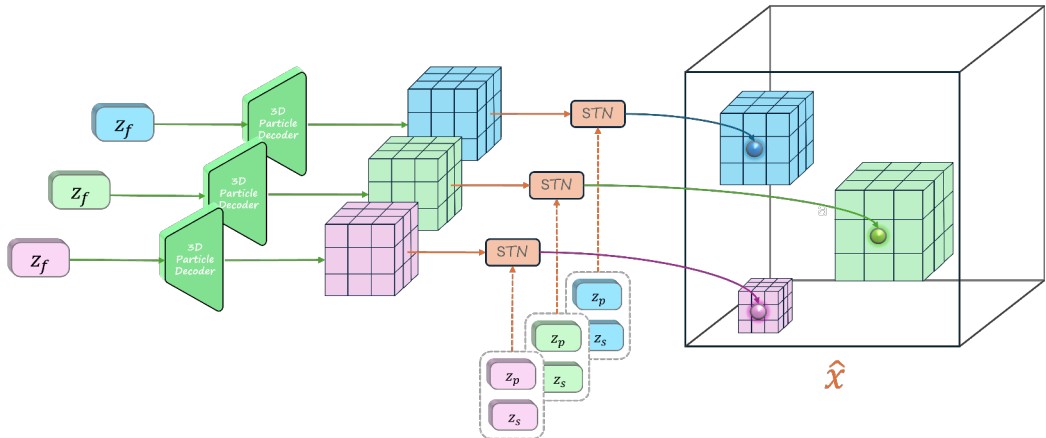

*Figure 9.* **3D-DLP-VC decoder architecture.** Each particle appearance latent $\mathbf{z}_f$ decodes to a canonical volume glimpse via 3D CNN. A 3D spatial transformer (STN) then uses spatial attributes $\mathbf{z}_s$ and $\mathbf{z}_p$ to scale and position the glimpse on the full-resolution canvas for the final reconstruction $\hat{x}$.

trilinear sampling to yield per-voxel fields

$$\alpha_m(\mathbf{u}) \in [0,1], \quad \mathbf{c}_m(\mathbf{u}) \in [0,1]^3$$

at voxel index $\mathbf{u}$. Each particle is gated by its transparency via

$$\bar{\alpha}_m(\mathbf{u}) = z_{\mathrm{t}}^m \alpha_m(\mathbf{u}).$$

**Background decoder.** The background latent $\mathbf{z}_{\mathrm{bg}}$ is decoded by a separate 3D upsampling CNN into a full-resolution background RGB volume

$$\mathbf{c}^{\mathrm{bg}}(\mathbf{u}) \in [0,1]^3.$$

**Reconstruction compositing.** Unlike RGB-D images that require depth-based occlusion ordering in 2D projection, RGB voxel grids admit a simpler per-voxel alpha mixture without explicit ordering. We compute normalized mixture weights from the gated alpha fields:

$$w_m(\mathbf{u}) = \frac{\bar{\alpha}_m(\mathbf{u})}{\sum_{j=1}^{M} \bar{\alpha}_j(\mathbf{u}) + \varepsilon}$$

where $\varepsilon = 1 \times 10^{-9}$, and reconstruct the foreground RGB volume via weighted summation:

$$\mathbf{c}^{\mathrm{obj}}(\mathbf{u}) = \sum_{m=1}^{M} w_m(\mathbf{u})\mathbf{c}_m(\mathbf{u}).$$

The background contribution is determined by a residual coverage mask:

$$m^{\mathrm{bg}}(\mathbf{u}) = 1 - \min\left(1, \sum_{m=1}^{M} \bar{\alpha}_m(\mathbf{u})\right),$$

yielding the final reconstruction

$$\mathbf{c}^{\mathrm{rec}}(\mathbf{u}) = m^{\mathrm{bg}}(\mathbf{u})\mathbf{c}^{\mathrm{bg}}(\mathbf{u}) + \left(1 - m^{\mathrm{bg}}(\mathbf{u})\right)\mathbf{c}^{\mathrm{obj}}(\mathbf{u}).$$

The reconstruction process is illustrated in Figure 9.

A.3.4. Loss

Similarly to DLP (Daniel & Tamar, 2024), our colored-voxels model 3D-DLP-VC is trained as a variational autoencoder (VAE) by maximizing an evidence lower bound (ELBO). For a single RGB voxel grid $\mathbf{x} \in \mathbb{R}^{3 \times D \times H \times W}$ the objective decomposes into:

$$\mathcal{L}_{\text{rgb-vox}} = \beta_{\text{rec}} \, \mathcal{L}_{\text{rec}}^{\text{rgb-vox}} + \beta_{\text{KL}} \, \mathcal{L}_{\text{KL}}^{\text{rgb-vox}} + \beta_{\text{obj}} \, \mathcal{L}_{\text{obj}}, \quad (16)$$

where $\mathcal{L}_{\text{KL}}^{\text{rgb-vox}} = \mathcal{L}_{\text{KL}}^{\text{occ}}$ is *identical* to 3D-DLP-V (Eq. (13)), $\mathcal{L}_{\text{obj}}$ is the unchanged DLP active particle regularizer, and we set $\beta_{\text{rec}} = 1$, $\beta_{\text{KL}} = \beta_{\text{obj}}$.

**RGB reconstruction loss $\mathcal{L}_{\text{rec}}^{\text{rgb-vox}}$.** We combine MSE with a chroma loss (Habermann et al., 2021) applied only on occupied (non-empty) voxels:

$$\mathcal{L}_{\text{rec}}^{\text{rgb-vox}} = \underbrace{\sum_{\mathbf{u}} \|\hat{\mathbf{x}}(\mathbf{u}) - \mathbf{x}(\mathbf{u})\|_2^2}_{\mathcal{L}_{\text{mse}}} + \lambda_{\text{chroma}} \underbrace{\sum_{\mathbf{u}} m(\mathbf{u}) \, \|\hat{\mathbf{C}}(\mathbf{u}) - \mathbf{C}(\mathbf{u})\|_2^2}_{\mathcal{L}_{\text{chroma}}}, \quad (17)$$

with balancing coefficient $\lambda_{\text{chroma}} = 500$. Here $\mathbf{u}$ indexes voxels and $m(\mathbf{u}) \in \{0, 1\}$ is the occupancy mask (Eq. (19)).

**Chroma loss.** Adapted from Habermann et al. (2021), chroma loss extracts *chrominance* (hue/saturation) by removing luminance (brightness). Per-voxel definitions are:

$$Y(\mathbf{u}) = \tfrac{1}{3} \sum_{c \in \{R,G,B\}} \mathbf{x}^{(c)}(\mathbf{u}), \quad \mathbf{C}(\mathbf{u}) = \mathbf{x}(\mathbf{u}) - Y(\mathbf{u})\mathbf{1},$$

$$\hat{Y}(\mathbf{u}) = \tfrac{1}{3} \sum_{c \in \{R,G,B\}} \hat{\mathbf{x}}^{(c)}(\mathbf{u}), \quad \hat{\mathbf{C}}(\mathbf{u}) = \hat{\mathbf{x}}(\mathbf{u}) - \hat{Y}(\mathbf{u})\mathbf{1}, \quad (18)$$

where $\mathbf{1} = [1, 1, 1]^{\top}$. $\mathcal{L}_{\text{chroma}}$ enforces color fidelity independent of brightness.

**Occupancy mask.** The mask $m(\mathbf{u}) \in \{0, 1\}$ is the *ground-truth occupancy channel* of the input RGBO voxel grid (RGB + occupancy), which identifies voxels that contain observed surface—both foreground objects and background. The chroma term is therefore evaluated only at occupied voxels, where color is defined, without relying on any magnitude threshold over RGB:

$$m(\mathbf{u}) = \mathbf{x}^{(O)}(\mathbf{u}) \in \{0, 1\}. \quad (19)$$

**Why chroma prevents gray collapse.** MSE alone can be minimized by matching luminance while predicting gray colors (zero chrominance). The luminance-invariant chroma term forces true color reproduction on foreground voxels, preventing this failure mode, as confirmed by our ablations (Sec. C).

# B. Datasets

We evaluate across four dataset families spanning simulated manipulation, controlled synthetic scenes, and real-world reconstructions. Throughout the paper, we consider three observation modalities: (i) **RGBD** (4-channel images), (ii) **occupancy voxels** (binary 3D grids), and (iii) **RGB voxels** (3-channel 3D grids). For voxel-based modalities, we first construct an RGB point cloud (either synthetic or fused from multi-view RGBD) and then voxelize it into a dense tensor of resolution $64^3$.

**Data formats and caching.** We store point clouds as `.ply` files with per-point XYZ and (optionally) RGB, and store voxelized scenes as `.pt` tensors together with metadata (workspace bounds `pmin`/`pmax`, voxel size, and grid shape) in a cached directory structure to enable fast loading during training and evaluation.

**Voxelization.** We voxelize point clouds to a $[64, 64, 64]$ grid with values in $[0, 1]$, indexed by voxel coordinates $\mathbf{u} = (u_z, u_y, u_x)$. For **occupancy voxels**, each voxel stores a binary value $x(\mathbf{u}) \in \{0, 1\}$ indicating empty (0) or occupied (1). For **RGB voxels**, each voxel stores a color vector $x(\mathbf{u}) \in [0, 1]^3$ corresponding to the RGB channels (aggregated via per-voxel averaging when multiple points fall in the same voxel).

## B.1. Simulated robotics benchmark: `MimicGen`

We use MimicGen (Mandlekar et al., 2023), which provides RoboSuite-based (Zhu et al., 2020) tabletop manipulation tasks with standardized demonstrations and environment-defined success metrics. We report results on tasks emphasizing object interaction and spatial reasoning (e.g., *Hammer Cleanup*, *Block Stacking*, *Coffee Preparation*). On `MimicGen` tasks, we evaluate models trained on **RGB-D**, **occupancy voxels**, and **RGB voxels**. For RGB-D, we train directly on the simulator's RGB-D observations. For voxel-based modalities, we fuse multi-view RGB point clouds from the RGBD observations of **two default static cameras**: `agentview` and `sideview`), and voxelize the resulting point clouds.

In the object discovery and scene reconstruction experiments on `MimicGen`, we train on 50 trajectories per task with an 80/20 train/eval split, totaling approximately 180,000 frames, while for policy learning we 200 trajectories per task. For each timestep, we back-project depth to 3D using camera intrinsics, transform points into a shared world frame using extrinsics, concatenate points across the two cameras, and crop to an axis-aligned workspace bounding box (AABB). This produces a fused RGB point cloud per frame, exported as a `.ply`. We voxelize each fused point cloud into a $64^3$ grid using a fixed Axis-Aligned-Bounding-Box (matching the crop bounds) or task-specific overrides. We save voxel tensors and per-sample metadata to a cache directory under each task, enabling efficient reuse across runs.

## B.2. Synthetic point clouds: `GenericShapes` and `ShapeNetScenes`

We generate synthetic tabletop point cloud scenes in two families: (i) `GenericShapes`: **primitive shapes** (e.g., cubes, spheres, cylinders) and (ii) `ShapeNetScenes`: **ShapeNet** (Chang et al., 2015) mesh priors. Each scene contains a random number of objects placed on a planar surface with non-overlapping footprints. We surface-sample each object and optionally add small Gaussian noise to emulate sensor noise. Scenes are exported as `.ply` point clouds and split into fixed train/val/test partitions. The primitive-shape generator samples objects from a fixed set of primitives, applies random scale and pose, places them collision-free on a table, and samples 3D points from object surfaces. We also generate an RGB-colored variant used for RGB-voxel experiments. For ShapeNet, we select a fixed set of object categories, randomly sample CAD models per category, normalize meshes into metric scale, place them on the table, and sample points from their surfaces. Each dataset contains 40,000 scenes with randomized object pose and scale.

## B.3. Synthetic RGB-D: `2DGenericShapes` and `BlenderShapes`

We additionally generate two synthetic RGB-D datasets used only for RGB-D representation learning: (i) `2DGenericShapes`: dataset formed by placing flat shapes in 3D and rendering RGB+depth, and (ii) `BlenderShapes`: Blender-rendered 3D shapes dataset with domain randomization. These datasets isolate RGB-D reconstruction behavior under controlled rendering conditions.

## B.4. Real-world: `UW RGB-D Scenes Dataset v2 (RGB-D-SD-v2)`

To test performance on real data, we use the `RGB-D-SD-v2` (Newcombe et al., 2015), which contains 14 RGB point cloud reconstructions of office spaces. Using the provided point cloud segmentation masks, we extract the tabletop region and objects on its surface, and generate an augmented corpus by rearranging objects on the table before voxelization.

**Tabletop rearrangement augmentation.** Given a labeled reconstructed scene, we synthesize diverse tabletop configurations by translating segmented object point clouds across the estimated table surface while enforcing simple collision constraints. Each scene provides a point cloud `XX.ply` and a per-point segmentation file `XX.label`. We identify the *table segment* as a label with sufficient support (at least 10,000 points) and planar extent (between 0.5m and 2.5m in XY span), and mark *background segments* (e.g., walls/floor) as labels with very large spatial extent (over 2.0m in any axis) or exceptionally large point count. All remaining labels are treated as object candidates. Since a single semantic label may contain multiple disconnected pieces, we further split each object candidate into connected components using voxel-grid connectivity (voxel size 2cm, 6-neighborhood BFS) and keep components with at least 500 points.

**Plane estimation and canonical table frame.** Let $\mathcal{T}$ denote the set of table points. We estimate the table plane in implicit form $\mathbf{n}^\top \mathbf{x} = d$ using RANSAC (Fischler & Bolles, 1981): we repeatedly sample three points, compute the candidate normal $\mathbf{n} \propto (\mathbf{p}_2 - \mathbf{p}_1) \times (\mathbf{p}_3 - \mathbf{p}_1)$, set $d = \mathbf{n}^\top \mathbf{p}_1$, and score the plane by the number of inliers whose point-to-plane distance is below 1cm. We run 200 iterations and keep the best plane. We then construct an orthonormal basis $(\mathbf{u}, \mathbf{v})$ spanning the plane and define UV coordinates by projection: $\mathrm{uv}(\mathbf{x}) = (\mathbf{u}^\top \mathbf{x}, \mathbf{v}^\top \mathbf{x})$. We orient the normal to point toward the objects by

```python
def fit_plane_ransac(points, n_iter=200, threshold=0.01):
    best_inliers, best_n, best_d = 0, None, None
    for _ in range(n_iter):
        p1, p2, p3 = points[np.random.choice(len(points), 3, replace=False)]
        n = np.cross(p2 - p1, p3 - p1)
        if np.linalg.norm(n) < 1e-6:  # degenerate
            continue
        n = n / np.linalg.norm(n)
        d = float(n @ p1)
        inliers = np.sum(np.abs(points @ n - d) < threshold)
        if inliers > best_inliers:
            best_inliers, best_n, best_d = inliers, n.astype(np.float32), d
    return best_n, best_d

def make_plane_frame(normal):
    n = normal / (np.linalg.norm(normal) + 1e-8)
    up = np.array([0,0,1], np.float32)
    if abs(up @ n) > 0.95: up = np.array([1,0,0], np.float32)
    u = np.cross(up, n); u = u / (np.linalg.norm(u) + 1e-8)
    v = np.cross(n, u); v = v / (np.linalg.norm(v) + 1e-8)
    return u, v

def move_object_on_plane(obj_xyz, n, d, u, v, target_uv):
    signed = obj_xyz @ n - d
    contact = obj_xyz[np.argmin(signed)]
    target_3d = u * target_uv[0] + v * target_uv[1] + n * d
    moved = obj_xyz + (target_3d - contact)[None, :]
    clearance = 0.003
    min_dist = float((moved @ n - d).min())
    if min_dist < clearance:
        moved = moved + n[None, :] * (clearance - min_dist)
    return moved

def check_collision(a_xyz, b_xyz, margin=0.02):
    amin, amax = a_xyz.min(0) - margin, a_xyz.max(0) + margin
    bmin, bmax = b_xyz.min(0), b_xyz.max(0)
    return bool(np.all(amax >= bmin) and np.all(bmax >= amin))
```

*Figure 10.* Core geometry used for real-world tabletop rearrangement: plane fitting via RANSAC, canonical UV frame construction, contact-point placement with clearance snapping, and collision-reject sampling with bounded retries.

checking the mean signed distance of object centroids (flipping $(\mathbf{n}, d)$ if needed). Finally, to align the plane with the *top* surface of the table, we shift $d$ so that the 98th percentile of signed distances of table points lies on the plane.

**Sampling collision-free placements.**    We compute robust tabletop UV bounds by projecting table points and taking the 2nd and 98th percentiles in each axis. For each object instance, we compute its UV footprint and sample a target UV uniformly from the feasible region after applying a 5cm boundary margin and accounting for half the footprint size. Given a target UV, we translate the object so that its *contact point* (minimum signed distance along $\mathbf{n}$) lands on the plane at the corresponding 3D location, then *snap* it to a small clearance (3mm) above the surface along $\mathbf{n}$. We reject placements that collide with previously placed objects using an expanded AABB overlap test (2cm margin), retrying up to 100 samples per object; if all retries fail, we keep the original pose. In the current implementation, we randomize object *translation and scale* on the table while preserving orientation.

**Real-world voxelization.**    We voxelize rearranged scenes to $64^3$ grids. In our implementation, we additionally apply a centering/scaling transform to ensure consistent coverage of the voxel grid across scenes, and aggregate RGB into voxels via per-voxel averaging.

| Setting | Masked PSNR ↑ | IoU ↑ |
|---|---|---|
| **Keypoint Proposal** | | |
| SSM Raw | $13.52 \pm 1.28$ | $0.509 \pm 0.098$ |
| SSM | $15.06 \pm 1.67$ | $0.570 \pm 0.053$ |
| No Chroma Loss | $19.37 \pm 0.30$ | $0.785 \pm 0.033$ |
| **Glimpse Ratio** | | |
| 0.125 | $11.75 \pm 1.28$ | $0.504 \pm 0.04$ |
| 0.0625 | $11.19 \pm 1.11$ | $0.527 \pm 0.04$ |
| **Full model** | $\mathbf{20.15 \pm 0.34}$ | $\mathbf{0.806 \pm 0.050}$ |

*Table 6.* Ablations on RGB voxel reconstruction.

| Dataset | PSNR ↑ | SSIM ↑ | LPIPS ↓ |
|---|---|---|---|
| Separate Depth/Feature Encoding | $34.964 \pm 0.16$ | $\mathbf{0.613 \pm 0.03}$ | $0.474 \pm 0.02$ |
| Unified Depth/Feature Encoding | $\mathbf{36.44 \pm 0.23}$ | $0.593 \pm 0.05$ | $\mathbf{0.447 \pm 0.02}$ |

*Table 7.* Ablation study of split vs unified depth and appearance encoding in 3D-DLP-D

## C. Extended Ablation Study

We split our ablation studies across the three modalities. We perform our ablation on the `MimicGen` dataset (`stack` task), with each model trained for 50 epochs.

**RGBD.** We ablate whether RGB and depth features are encoded jointly or separately. Concretely, we compare our default *split encoding* (predicting separate per-particle appearance codes $z_{f,\mathrm{rgb}}^m$ and $z_{f,\mathrm{depth}}^m$) to a *unified encoding* variant that follows the original 2D-DLP scheme (Daniel & Tamar, 2024), where we predict a single feature vector $z_f^m$ for the full 4-channel RGBD input. This choice also changes decoding and composition: instead of decoding RGB and depth patches separately and stitching them with modality-specific composition, the unified variant decodes a 4-channel RGBD patch per particle and stitches these patches directly to form the final RGBD observation. In a *unified* formulation, each particle has a single appearance feature $\mathbf{z}_f^m$ and a single decoder predicts all channels jointly (e.g., $\alpha$+RGB+D). We still use the same per-particle ordering/visibility weights (from $z_c^m$) when compositing RGB and depth into full-resolution outputs. Split features reduce competition between RGB and depth during optimization (RGB typically dominates gradients), which improves depth fidelity and stabilizes training in practice. The results for this ablation can be found in 7

**Occupancy voxels.** We ablate the reconstruction loss used for occupancy voxel prediction, comparing binary cross-entropy (BCE) against mean-squared error (MSE). Since all prior DLP works utilized MSE loss we considered it worth trying.

**RGB voxels.** For voxel inputs, a major design choice is the keypoint proposal mechanism. We ablate moving away from the learned encoder heatmaps + spatial softmax (SSM) used in 2D-DLP (Daniel & Tamar, 2022) and instead using K-means as the keypoint prior. Specifically, we compare: (i) K-means initialized keypoints, (ii) the original Encoder + SSM pipeline, and (iii) applying SSM directly on the raw voxel grid. The motivation for (iii) is that the encoder-based heatmap predictor may struggle in voxel grids due to the prevalence of empty space, whereas applying SSM on the raw occupancy structure may more directly highlight boundaries between occupied and unoccupied regions as peaked responses, potentially providing a useful keypoint prior.

We additionally ablate the particle glimpse size in the voxel DLP framework. The glimpse size controls how many voxels are contained in a particle's local encoding region. On RGB voxels, we evaluate two smaller glimpse sizes (0.125 and 0.0625; Table 6), compared to our default setting of 0.25.

**Chroma loss ablation.** We ablate our chroma loss (Habermann et al., 2021) for RGB voxels. As shown in Figure 5, chroma loss prevents gray collapse under extreme sparsity (most voxels are empty), dramatically improving color fidelity, and also confirmed quantitatively in Table 6.

| Setting | | IoU ↑ |
|---|---|---|
| MSE | — | $0.090 \pm 0.002$ |
| Full Model (BCE) | Ours | $\mathbf{0.2623 \pm 0.0394}$ |

*Table 8.* Ablations on Occupancy Voxel Reconstruction, comparing the usage of MSE and BCE in occupancy voxel object-centric reconstruction.

| Method | PSNR ↑ | SSIM ↑ | LPIPS ↓ |
|---|---|---|---|
| 3D-DLP-D (Ours) | $\mathbf{39.39 \pm 2.60}$ | $\mathbf{0.9891 \pm 0.0124}$ | $\mathbf{0.0089 \pm 0.009}$ |
| SAVi | $26.54 \pm 2.25$ | $0.9204 \pm 0.0356$ | $0.1089 \pm 0.0404$ |
| SLATE | $24.91 \pm 2.20$ | $0.88 \pm 0.04$ | $0.07 \pm 0.03$ |

*Table 9.* Comparison with slot-based object-centric methods on `MimicGen` RGB-D. 3D-DLP-D substantially outperforms SAVi and SLATE across all metrics.

## D. Additional Results

### D.1. Comparison with slot-based methods on RGB-D

To compare particle-based and slot-based object-centric representations, we evaluate 3D-DLP-D against SAVi (Kipf et al., 2022) and SLATE (Singh et al., 2022), adapted to support 4-channel RGB inputs, on the `MimicGen` RGB-D benchmark (Table 9). 3D-DLP-D substantially outperforms both slot-based methods across all metrics, suggesting that particle-based representations are better suited to our 3D setting.

Beyond the quantitative gap in Table 9, the qualitative decompositions in Figures 11 and 12 make the failure mode visible: across the `MimicGen` RGB-D scenes we examined, SAVi (Kipf et al., 2022) and SLATE (Singh et al., 2022) leave most of their slots empty or near-empty—only a handful of slots receive any signal, and even those tend to over-segment a single object across multiple slots or split an object's body from its shadow rather than isolating discrete scene entities. The reconstructions consequently miss or blur task-relevant objects (e.g., the coffee cup, the threading peg, the three-piece assembly parts). Two factors compound this. First, slot-based decomposition relies on iterative competitive assignment over a fixed slot budget, which is known to under-utilize slots when the scene's visual statistics are dominated by a uniform background (Seitzer et al., 2023)—a regime that describes the `MimicGen` tabletop. Second, extending these methods to volumetric inputs (occupancy or RGB voxels) is non-trivial: slot attention's pixel-level competition has no direct voxel analogue, transformer-based slot decoders scale poorly with $D{\times}H{\times}W$ token counts, and the architecture provides no natural mechanism for assigning slots to 3D positions. Our particle-based formulation sidesteps both issues—it anchors latents at local keypoints (avoiding global slot competition) and admits a direct 2D→3D lift via 3D convolutions and a 3D STN, which is what makes 3D-DLP-V/VC tractable in the first place.

**RGB-D Scene Decomposition and Reconstruction.** Table 10 presents quantitative results for our RGB-D variant, 3D-DLP-D, against non-object-centric baselines, where it substantially outperforms them. Figure 13 shows 3D-DLP-D scene decomposition results for both RGB and depth maps.

**Occupancy Voxels Scene Decomposition and Reconstruction.** Table 11 presents quantitative results for 3D-DLP-V against non-object-centric baselines. Our object-centric approach consistently leads or matches baselines across datasets. In Figure 14 we show 3D-DLP-V scene decomposition results on the various datasets, and qualitative scene reconstruction comparisons are shown in Figure 15.

**Latent modification.** In Figure 16 we present more latent modification results for translating and scaling the particles.

## E. Policy Learning via Entity-Centric Diffusion with 3D Particles

This section provides full details on the EC-Diffuser (Qi et al., 2025) adaptations: an overview of the EC-Diffuser backbone, the proprioceptive token added for robot-state-aware denoising, the removal of goal-image conditioning for per-task policies, and the language-token path used on `RLBench`.

**EC-Diffuser overview (Qi et al., 2025).** EC-Diffuser is a behavioral cloning method for multi-object manipulation that combines object-centric perception with diffusion-based sequence generation. It first encodes each image into a set of latent

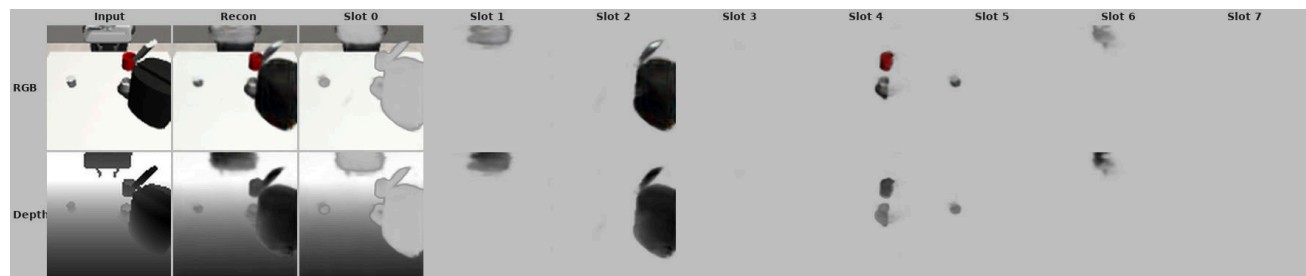

*(a)* SAVi on `coffee`.

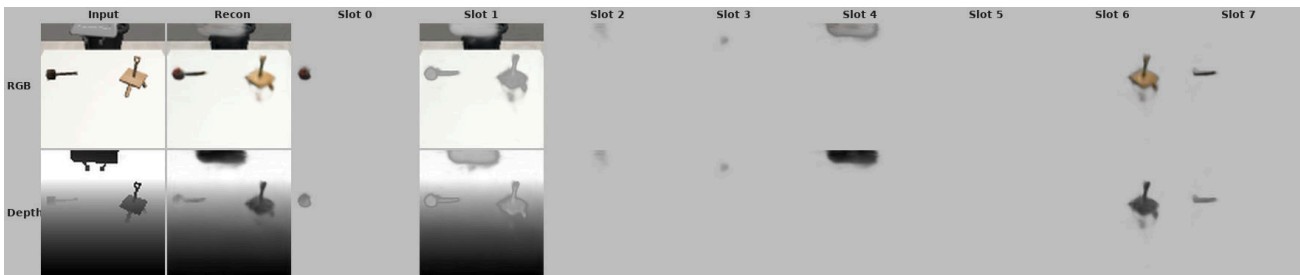

*(b)* SAVi on `threading`.

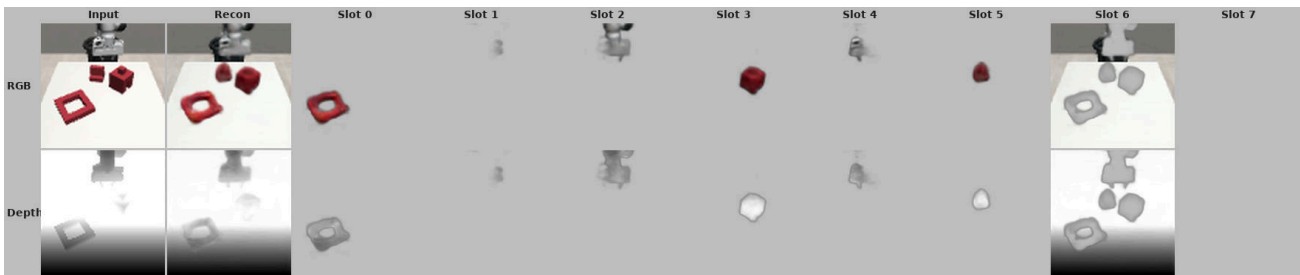

*(c)* SAVi on `three_piece_assembly`.

*Figure 11.* **SAVi decompositions on `MimicGen` RGB-D.** Each strip shows, left-to-right: input RGB-D (row 1: RGB, row 2: depth), the method's reconstruction, and the individual slot reconstructions (Slot 0–7). Across all three scenes, the majority of slots are blank or carry near-uniform mass; the few populated slots either fragment a single object or mix object and background, and the overall reconstruction loses task-relevant scene entities. See Figure 12 for the corresponding SLATE decompositions.

particles using DLP (Daniel & Tamar, 2024), replacing the standard single global feature vector. A permutation-equivariant, entity-centric Transformer (Vaswani et al., 2017) (PINT) then runs a diffusion process jointly over future particle states and continuous actions, conditioned on the current scene, and is executed in an MPC-style loop by taking the first denoised action at each step. This design captures multi-modal action distributions and the combinatorial structure that arises when manipulating several objects, while preserving invariance to object ordering. In our work, we replace the 2D latent particles with our learned 3D latent particles.

**Proprioceptive token.** Particle-based object-centric representations—whether 2D-DLP or 3D-DLP—are trained to decompose the scene and do not expose a structured robot end-effector state; the original EC-Diffuser leaves proprioception to be inferred indirectly from particle tokens. We therefore augment the PINT input sequence with a dedicated proprioceptive token that is shared across all particle-based variants (2D single-view, 2D multi-view, and 3D) so that representation is the only variable at test time. At each timestep $t$ we form

$$\mathbf{p}_t = \left[\, \mathbf{x}_t^{\text{eef}} \in \mathbb{R}^3, \ \mathbf{r}_t^{6\text{D}} \in \mathbb{R}^6, \ g_t \in \mathbb{R} \,\right] \in \mathbb{R}^{10},$$

where $\mathbf{x}_t^{\text{eef}}$ is the end-effector position, $\mathbf{r}_t^{6\text{D}}$ is the 6-D continuous rotation representation (Zhou et al., 2019) of the end-effector orientation (the first two columns of the rotation matrix), and $g_t$ is a normalized gripper-openness scalar obtained from the robot gripper joint positions. This token is projected into the PINT token dimension by a two-layer MLP with its own learned type embedding (distinct from the action and particle type embeddings), participates in the joint self-attention alongside the action token and the particle tokens, and is decoded back to $\mathbb{R}^{10}$ by a dedicated MLP head. Crucially, the

| Dataset | Method | PSNR ↑ | SSIM ↑ | LPIPS ↓ |
|---|---|---|---|---|
| `BlenderShapes` (RGB-D) | 3D-DLP-D (Ours) | $32.38 \pm 2.99$ | $0.9423 \pm 0.0036$ | $0.1490 \pm 0.069$ |
| | AE | $\mathbf{34.14 \pm 5.7}$ | $\mathbf{0.975 \pm 0.02}$ | $\mathbf{0.019 \pm 0.01}$ |
| | VAE | $16.28 \pm 1.94$ | $0.251 \pm 0.55$ | $0.45 \pm 0.06$ |
| `2DGenericShapes` (RGB-D) | 3D-DLP-D (Ours) | $\mathbf{39.16 \pm 6.23}$ | $\mathbf{0.989 \pm 0.012}$ | $\mathbf{0.035 \pm 0.023}$ |
| | AE | $34.14 \pm 5.8$ | $0.975 \pm 0.02$ | $0.159 \pm 0.05$ |
| | VAE | $16.28 \pm 1.94$ | $0.251 \pm 0.05$ | $0.451 \pm 0.06$ |
| `MimicGen` (RGB-D) | 3D-DLP-D (Ours) | $\mathbf{39.39 \pm 2.60}$ | $\mathbf{0.9891 \pm 0.0124}$ | $\mathbf{0.0089 \pm 0.009}$ |
| | AE | $28.97 \pm 1.36$ | $0.937 \pm 0.01$ | $0.172 \pm 0.02$ |
| | VAE | $22.13 \pm 1.31$ | $0.8396 \pm 0.035$ | $0.079 \pm 0.02$ |

*Table 10.* RGB-D reconstruction metrics (higher is better for PSNR/SSIM, lower is better for LPIPS).

| Dataset | Method | IoU ↑ |
|---|---|---|
| `GenericShapes` | AE | $0.229 \pm 0.04$ |
| | VAE | $0.090 \pm 0.01$ |
| | 3D-DLP-V (Ours) | $\mathbf{0.322 \pm 0.05}$ |
| `ShapeNetScenes` | AE | $0.117 \pm 0.05$ |
| | VAE | $0.070 \pm 0.00$ |
| | 3D-DLP-V (Ours) | $\mathbf{0.262 \pm 0.04}$ |
| `MimicGen` | AE | $\mathbf{0.631 \pm 0.03}$ |
| | VAE | $0.574 \pm 0.04$ |
| | 3D-DLP-V (Ours) | $0.569 \pm 0.04$ |

*Table 11.* **Occupancy voxel reconstruction.** We report IoU (higher is better). 3D-DLP-V consistently outperforms non-object-centric baselines (AE, VAE).

proprioceptive token is denoised jointly with the action and particle tokens, so the policy learns a consistent future trajectory over robot state, scene state, and control.

**Goal-token removal for per-task policies.** The original EC-Diffuser conditions action denoising on both the current observation and a goal-image particle set. Because we train one policy per task, a goal image provides no task-discriminative signal and, if retained, introduces a train/rollout distribution gap (goals are always available in demonstrations but absent at deployment). Rather than zeroing the goal stream—which we found unstable in practice—we remove it from the token sequence entirely. The resulting token layout at each timestep is $[\,\mathbf{a}_t,\ \mathbf{p}_t,\ \{\mathbf{z}_t^m\}_{m=1}^M\,]$, consisting of the action, proprioceptive, and particle tokens only.

**Language-token path (`RLBench`).** In RLBench, the task is specified by a natural-language instruction rather than a goal image, and we use the instruction in place of the removed goal stream. The 3D-DLP visual encoder remains frozen after representation learning and produces the scene state as a compact set of latent entity tokens.

To encode language, we use a frozen CLIP text encoder (ViT-B/32) and precompute text features for all instruction strings in the dataset. Each training episode stores a set of paraphrases for the same task, and at training time we sample one paraphrase per episode, providing a simple form of language augmentation while keeping the instruction constant across all timesteps in the trajectory. The resulting CLIP token embeddings are projected into the PINT token dimension with a small MLP and appended to the policy token sequence. The full transformer input therefore consists of action, proprioceptive, visual-latent, and language tokens in a shared token space.

Rather than introducing a separate cross-attention module, we treat language as additional context tokens that participate directly in the transformer's joint self-attention: language tokens are concatenated with the robot and visual tokens before the transformer layers, and standard attention masking is used to ignore padded text positions. The diffusion timestep embedding is added uniformly to all tokens, including language, so the model jointly reasons over task semantics, scene entities, and control variables throughout denoising. After transformer processing, the language tokens are discarded and only the action-related tokens are decoded into the predicted control sequence.

At inference, the instruction is encoded once at the beginning of an episode and cached across replanning steps. The policy then predicts a sequence of future actions conditioned on the current encoded observation and the language instruction,

executing only the first action before replanning, following the same receding-horizon control scheme as EC-Diffuser.

## F. Additional Imitation Learning Experiment Details

For our robotics experiments, we evaluate on 12 single-arm `MimicGen` tasks (Stack, Stack Three, Nut Assembly, Coffee, Pick Place, Coffee Preparation, Hammer Cleanup, Mug Cleanup, Kitchen, Three Piece Assembly, Threading, and Square) and 10 language-conditioned `RLBench` tasks from the PerACT subset. Policy training follows a two-stage pipeline. First, we train 3D-DLP representation models using 50 trajectories per task with an 80/20 train/eval split for 100 epochs. Second, we run the frozen 3D-DLP encoder over the 200 (`MimicGen`) / 100 (`RLBench`) demonstration trajectories used for imitation learning to extract latent particle tokens for each state–action pair, and train a per-task diffusion policy on the resulting latent trajectories using the adapted EC-Diffuser backbone.

**Why no EC-Diffuser-on-raw-voxels baseline.** EC-Diffuser's full self-attention is tractable for the compact particle set ($M \leq 40$) but prohibitively expensive on dense $64^3$ voxel grids, where the token count far exceeds $M$. Attention-based voxel policies (Shridhar et al., 2022) therefore rely on efficient attention variants such as Perceiver IO (Jaegle et al., 2022)—a contrast that directly motivates our compact particle representation and is why we omit raw-voxel EC-Diffuser as a baseline.

**Plan imagination with EC-Diffuser.** EC-Diffuser (Qi et al., 2025) jointly denoises actions and states. While we execute the denoised actions in the environment, we can simultaneously render the denoised 3D particles representing the predicted future states. In Figure 17, the rendered particle imagination strongly correlates with environment execution, demonstrating tight coupling between policy learning and 3D state prediction.

## G. Hyperparameters and Additional Training Details

| Attribute | Distribution | **Parameters** (glimpse_ratio $= 0.25$) | **Parameters** (glimpse_ratio $= 0.125$) |
|---|---|---|---|
| Position Offset $\Delta \bar{z}_p$ | Normal, $\mathcal{N}(\mu, \sigma^2)$ | $\mu = 0, \sigma = 0.2$ | $\mu = 0, \sigma = 0.1$ |
| Scale $z_s$ | Normal, $\mathcal{N}(\mu, \sigma^2)$ | $\mu = \text{Sigmoid}^{-1}(0.25), \sigma = 0.3$ | $\mu = \text{Sigmoid}^{-1}(0.125), \sigma = 0.15$ |
| Composite order $z_c$ | Normal, $\mathcal{N}(\mu, \sigma^2)$ | $\mu = 0, \sigma = 1$ | $\mu = 0, \sigma = 1$ |
| Transparency $z_t$ | Beta, $\text{Beta}(a, b)$ | $a = 0.01, b = 0.01$ | $a = 0.01, b = 0.01$ |
| Appearance Features $z_f, z_{\text{bg}}$ | Normal, $\mathcal{N}(\mu, \sigma^2)$ | $\mu = 0, \sigma = 1$ | $\mu = 0, \sigma = 1$ |

*Table 12.* Prior distribution parameters for different glimpse (patch) ratios. Glimpses are patches taken around keypoints, where glimpse_ratio $= \frac{\text{glimpse size}}{\text{image size}}$.

| Hyperparameter | MimicGen | RGB-D-SD-v2 | GenericShapes | ShapeNetScenes |
|---|---|---|---|---|
| Resolution | $64 \times 64 \times 64$ | $64 \times 64 \times 64$ | $64 \times 64 \times 64$ | $64 \times 64 \times 64$ |
| $M$ (# Particles) | 40 | 24 | 16 | 24 |
| $N_{\text{patch}}$ (# KP Proposals) | 64 | 64 | 64 | 64 |
| $\beta_{\text{KL}}$ | 0.02 | 0.2 | 0.2 | 0.02 |
| $\beta_f$ | 0.005 | 0.005 | 0.005 | 0.005 |
| $\beta_{\text{obj}}$ | 0.02 | 0.02 | 0.02 | 0.02 |
| K-means Iter. | 5 | 5 | 5 | 5 |
| Glimpse Ratio | 0.25 | 0.125 | 0.25 | 0.25 |
| $d_{\text{obj}}$ | 4 | 4 | 4 | 5 |
| $d_{\text{bg}}$ | 4 | 4 | 4 | 5 |
| FG CNN Ch. Mult. | $[2, 2, 4]$ | $[1, 2, 4]$ | $[1, 2, 4]$ | $[1, 2, 4]$ |
| BG CNN Ch. Mult. | $[1, 1, 1, 2, 4]$ | $[1, 1, 1, 2, 4]$ | $[1, 1, 1, 2, 4]$ | $[1, 1, 1, 2, 8]$ |
| # Epochs | 16 | 15 | 18 | 200 |

*Table 13.* Hyperparameters across datasets for 3D-DLP-V and 3D-DLP-VC. Base CNN channels count is 32.

| Hyperparameter | MimicGen | RGB-D-SD-v2 | GenericShapes |
|---|---|---|---|
| Resolution | $64 \times 64$ | $64 \times 64$ | $64 \times 64$ |
| $M$ (# Particles) | 16 | 64 | 30 |
| $N_{\text{patch}}$ (# KP Proposals) | 64 | 256 | 64 |
| $\beta_{\text{KL}}$ | 0.02 | 0.02 | 0.08 |
| $\beta_f$ | 0.005 | 0.005 | 0.005 |
| $\beta_{\text{obj}}$ | 0.01 | 0.02 | 0.08 |
| KP Proposal Patch Size | 8 | 8 | 8 |
| Glimpse Ratio | 0.25 | 0.125 | 0.25 |
| $d_{\text{obj}}$ | 4 | 4 | 4 |
| $d_{\text{bg}}$ | 4 | 4 | 4 |
| FG CNN Ch. Mult. | $[1, 2, 4]$ | $[2, 4, 8]$ | $[1, 4, 8]$ |
| BG CNN Ch. Mult. | $[1, 1, 1, 2, 4]$ | $[1, 1, 1, 2, 4]$ | |
| # Epochs | 20 | 15 | 18 |

*Table 14.* Hyperparameters across datasets for 3D-DLP-D. Base CNN channels count is 32.

| Hyperparameter | Value |
|---|---|
| # Particles ($M$) | 40 |
| Particle Feature Dim | 12 |
| Planning Horizon ($H$) | 16 |
| Execution Steps | 8 |
| Diffusion Steps ($T$) | 5 |
| Transformer Layers / Heads | 6 / 8 |
| Batch Size / Learning Rate | $512 / 1 \times 10^{-4}$ |

*Table 15.* Hyperparameters for training EC-Diffuser on 3D-DLP-VC. Transformer layers and heads apply to both encoder and decoder.

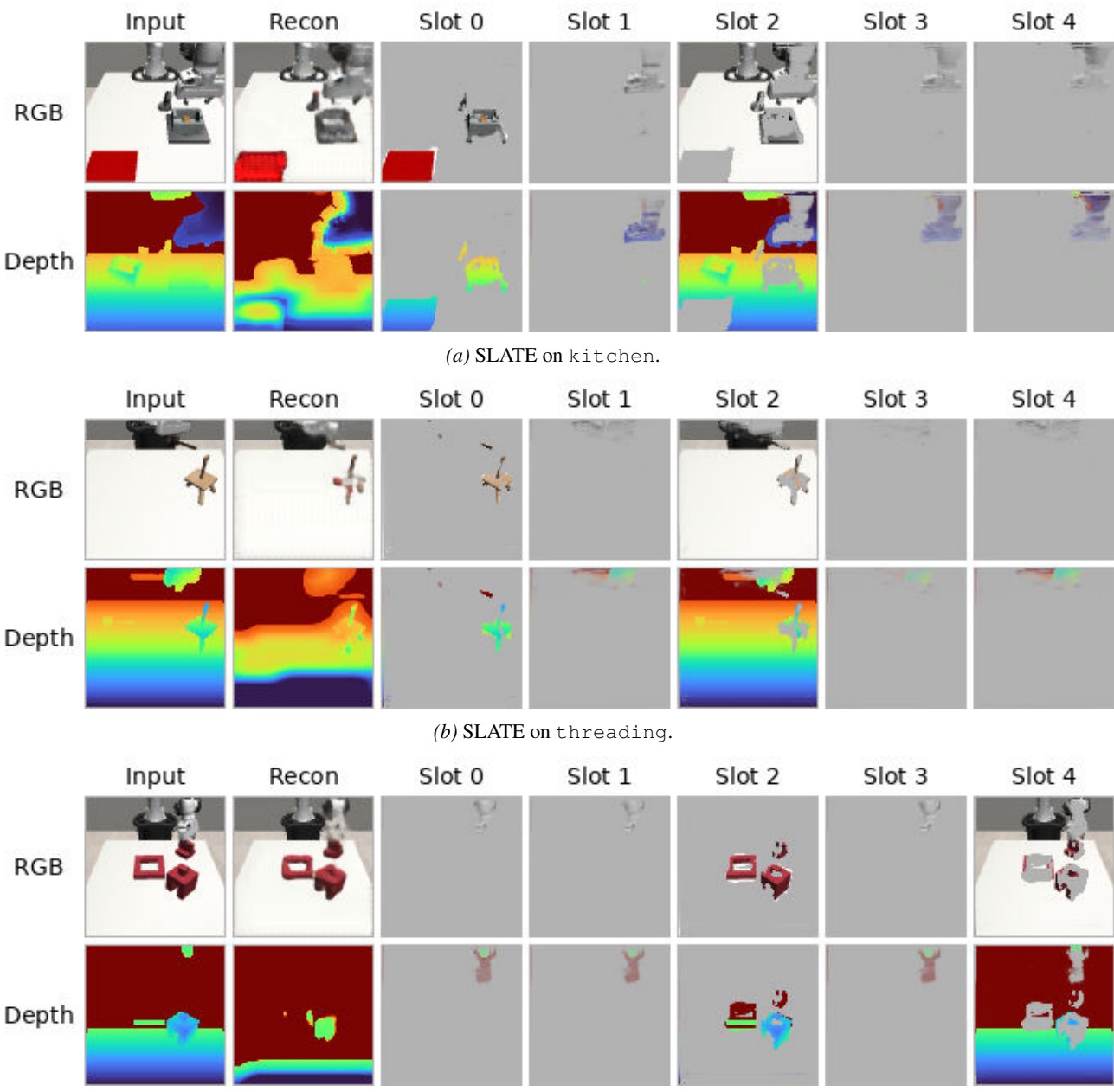

*(a)* SLATE on `kitchen`.

*(b)* SLATE on `threading`.

*(c)* SLATE on `three_piece_assembly`.

*Figure 12.* **SLATE decompositions on `MimicGen` RGB-D.** Same layout as Figure 11: input RGB-D, reconstruction, and per-slot reconstructions. SLATE exhibits the same under-utilization—most slots are nearly blank, and object/background separation is poor— qualitatively corroborating the metric gap in Table 9 and motivating our particle-based design, which anchors latents at local keypoints rather than competing over a fixed global slot budget.

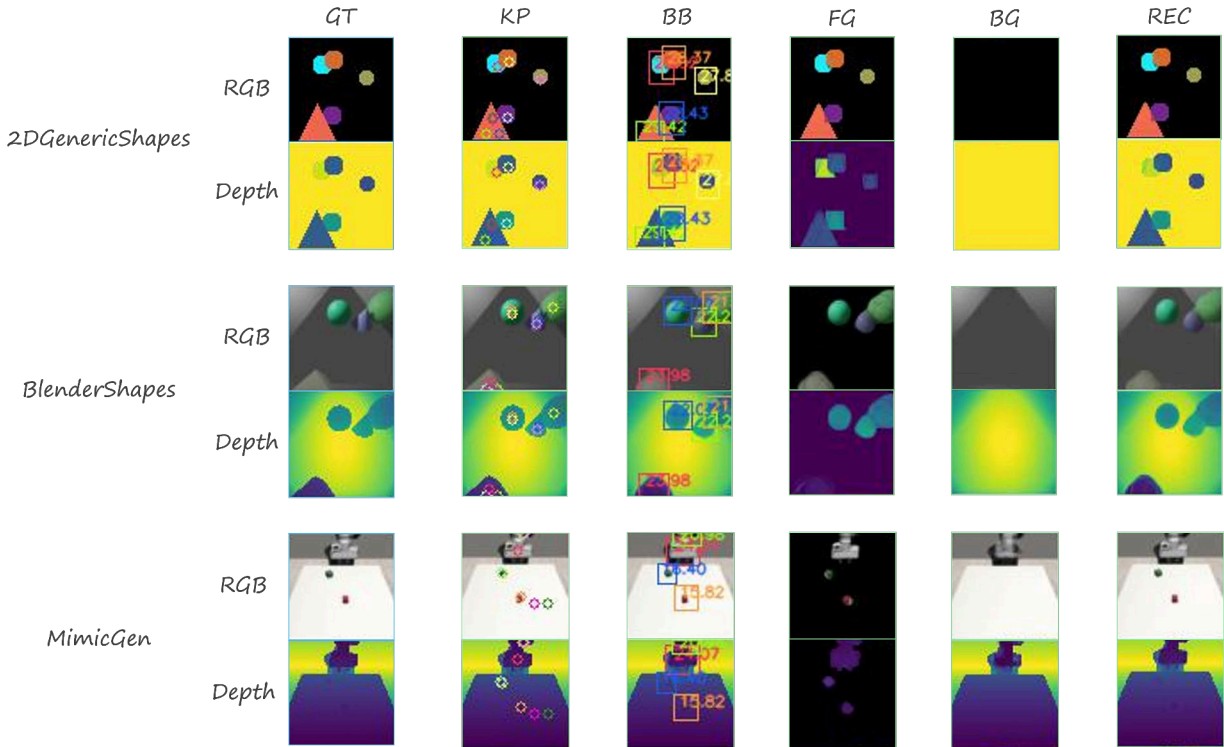

*Figure 13.* **3D-DLP-D Scene Decomposition.** 3D-DLP-D extends 2D DLP by learning latent particles jointly from RGB and depth images. It models appearance features separately for color and depth channels, while explicit attributes such as keypoints and bounding boxes are learned jointly across all channels.

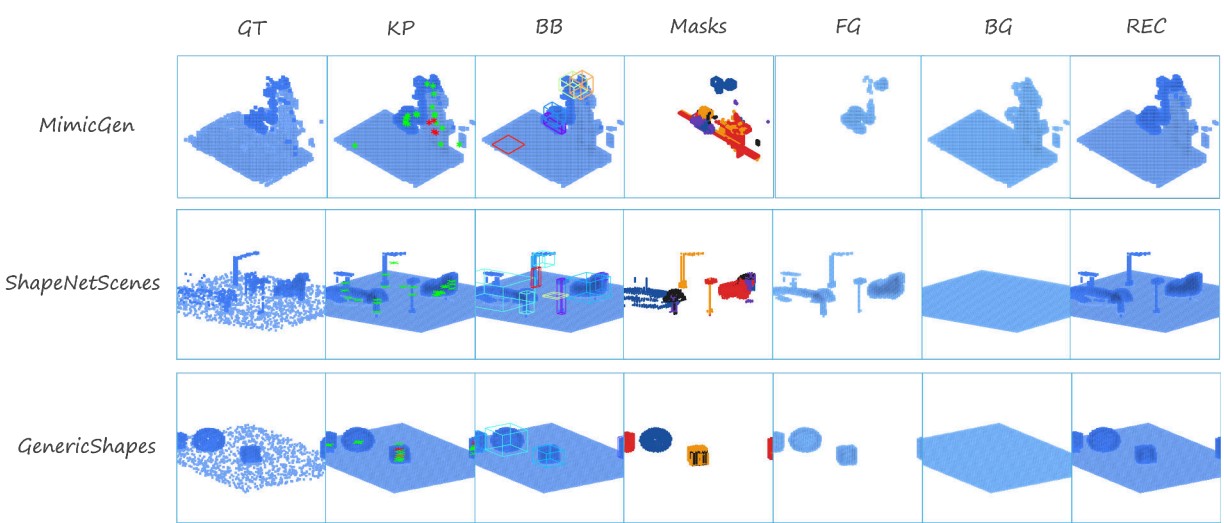

*Figure 14.* **3D-DLP-V Scene Decomposition.** From input occupancy voxels, 3D-DLP-V infers latent particles with explicit attributes (keypoints, scales) and produces object/background masks entirely without supervision, compositing the input scene.

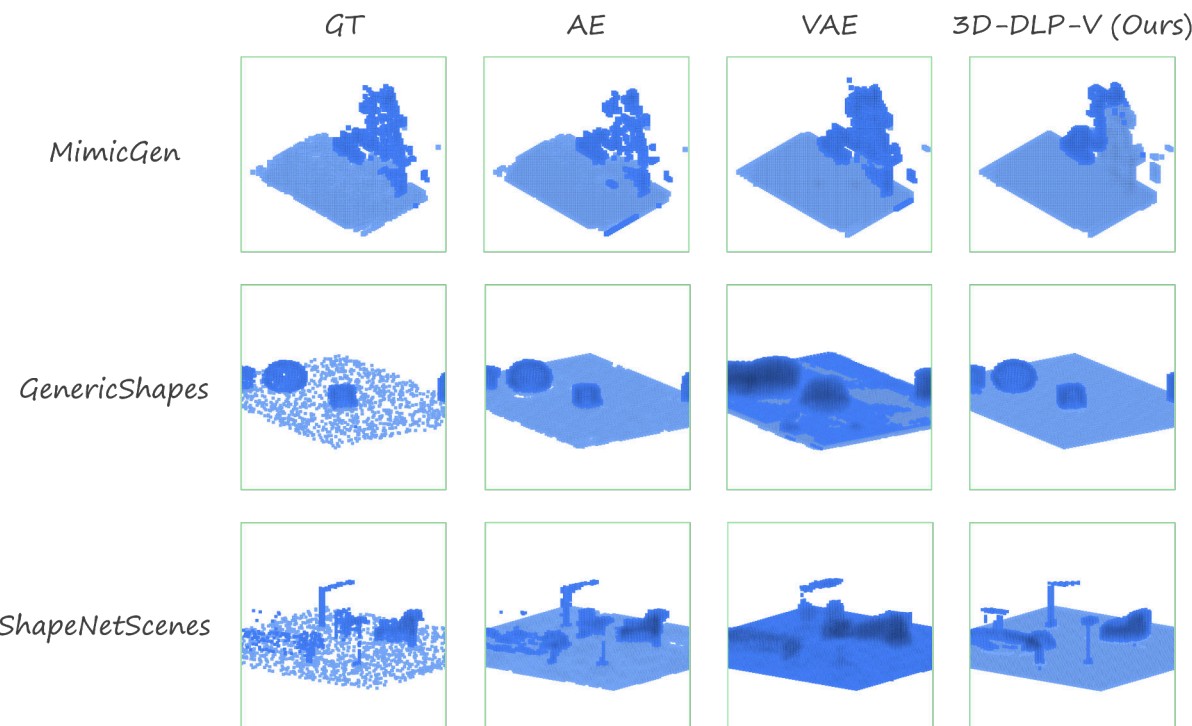

*Figure 15.* **Occupancy Voxels Scene Reconstruction Comparison.** Reconstruction of input occupancy voxels from various datasets. AE-Autoencoder, VAE-Variational autoencoder.

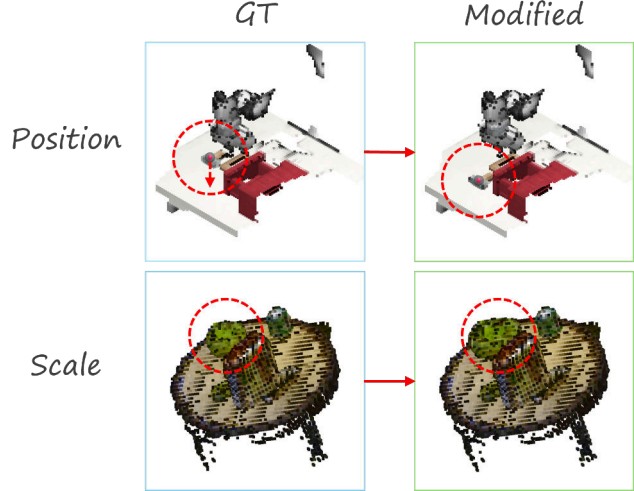

*Figure 16.* **Latent space controllability.** Modifying individual particle attributes–3D position (top row) and scale (bottom row)–directly translates to intuitive scene changes: object translation and resizing.

*T*

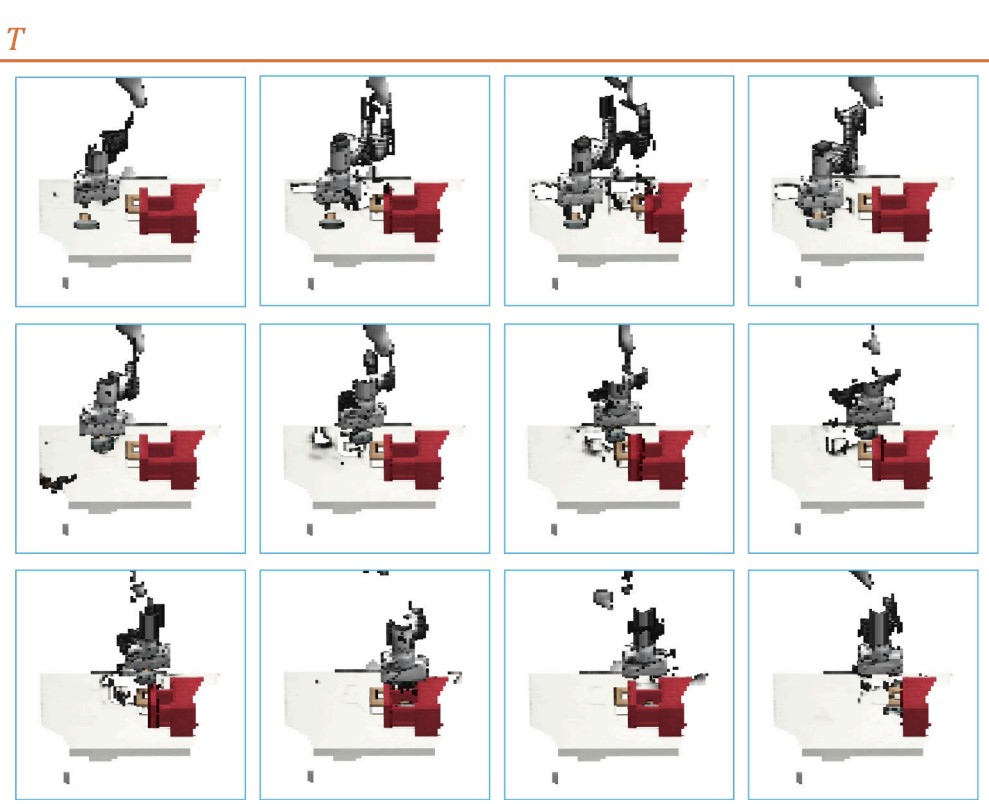

*Figure 17.* **Plan imagination with EC-Diffuser.** EC-Diffuser denoises states and actions together. We visualize the imagined plan over time (left-to-right) by rendering the denoised 3D latent particles, representing the state in EC-Diffuser. The visualized plan closely matches real outcomes.

