# OpenReview forum: "3D-DLP: Self-supervised 3D Object-centric Scene Representation Learning"
_ICML.cc/2026/Conference — ICML 2026 regular_

### Official Review · Reviewer_3qyn · 2026-02-24

**Soundness:** 3
**Presentation:** 3
**Significance:** 2
**Originality:** 2
**Overall Recommendation:** 4
**Confidence:** 4

**Summary:**

This paper introduces a 3D adaptation of the Deep Latent Particles (DLP) framework which yields an interpretable and controllable latent space. The authors claim that this enables better 3D understanding, and thus better performance in manipulation tasks.

[1]  “Unsupervised Image Representation Learning with Deep Latent Particles”, ICML 2022.

**Compliance With Llm Reviewing Policy:**

Affirmed.

**Final Justification:**

The paper shows improvements in manipulation performance with the proposed 3D-DLP representations. During the rebuttal, the authors have addressed my biggest concern [Q2] and clarified a misunderstanding regarding EC-diffuser [Q4]. I still find the contribution somewhat limited due to [Q1,3], but consider it sufficient for acceptance and increase my score accordingly.

**Key Questions For Authors:**

My main concerns are summarized in the **Major Weaknesses** section. If the authors either address the following questions regarding these weaknesses **[Q1~4]** or provide a convincing explanation regarding these issues, I am prepared to raise the score accordingly.

- **[Q1] Comparison against RVT.** Could you provide experiment results on RLBench [1] to show direct comparisons against PerAct and RVT, both of which use 3D representations to improve manipulation performance, and are thus closely related works?
- **[Q2] Incomplete experiment results in Table 4.** EquiDIff [2], the strongest baseline in Table 4., presents results on all 12 tasks in MimicGen. Could you provide results for 3D-DLP-VC in all tasks as well for a more fair and comprehensive comparison against EquiDiff? If the authors believe that the difference in observation is the reason for performance difference, they should either test following EquiDiff’s observation (Voxel + In-Hand), or provide results for EquiDiff with only voxel observations (*i.e.*, removing in-hand observations). Details on architecture modifications should be included to handle these differences in inputs.
- **[Q3] Latent controllability.** Could you provide more complex examples? Showing examples such as the following would suffice to claim latent controllability on the learned 3D representation.
(1) What happens when the selected particle is translated towards another particle or occupied background voxel? Taking Figure 4 as an example, if the hammer is translated towards the robot arm, does modified reconstruction show a superposition of voxels corresponding to the hammer and the robot arm, or does it show the hammer rotating (yaw-wise) and yielding a plausible scene?
(2) Assuming we can train the representation such that the robot arm’s end-effector is a particle, does translating the end-effector to random locations yield plausible joint configurations for the robot arm?
- **[Q4] Ablating the proposed 3D representation.** The most straightforward way to show that the proposed 3D adaptation of DLP improves 3D understanding and thus, manipulation performance, is by simply ablating the use of such 3D representation. Could you provide results of EC-diffuser against EC-diffuser + 3D-DLP-VC on manipulation tasks?

[1] “RLBench: The Robot Learning Benchmark & Learning Environment”, RA-L 2020.\
[2] “Equivariant Diffusion Policy”, CoRL 2024.\
[3] “EC-Diffuser: Multi-Object Manipulation via Entity-Centric Behavior Generation”, ICLR 2025.

**Limitations:**

yes

**Strengths And Weaknesses:**

**Strengths:**

- The authors focus on the right ideas that are vital for better manipulation performance: better 3D understanding, and a scalable (self-supervised) way to learn such representations.
- The paper is easy to follow and well-explained. The authors provide a very detailed Appendix with thorough explanations on datasets, model architecture, and experiment setups.

**Minor suggestions for presentation:**

- Line 134 should start with something like “Disadvantages of point cloud inputs” instead of “From pixels to voxels”.
- The Encoder paragraph (lines 143 ~ 153, second column) is referenced to Appendix A.2.1 (Prior) instead of A.2.2 (Encoder).
- Figure 1 could be more detailed (similar to the Figure 8 in the Appendix). Visualization or simply adding notations of each of the following components could help the readers follow through with the explanations: Input (colored voxels), Prior, Encoder, Particle attributes, Decoder, Decoded components (bounding boxes, masks, FG, BG, full reconstruction), and loss functions.

**Major Weaknesses:**

- Main experiment results are focused on the wrong directions. The authors provide three results (Tables 1~3) focused on reconstruction quality of the input (occupancy voxel and RGB voxel). The main narrative of the paper, as stated in the introduction, is that “3D representations are increasingly vital for robotic decision-making”. Therefore, results on downstream manipulation tasks should be the main focus instead of reconstruction quality (only Table 4 provide such results). On top of that, if the main focus was on scene reconstruction, then the authors should have provided stronger baselines instead of simple AE and VAE architectures. If the focus was on unsupervised scene decomposition, the authors should have provided comparisons against SPAIR3D [1], as stated in their related works.
- Lack of experiments with direct comparisons to the closest related works. Since the focus, to my understanding, is on robotic manipulation tasks using 3D representations, there should be direct comparisons against PerAct [2] and RVT [3, 4].
- The claims behind latent controllability are too far-fetched and lack experimental support. The authors provide way too simplistic examples: moving the position of a particle and scaling it. Both of these examples can be simply done by translating and scaling the voxels corresponding to the selected particle. In order to show more convincing results for latent controllability, the authors should provide more complex examples that are not straightforward (*e.g.*, Figure 2 in the original DLP paper [5] shows how the facial structure is modified according to translated particles, yielding realistic faces even when there is collision between two particles, such as when they close the bottom lips: “Move Mouth KP”). Please refer to **[Q3]** for suggested examples that show latent controllability.

**Minor Weaknesses:**

- The proposed use of RGB voxels could introduce some problems. The authors define empty voxels as zero-valued voxels. This could pose some problems in distinguishing empty voxels from black colored objects.
- The proposed framework is quite limited in novelty, as it is a straightforward adaptation of DLP.
- It is not clear how the use of voxels solve the occlusion problem, since the voxelization process simply uses point cloud inputs.

[1] “Spatially Invariant Unsupervised 3D Object-Centric Learning and Scene Decomposition”, ECCV 2022.\
[2] “Perceiver-Actor: A Multi-Task Transformer for Robotic Manipulation”, CoRL 2022.\
[3] “RVT: Robotic View Transformer for 3D Object Manipulation”, CoRL 2023.\
[4] “RVT-2: Learning Precise Manipulation from Few Examples”, RSS 2024.\
[5]  “Unsupervised Image Representation Learning with Deep Latent Particles”, ICML 2022. https://taldatech.github.io/deep-latent-particles-web/. \
[6] “DDLP: Unsupervised Object-Centric Video Prediction with Deep Dynamic Latent Particles”, TMLR 2024.

---

> ### Author Rebuttal · Authors · 2026-03-31
>
> We thank the reviewer for the detailed and thoughtful feedback. We address your concerns below.
>
> At a high level, we propose a self-supervised 3D object-centric method that decomposes voxel scenes into latent particles. To the best of our knowledge, this is the first self-supervised object-centric approach operating directly on voxel grids rather than point clouds, neural fields, or 2D images. The goal of the paper is twofold: (1) to introduce this 3D extension of DLP and its associated design choices, and (2) to show that the resulting representation is interpretable and useful for downstream decision-making. Because it is a vision-based representation, we evaluate it first through scene decomposition and reconstruction, comparing against matched non-object-centric voxel baselines. We do not aim to compete with large-scale 3D reconstruction methods; instead, reconstruction serves as a learning signal and a way to probe latent controllability.
> Our claims are correspondingly narrow: 3D-DLP learns object-centric, controllable 3D latents that match or outperform non-object-centric baselines under the same voxel observation regime. Rather than claiming state-of-the-art benchmark performance, we aim to show that this representation is interpretable and promising for downstream decision-making.
>
>
> **Presentation**: Thank you for pointing out the suggested presentation improvements, we will modify accordingly in the revision.
>
>
> **Comparison with SPAIR3D**: SPAIR3D [16] is a self-supervised object-centric method for colorless point clouds, not voxel grids, so it is not a like-for-like baseline. Our method is also particle-based [15, 2], with local latent variables modeling small spatial regions, whereas SPAIR3D is patch-based [17, 18]. These correspond to different inductive biases and training regimes, and extending SPAIR3D to voxel observations would require non-trivial architectural changes.
>
> **EC-Diffuser / downstream imitation learning**
> We provide additional imitation-learning results on the full MimicGen suite; the full table is included in our response to [Reviewer Kdez](https://openreview.net/forum?id=vIotI25gJz&noteId=snJordAtMH). Overall, 3D-DLP + EC-Diffuser achieves the most task wins (6/12) and the highest mean success rate (48.1) , supporting the usefulness of the proposed 3D object-centric representation for downstream policy learning. EC-Diffuser with raw voxels is not included because full attention is tractable for compact particle latents but prohibitively expensive for dense voxel grids.
>
> **Benchmark and baselines**: As noted above, our goal is to evaluate the usefulness of 3D-DLP representations, not to establish state-of-the-art results across all manipulation benchmarks. We therefore build on Equidiff [14], which already uses voxel observations. PerAct [19] and RVT [20] are highly relevant, but they are language-conditioned methods evaluated primarily on RLBench; adapting our representation to that setting is beyond the scope of this work.
>
>
> **Latent controllability**: Our 3D extension is based on DLPv2/v3-style architectures [2,6], which omit the GNN decoder used in the original DLP [15]. As a result, manipulating particles in latent space does not model physical collisions or joint constraints. Our claim is therefore modest: individual particles correspond to coherent 3D components whose position, scale, and transparency can be modified in latent space to yield consistent 3D reconstructions. This shows interpretability and controllability of the representation, not full physical interaction modeling.
>
>
> **Novelty**: We are glad that the method appears simple and easy to understand; however, the 3D extension required several non‑trivial design changes compared to 2D DLP (e.g., K-means prior, chroma loss). To the best of our knowledge, **there are no prior self‑supervised object‑centric methods that operate directly on voxels**, so we believe our method fills an important gap between 2D object‑centric approaches and 3D representations used in robotics and vision.
>
>
> **RGB Voxels**: In our RGB‑only voxel formulation, the occupancy mask used for the chroma loss was induced via an RGB‑magnitude threshold, which could indeed miss very dark voxels. This effect was partially mitigated by the MSE loss applied over all voxels, which still penalized reconstruction errors at those locations, but we agree this is a limitation that may be alleviated by, e.g., jointly modeling occupancy and RGB.
>
>
> **Occlusion**: Our intention was not to claim that voxels “solve” occlusion, but rather that a structured 3D grid provides a more uniform representation for learning from partial observations, which may help policies and representations become more robust when combined with appropriate priors or multi‑view data. We will rephrase this part of the paper to avoid overstating the role of voxelization with respect to occlusion.
>
> * Note: referenced citations are in the response for Reviewer PpcE

---

> > ### Author Rebuttal · Reviewer_3qyn · 2026-04-04
> >
> > As the authors have mentioned, the main goal of the paper is to introduce a 3D representation that shows (1) latent controllability and interpretability, and (2) improves downstream manipulation tasks. Therefore, comparing against other object-centric representations or 3D scene representations unrelated to (2) is not within the scope of this work (e.g., OSRT, 3DGS, NeRF, etc.). Precisely because of this, I found it odd that in the initial manuscript most results were unrelated to (2). I will provide my response to the rebuttal based on this, and the key questions I have provided.
> >
> > Note: The provided reference for [14] is wrong. It should be “Equivariant Diffusion Policy”.
> >
> > **[Q1] Comparison against RVT (Unresolved).** While I agree that adapting 3D-DLP into a language-conditioned method is not trivial or of interest for this work, providing results on RLBench could provide a better understanding on the proposed representations’ capabilities. On top of that, PerAct provides results without language conditioning on their paper. Could you provide results on RLBench without language conditioning?
> >
> > **[Q2] Incomplete experiment results in Table 4 (Resolved).** The authors have provided the results, showing higher average success rates than EquiDiff with only voxel observations (48.1% vs. 47.3%).
> >
> > **[Q3] Latent controllability (Unresolved).** The authors show examples of latent controllability by modifying the position and scale of the particles (Figure 4). In the given examples, the modifications in the latent space seem to have a direct 1-to-1 correspondence with the output 3D space. This means that, there exists a finite set of modifications that yield physically infeasible (or impossible) configurations of the scene (see original review’s **[Q3]** for examples). From the RL literature I’m familiar with, latent controllability is typically associated with learning physically feasible configurations [1~4]. As a result, it is unclear whether 3D-DLP shows latent controllability, but rather, a direct parameterization of objects in the scene.
> >
> > **[Q4] Ablating the proposed 3D respresentation (Unresolved).** The authors stated that directly using EC-Diffuser with raw voxels is computationally infeasible, and thus were unable to provide comparisons. My original question, however, was a comparison of EC-Diffuser *with multi-view image observations* against EC-Diffuser + 3D-DLP-VC. Additionally, the authors could reinforce the effectiveness of 3D-DLP representations by providing results of EC-Diffuser with raw voxels by either using a larger 3D patch size or downsampling before attention operations.
> >
> > [1] “Dynamics-aware unsuperivsed discovery of skills”, ICLR 2020.\
> > [2] “Controllability-aware unsupervised skill discovery”, ICML 2023.\
> > [3] “METRA: Scalable Unsupervised RL with Metric-Aware Abstraction”, ICLR 2024.\
> > [4] “Foundation Policies with Hilbert Representations”, ICML 2024.

---

> > > ### Author Response · Authors · 2026-04-05
> > >
> > > Thank you; we sincerely appreciate the effort you put into reviewing our work, and we believe your suggestions in the original review contributed to improving the paper.
> > >
> > > **[Q1] More benchmarks (RLBench).**
> > > We thank the reviewer for the suggestion. We agree that adding more benchmarks would broaden the impact of the paper, but we also believe that the current set of experiments provides sound evidence for our core claims: that 3D‑DLP learns an interpretable, object‑centric 3D representation that is useful for downstream manipulation in the voxel‑based setting of MimicGen. Our evaluation spans multiple tasks, multiple baselines, and both 2D‑ and 3D‑DLP latent spaces, which is consistent in scale with similar representation‑learning works that focus on a specific environment and a set of tasks within it.
> > >
> > > That said, while extending 3D‑DLP to language‑conditioned RLBench is non‑trivial and would require additional engineering and training, we will make an effort to provide results on RLBench in the final version, if time and compute permit. This will help further contextualize the capabilities of the representation beyond the current benchmark.
> > >
> > > **[Q3] Latent controllability.**
> > > The reviewer writes:
> > > > “it is unclear whether 3D‑DLP shows latent controllability, but rather, a direct parameterization of objects in the scene.”
> > >
> > > We fully agree with this characterization. Our latent‑modification experiments (e.g., moving and scaling particles in Figure 4) are intended as a **qualitative visualization tool** to assess the object‑centric nature of the representation, not as a demonstration of RL‑style controllability or skill discovery. In this sense, the latent particles are indeed a reparameterization of objects in the scene, and the reconstruction changes induced by modifying these parameters are deterministic and geometric, not dynamical or physically constrained.
> > >
> > > We do not claim that our particles correspond to “skills” or that they encode physically feasible trajectories in the same sense as the RL works cited. Our controllability claim is limited to the observation that individual particles correspond to coherent 3D components whose position and scale can be manipulated in latent space, yielding consistent 3D reconstructions. This is a weaker form of controllability than the RL‑oriented notion, but it is sufficient for our goal of demonstrating interpretability and object‑centric decomposition.
> > >
> > > **[Q4] Ablating the proposed 3D representation (EC‑Diffuser + multiview vs. 3D‑DLP).**
> > > We kindly note that we **did perform the experiment requested in your original question**: EC‑Diffuser with multi‑view 2D‑DLP versus EC‑Diffuser with 3D‑DLP‑VC, isolating the effect of 2D versus 3D particle‑based representations under the same policy architecture. We reported these results in the response to [Reviewer Kdez](https://openreview.net/forum?id=vIotI25gJz&noteId=qBx4UiUmJ8), and reproduce them here for clarity:
> > > Table 3: We compare 3D-DLP + EC-Diffuser with multi-view 2D-DLP + EC-Diffuser, isolating the effect of 2D versus 3D particle-based representations under the same policy architecture, and with a voxel-only EquiDiff [14] baseline that removes the eye-in-hand RGB input.
> > > | Method | Stack | Stack Three | Nut Assembly | Coffee | Pick Place | Coffee Preparation | Hammer Cleanup | Mug Cleanup | Kitchen | Three Piece Assembly | Threading | Square |
> > > |---|---:|---:|---:|---:|---:|---:|---:|---:|---:|---:|---:|---:|
> > > | 3D-DLP | **94.6 ± 0.9** | **70.0 ± 1.6** | 6.0 ± 1.6 | 36.0 ± 1.6 | 0.0 ± 0.0 | 0.0 ± 0.0 | **94.6 ± 0.9** | **64.0 ± 4.3** | 86.7 ± 3.4 | **38.0 ± 4.0** | 36.0 ± 1.6 | **51.3 ± 0.9** |
> > > | 2D-DLP multiview | 78.0 ± 2.8 | 14.7 ± 6.2 | 8.7 ± 2.5 | **82.0 ± 2.8** | 4.7 ± 2.5 | 0.0 ± 0.0 | 66.7 ± 2.5 | 34.7 ± 7.5 | 0.0 ± 0.0 | 29.3 ± 7.7 | **45.3 ± 6.8** | 45.3 ± 10.5 |
> > > | EquiDiff Voxel Only | 82.0 ± 0.0 | 12.7 ± 5.7 | **10.7 ± 2.1** | 70.7 ± 3.4 | **12.7 ± 2.2** | **34.0 ± 4.9** | 92.6 ± 3.3 | 34.0 ± 2.8 | **95.0 ± 2.5** | 31.3 ± 3.7 | 42.0 ± 0.0 | 50.0 ± 2.8 |
> > >
> > > 3D‑DLP‑VC + EC‑Diffuser achieves the most task wins (6 of 12), and also the highest mean success rate across tasks (48.1% vs. 47.3% for 2D‑DLP multiview and 34.1% for EquiDiff voxel‑only). The results suggest that explicit object‑centric 3D representations are often beneficial for downstream policy learning, though the benchmark also reveals important failure cases (e.g., Coffee Preparation, where the coffee cup is not cleanly isolated in the learned decomposition, likely limiting downstream control).
> > >
> > > Regarding EC‑Diffuser with raw voxels, the key limitation is computational: full attention over particle latents is tractable, but it becomes prohibitively expensive for dense voxel grids, where the number of voxel tokens far exceeds the number of latent particles. One can indeed reduce the token count via larger voxel patches, but then the representation loses its 3D locality, which is a key motivation for introducing 3D‑DLP in the first place.

---

### Official Review · Reviewer_Kdez · 2026-03-12

**Soundness:** 2
**Presentation:** 3
**Significance:** 3
**Originality:** 2
**Overall Recommendation:** 3
**Confidence:** 4

**Summary:**

This paper introduces 3D-DLP, a self-supervised, object-centric models that extend Deep Latent Particles (DLP) from 2D images to 3D observations. The proposed approach represents scenes as a collection of 3D latent "particles," each with disentangled attributes, such as position, scale, transparency, and appearance, alongside a background latent. 3D-DLP demonstrates good reconstruction quality and qualitative object discovery. It can also be integrated into downstream multi-object robotic manipulation tasks and show promising results.

**Compliance With Llm Reviewing Policy:**

Affirmed.

**Final Justification:**

The paper is still weak in evaluation baselines and metrics and I would keep my rating, but i won't be upset if it is accepted.

**Key Questions For Authors:**

1. How sensitive is performance to the number of particles M, and how do you choose M per dataset? Is there slot collapse or persistent over-segmentation?

2. When using K-means in the joint appearance-geometry space, will objects with less distinctive chromatic appearance or at darker places be ignored?

3. Can you provide AP/ARI to quantitatively evaluate object discovery ability?

4. To prove the effectiveness of 3D-DLP in imitation learning, can you add two more baselines? (a) EC-Diffuser with voxels (b) EC-Diffuser with 2D-DLP.

**Limitations:**

Yes.

**Strengths And Weaknesses:**

Strengths:

1. The paper is well-written with a clear structure and helpful visualization.

2. It extends object-centric representations to 3D scenes in a self-supervised manner, which is a promising direction for efficient scene understanding and manipulation.

3. This paper also proves the usefulness of the self-supervised, object-centric scene representation in downstream simulated multi-object robotic manipulation tasks.

Weaknesses:

1. Missing the latest related works. The literature mainly discusss works before 2024, missing recent unsupervised/weakly-supervised 3D object-centric frameworks such as [1, 2, 3]. Although these may use different inputs, they are highly relevant to the quality and context of object-centric 3D decomposition and reconstruction. Importantly, a very recent work [4] that also models 3D scenes with latent particles and also applies it to robotic manipulation tasks, but there is no disucssion at all.

2. Baselines are weak. Reconstruction baselines are limited to AE and VAE; there is no comparison to established 3D reconstruction/occupancy methods (e.g., Occupancy Networks, NeRF/voxel autoencoding baselines, or recent self-supervised occupancy from multi-view), which weakens reconstruction claims.

3. Object discovery is not quantitatively evaluated. Object-centric discovery is supported mainly with visuals; standard object-centric metrics (e.g., ARI, AP) are not reported.

4. Insufficient experiments for imitation learning. There is no ablation comparing the same EC-Diffuser policy with raw voxels or with 2D-DLP to isolate the specific contribution of 3D-DLP latents.

5. Comparison is not strictly controlled. From Table 4, since different methods use different observation modalities, it is unclear whether the performance superiority is due to voxel input or the algorithm itself.

[1] Unsupervised Discovery of Object-Centric Neural Fields (TMLR 2025)

[2] Dynamic Scene Understanding through Object-Centric Voxelization and Neural Rendering (TPAMI2025)

[3] GrabS: Generative Embodied Agent for 3D Object Segmentation without Scene Supervision (ICLR 2025)

[4] Latent Particle World Models: Self-supervised Object-centric Stochastic Dynamics Modeling (ICLR 2026)

---

> ### Author Rebuttal · Authors · 2026-03-31
>
> We thank the reviewer for the thoughtful feedback and address the main concerns below.
>
>
> **Related work**: We thank the reviewer for pointing out [11,12,13], which we will add in revision. These works are relevant but not like-for-like baselines: [11,12] are image/neural-rendering based, [13] is point-cloud segmentation, and [6] is post-deadline and 2D RGB, whereas our setting is self-supervised object-centric learning directly from voxel observations.
>
>
> **Baselines**: Our goal is to introduce a 3D object-centric representation—a latent particle decomposition of voxel scenes—and show that it is useful for downstream decision-making. We therefore compare against matched non-object-centric voxel baselines (AE/VAE) under the same objective. We do not aim to compete with large-scale 3D reconstruction methods; our claim is simply that the object-centric inductive bias improves downstream usefulness while maintaining comparable or better reconstruction than non-object-centric baselines.
>
>
> **Metrics**: We agree that ARI/AP would provide additional evidence, but they are not straightforward here because ground-truth instance masks are unavailable. SAM- or SAM-3D-style masks [10] could be used, but they are not fully automatic and may not align with decompositions learned by a self-supervised model. We therefore evaluate primarily through qualitative decompositions and downstream imitation-learning performance.
>
> **EC-Diffuser / downstream imitation learning**
> We provide additional imitation-learning results on the full MimicGen suite.
>
> Table 3: We compare 3D-DLP + EC-Diffuser with multi-view 2D-DLP + EC-Diffuser, isolating the effect of 2D versus 3D particle-based representations under the same policy architecture, and with a voxel-only EquiDiff [14] baseline that removes the eye-in-hand RGB input.
>
> | Method | Stack | Stack Three | Nut Assembly | Coffee | Pick Place | Coffee Preparation | Hammer Cleanup | Mug Cleanup | Kitchen | Three Piece Assembly | Threading | Square |
> |---|---:|---:|---:|---:|---:|---:|---:|---:|---:|---:|---:|---:|
> | 3D-DLP | **94.6 ± 0.9** | **70.0 ± 1.6** | 6.0 ± 1.6 | 36.0 ± 1.6 | 0.0 ± 0.0 | 0.0 ± 0.0 | **94.6 ± 0.9** | **64.0 ± 4.3** | 86.7 ± 3.4 | **38.0 ± 4.0** | 36.0 ± 1.6 | **51.3 ± 0.9** |
> | 2D-DLP multiview | 78.0 ± 2.8 | 14.7 ± 6.2 | 8.7 ± 2.5 | **82.0 ± 2.8** | 4.7 ± 2.5 | 0.0 ± 0.0 | 66.7 ± 2.5 | 34.7 ± 7.5 | 0.0 ± 0.0 | 29.3 ± 7.7 | **45.3 ± 6.8** | 45.3 ± 10.5 |
> | EquiDiff Voxel Only | 82.0 ± 0.0 | 12.7 ± 5.7 | **10.7 ± 2.1** | 70.7 ± 3.4 | **12.7 ± 2.2** | **34.0 ± 4.9** | 92.6 ± 3.3 | 34.0 ± 2.8 | **95.0 ± 2.5** | 31.3 ± 3.7 | 42.0 ± 0.0 | 50.0 ± 2.8 |
>
> 3D-DLP + EC-Diffuser achieves the most task wins, performing best on 6 of 12 tasks, and also obtains the highest mean success rate across tasks (48.1 vs. 47.3 and 34.1). It performs especially well on some tasks suggesting that an explicit object-centric 3D representation is often beneficial for downstream policy learning. At the same time, the benchmark reveals important failure cases: for example, both 3D-DLP and 2D-DLP perform poorly on Coffee Preparation, where the coffee cup is not cleanly isolated in the learned decomposition, likely limiting downstream control. Regarding EC-Diffuser with raw voxels, the key limitation is computational. Full attention is feasible for compact particle latents but prohibitively expensive for dense voxel grids, where the number of voxel tokens is far larger than the number of latent particles.
>
> **K-means**: Our method does not cluster on color alone: each occupied voxel is represented by CIELAB color concatenated with normalized 3D position. Weak chromatic distinctiveness alone therefore does not prevent a region from being proposed. That said, the concern is partly valid for very dark regions, since the proposal stage uses a lightness-weighted K-means variant that biases anchors toward visually informative surface regions. These initial centers are then refined by the attribute encoder, which predicts offsets and can correct imperfect anchors.
>
> **Number of particles $M$**: Please see Table 2 in our response to [Reviewer 5K7r](https://openreview.net/forum?id=vIotI25gJz&noteId=3Er3w2kH0w); briefly, $M=24$ performs best, while 16 is capacity-limited and 40 adds optimization difficulty without gains.
>
> **Slot collapse**: Particles and slots are fundamentally different: particles are local and model small, spatially localized scene parts, whereas slots typically compete over the full image extent. As noted in [17, 2], this locality bias leads to different behavior from slot-based models and tends to improve stability in practice [2], including with respect to slot collapse [5]. While there is no formal guarantee on which regions are assigned to the background particle, we do not observe severe persistent over-segmentation; static objects that never move across the dataset may instead be absorbed into the background.
>
> * Note: referenced citations are in the response for Reviewer PpcE

---

> > ### Author Rebuttal · Reviewer_Kdez · 2026-04-03
> >
> > Thanks for providing the rebuttal. However, the baselines and evaluations remain insufficient. For example, as also noted by reviewer 3qyn, the authors claim that the paper focuses on voxels rather than point clouds. This is not a convincing justification for omitting comprehensive comparisons with baselines, as point clouds and voxels can be converted easily.

---

> > > ### Author Response · Authors · 2026-04-03
> > >
> > > Thank you for acknowledging our rebuttal. We appreciate the time you invested and understand the burden, as we also serve as reviewers ourselves.
> > >
> > > We kindly and respectfully would like to point out that during the short rebuttal period we
> > >
> > > (i) implemented and evaluated two additional baselines (EC‑Diffuser with 2D‑DLP inputs and EquiDiff without the eye‑in‑hand camera),
> > > (ii) implemented and evaluated two Slot‑Attention variants (SLATE and SAVi) for RGB‑D inputs,
> > > (iii) performed the requested ablations, and
> > > (iv) evaluated our method on the entire MimicGen suite.
> > >
> > > These additions required substantial implementation effort and computational resources within the limited rebuttal window, and we believe they meaningfully expand the empirical coverage of the paper.
> > > Regarding the statement that “baselines and evaluations remain insufficient,” in order to improve our paper we kindly ask the reviewer to clarify and explicitly state what additional experiments would have changed their assessment. Given the new baselines, ablations, and extended evaluation we have provided, we believe we have addressed the main concerns raised in the original review, and we do not understand what is still considered missing.
> > >
> > > Regarding the comment that “point clouds and voxels can be converted easily,” we also do not understand what is intended here. The fact that point clouds can be converted to voxels **does not imply** that it is trivial to design an object‑centric architecture for point clouds, nor that a point‑cloud architecture can be trivially converted to voxels. We currently do not have a point‑cloud version of our model, nor a voxel version of SPAIR3D, and we do not see an “easy” way to implement either direction without substantial additional work. Instead, we built our evaluation on the established voxel‑based benchmark from EquiDiff (CoRL 2024), which already includes methods that use **point clouds** as part of its comparison (see Table 4 in our paper and the original EquiDiff work). If the reviewer meant something else by this comment, we kindly ask for clarification so that we can improve the paper accordingly.
> > >
> > > Thank you again for your time, it is appreciated.

---

### Official Review · Reviewer_5K7r · 2026-03-12

**Soundness:** 3
**Presentation:** 4
**Significance:** 2
**Originality:** 3
**Overall Recommendation:** 4
**Confidence:** 3

**Summary:**

The paper proposes a self-supervised learning scheme that learns object-centric representations from RGB-D data. The method, 3D-DLP, is built upon the Deep Latent Particles (DLP) framework, which decomposes a visual scene into “particles” corresponding to the notion of “objects.” The proposed method first performs voxelization, then trains the model using a DLP-style encoder-decoder network with reconstruction loss. The learned representations support object discovery, reconstruction, and robotic manipulation, and show improvements over AE and VAE baselines.

**Compliance With Llm Reviewing Policy:**

Affirmed.

**Final Justification:**

Most of my technical concerns are addressed. The scope of the paper is still its weakness but overall I lean towards acceptance.

**Key Questions For Authors:**

The method seems to depend on relatively simple and fixed backgrounds. This limitation is acknowledged by the authors. It would be interesting to discuss whether incorporating pre-trained semantic 2D visual representations (e.g., DINO or CLIP) could help improve object disentanglement and downstream manipulation performance.

**Limitations:**

Yes

**Strengths And Weaknesses:**

Strengths:

- The paper’s descriptions are clear, and the method is easy to understand. In general, it is easy to follow, and I enjoy reading it.

- The idea and proposed method of extending DLP to 3D are well described. It is reasonable to build a self-supervised object-centric representation learning scheme on 3D voxels. The overall design is essentially a direct translation from DLP to 3D data, which makes the method straightforward.

- The results show improvements on multiple tasks over AE and VAE, and the authors provide nice examples showing that the learned representation supports disentangling independent objects. The demo website included in the paper provides very interesting visualizations.

Weaknesses:

1. The scope of the paper is my biggest concern. The method is based on DLP and is only compared with non-object-centric baselines such as AE and VAE. However, there are many other object-centric learning methods, for example, Slot Attention. Extending such methods to RGB-D or voxel data would likely require minimal engineering effort. Improving over AE and VAE with an object-centric scheme is expected and is not particularly impressive. It would be helpful if the authors could compare with other object-centric baselines.

2. Following up on the scope of the paper, the goal is to extend DLP to 3D, which is a reasonable direction. However, 3D data goes far beyond RGB-D. For example, multi-view representations, Gaussian splatting, and NeRF-based representations are increasingly common. Voxelization can also be applied to many of these representations. Experiments, or at least a discussion covering different types of 3D data representations, would greatly strengthen the paper.

3. The paper does not provide a sensitivity analysis of M, the number of objects (particles), and this number is manually chosen by the authors depending on the dataset. This limits the practicality of the method, especially when dealing with real-world complex 3D scenes.

Overall, I am positive about the method. The approach is reasonable and appears to work as described. However, comparing primarily against simple non-object-centric baselines such as AE and VAE in most experiments weakens the empirical evaluation. Currently, my rating is weak accept, but I may update my score based on the authors’ rebuttal and other reviewers’ comments.

---

> ### Author Rebuttal · Authors · 2026-03-30
>
> We sincerely thank the reviewer for the constructive and encouraging feedback. We are grateful that you found the paper clear and the visualizations compelling, particularly the evidence that the learned representation disentangles independent objects.
>
> **Baselines**: To our knowledge, our method is the first self-supervised 3D object-centric decomposition operating directly on voxel observations. Extending DLP to 3D is not a trivial translation: it requires new objectives (e.g., the chroma loss) and replacing spatial-softmax with K-means proposals. We also do not view Slot-Attention as a straightforward voxel baseline. Slot-based methods were designed for 2D spatial attention and often require substantial tuning even on images [2, 7]; adapting them to sparse, irregular 3D voxels would require a substantial redesign. While this is an important direction, it is beyond the scope of the current paper. To strengthen the empirical comparison, we additionally include RGB-D results with SLATE [3] and SAVi [4], together with qualitative examples on the project website.
>
> Table 1: RGB-D - Slot-based vs. 3D-DLP-D
> | Method | PSNR $\uparrow$ | SSIM $\uparrow$ | LPIPS $\downarrow$ |
> |---|---:|---:|---:|
> | 3D-DLP-D (ours) | **39.39 $\pm$ 2.60** | **0.9891 $\pm$ 0.0124** | **0.0089 $\pm$ 0.009** |
> | SAVi | 26.54 $\pm$ 2.25 | 0.9204 $\pm$ 0.0356 | 0.1089 $\pm$ 0.0404 |
> | SLATE | 24.91 $\pm$ 2.20 | 0.88 $\pm$ 0.04 | 0.07 $\pm$ 0.03 |
>
> These results suggest that particle-based representations are better suited to our setting than slot-based ones: 3D-DLP-D substantially outperforms both SAVI and SLATE across all metrics.
>
> **Other 3D Object-centric works**: Thank you for pointing this out. We will provide an extended overview of 3D object‑centric approaches that are based on other types of 3D inputs. We focus our exposition on voxels because this modality has proven effective for downstream control tasks, but self‑supervised object‑centric decompositions in this setting have not been explored before. Our paper aims precisely to fill this gap and show how 3D‑DLP can be used to learn object‑centric scene representations that are suitable for downstream manipulation.
>
>
> **Number of particles $M$**: We provide an ablation on the number of particles $M$ below. Since each particle is low-dimensional, the main requirement is to have enough particles to cover the objects present in the scene. In practice, we typically use more particles than are ultimately active: for example, on MimicGen we use 24 particles, but only about 15 are active on average (i.e., with transparency close to 1), indicating that the model can naturally ignore redundant particles.
>
> Table 2: Number of particles $M$ ablation
>
> | Number of particles \(M\) | PSNR $\uparrow$ | IoU $\uparrow$ |
> |---:|---:|---:|
> | 8 | 21.30 ± 0.29 | 0.72 ± 0.0 |
> | 16 | 22.48 ± 0.31 | 0.810 ± 0.01 |
> | 24 | **24.41 ± 0.26** | **0.910 ± 0.0** |
> | 40 | 23.93 ± 0.35 | 0.890 ± 0.0 |
>
> Table 2 shows that increasing the particle budget from 16 to 24 improves reconstruction quality, but increasing it further to 40 does not. The weaker performance at 16 particles is consistent with insufficient capacity: although the 24-particle model activates only about 15 particles on average, this average masks variation across scenes, and more complex scenes can require substantially more active particles. Conversely, the lack of improvement at 40 particles suggests that simply adding more particles does not yield more useful capacity in this regime, and may instead make decomposition harder to optimize through redundancy or fragmented assignments.
>
> **Simple backgrounds and generalization**: We agree that, like many 2D self-supervised object-centric methods, our approach currently works best in relatively clean, in-domain environments and may struggle on complex real-world scenes. While there has been progress toward more robust object-centric learning in 2D [6], strong generalization remains an open challenge, especially in 3D. A promising direction is to combine 3D object-centric modeling with pre-trained self-supervised features [7], such as DINO [8], or 3D-aware foundation models [9].
>
> * Note: referenced citations are in the response for [Reviewer PpcE](https://openreview.net/forum?id=vIotI25gJz&noteId=qcKLjBEdJI)

---

> > ### Author Rebuttal · Reviewer_5K7r · 2026-04-03
> >
> > Thanks to the authors for the rebuttal. The additional results resolved my concerns about the experiments. The authors promised to discuss other forms of 3D representations, and my weak accept recommendation is conditional on that.

---

> > > ### Author Response · Authors · 2026-04-03
> > >
> > > Thank you for the acknowledgement and for the constructive suggestion. Because the rebuttal process does not permit submitting a revised manuscript, we cannot incorporate this change directly into the current version. In the next revision, we will add the following paragraph to better position our work with respect to other 3D representations:
> > >
> > > Recent self-supervised object-centric methods for image and video decomposition can be broadly grouped into patch-based [1], slot-based [2, 3, 4, 5], and particle-based approaches [6,7], which encode a set of latent entities and reconstruct scenes by composing their decoded representations. Most of this literature, however, focuses on 2D RGB observations. On the 3D side, SPAIR3D [8] extends SPAIR to point clouds and demonstrates unsupervised object-centric decomposition in synthetic, colorless settings. More recent work has broadened the scope of 3D object-centric learning, several works using neural-based rendering [9, 10, 11, 12, 13]. Specifically, uOCF [9] learns object-centric neural fields from image observations via inverse rendering; DynaVol-S [12] studies dynamic scene decomposition through object-centric voxelization and neural rendering; and GrabS [13] tackles unsupervised 3D object segmentation in point clouds using object-centric priors and embodied querying. These works are complementary to ours, but differ in both representation and learning setup. In contrast to neural-field or rendering-based approaches, our goal is to extend the Deep Latent Particles (DLP [6]) framework itself to **explicit 3D observations**, operating directly on RGB-D and voxel inputs with a direct reconstruction objective in 3D. This preserves DLP’s particle structure while avoiding dependence on inverse rendering and continuous scene querying, yielding a compact object-centric representation that is directly reconstructive in 3D and readily usable for downstream manipulation and control.
> > >
> > > [1] [Chen, Chang, Fei Deng, and Sungjin Ahn. “ROOTS: Object-Centric Representation and Rendering of 3D Scenes.”](https://arxiv.org/abs/2006.06130)
> > >
> > > [2] [Jabri, Allan, Sjoerd van Steenkiste, Emiel Hoogeboom, Mehdi S. M. Sajjadi, and Thomas Kipf. “DORSal: Diffusion for Object-centric Representations of Scenes.”](https://arxiv.org/abs/2306.08068)
> > >
> > > [3] [Li, Nanbo, Cian Eastwood, and Robert B. Fisher. “Learning Object-Centric Representations of Multi-Object Scenes from Multiple Views.”](https://arxiv.org/abs/2111.07117)
> > >
> > > [4] [Sajjadi, Mehdi S. M., Daniel Duckworth, Aravindh Mahendran, Sjoerd van Steenkiste, Filip Pavetić, Mario Lučić, Leonidas J. Guibas, Klaus Greff, and Thomas Kipf. “Object Scene Representation Transformer.”](https://arxiv.org/abs/2206.06922)
> > >
> > > [5] [Yuan, Jinyang, Bin Li, and Xiangyang Xue. “Unsupervised Learning of Compositional Scene Representations from Multiple Unspecified Viewpoints.”](https://arxiv.org/abs/2112.03568)
> > >
> > > [6] [Daniel, Tal, and Aviv Tamar. "DDLP: Unsupervised object-centric video prediction with deep dynamic latent particles."](https://arxiv.org/abs/2306.05957)
> > >
> > > [7] [Daniel, Tal, et al. "Latent Particle World Models: Self-supervised Object-centric Stochastic Dynamics Modeling."](https://arxiv.org/abs/2603.04553)
> > >
> > > [8] [Wang, Tianyu, Miaomiao Liu, and Kee Siong Ng. "Spatially invariant unsupervised 3d object-centric learning and scene decomposition."](https://arxiv.org/abs/2106.05607)
> > >
> > > [9] [Luo, Rundong, Hong-Xing Yu, and Jiajun Wu. “Unsupervised Discovery of Object-Centric Neural Fields.”](https://arxiv.org/abs/2402.07376)
> > >
> > > [10] [Smith, Cameron Omid, Hong-Xing Yu, Sergey Zakharov, Frédo Durand, Joshua B. Tenenbaum, Jiajun Wu, and Vincent Sitzmann. “Unsupervised Discovery and Composition of Object Light Fields.”](https://arxiv.org/abs/2205.03923)
> > >
> > > [11] [Stelzner, Karl, Kristian Kersting, and Adam R. Kosiorek. “Decomposing 3D Scenes into Objects via Unsupervised Volume Segmentation.”](https://arxiv.org/abs/2104.01148)
> > >
> > > [12] [Zhao, Yanpeng, et al. “Dynamic Scene Understanding through Object-Centric Voxelization and Neural Rendering.”](https://arxiv.org/abs/2407.20908)
> > >
> > > [13] [Zhang, Zihui, et al. "GrabS: Generative Embodied Agent for 3D Object Segmentation without Scene Supervision."](https://arxiv.org/abs/2504.11754)

---

### Official Review · Reviewer_PpcE · 2026-03-13

**Soundness:** 3
**Presentation:** 3
**Significance:** 3
**Originality:** 3
**Overall Recommendation:** 4
**Confidence:** 3

**Summary:**

This paper extends the Deep Latent Particles framework to 3D observations to learn object-centric 3D scene representations in a self-supervised manner. The model represents scenes using a set of latent particles that encode object position, scale, and appearance, and reconstructs the scene using volumetric rendering. Experiments demonstrate the method's usefulness on downstream task such as robotic manipulation tasks.

**Compliance With Llm Reviewing Policy:**

Affirmed.

**Final Justification:**

I believe this paper makes an interesting empirical contribution, however, the metrics used to evaluate the models considered are not completely reflected of the "object-centric" aspect of the method. It is thus somewhat difficult to assess the role that object centric representations play in the methods performance as well as how the method compares to other object centric methods.

Nevertheless, I think this paper makes an empirical contribution and is well written. Thus, I would lean towards recommending acceptance.

**Key Questions For Authors:**

* Why did the authors not benchmark against slot-based approaches for object-centric representation learning?

* How do the authors believe their method would scale to more real-world robotics tasks?

* Can the author's comment on their choice of metrics for object discovery?

**Limitations:**

yes

**Strengths And Weaknesses:**

**Strengths**

* The paper addresses an important problem. Namely, object-centric representation learning at scale on 3D data.

* The paper is well written and easy to follow.

* The method is a nice and clean extension of Deep Latent Particles and seems relatively straightforward to implement.

* The experimental results showing 3D controllability as well as performance gains in robotic manipulation tasks are quite interesting.

**Weaknesses**

* There exist another paradigm for learning object-centric representations, namely slot-based models, in which patch tokens are mapped to a set of slot tokens. The author's mention this line of work in the related work, however, they do not compare with any slot-based approaches for 3D data in their experiments, e.g., (OSRT https://arxiv.org/abs/2206.06922).

* The method relies on a specialized pipeline involving K-means-based particle proposals, and 3D convolutional networks. While this is not inherently a bad thing, I am skeptical of the scalability and broader applicability of the authors approach in computer vision task given that it uses this specialized pipeline.

* As I understand it, the experiments primarily report reconstruction metrics, while evidence for object discovery is presented more qualitatively. Reporting results using object discovery metrics such as ARI, would strengthen the evidence for the utility of this approach for object-centric representation learning.

---

> ### Author Rebuttal · Authors · 2026-03-31
>
> We thank the reviewer for their feedback and for highlighting the importance of scalable 3D object-centric representation learning, the clarity of the paper, and the promise of the robotic manipulation results.
>
> **Slot-Attention comparison**:  OSRT [1] is not directly comparable, as it is a multi-view RGB model based on Slot-Attention and trained for novel-view RGB synthesis. Our method instead learns explicit object-centric latent particles directly in the target 3D representation. Prior work [2, 3] has shown that DLP outperforms slot-based methods on RGB decision-making tasks, suggesting that particles are a better fit for downstream control. Extending slot-attention-style architectures to voxel inputs is non-trivial and beyond our scope. To strengthen the comparison, we include RGB-D results with SLATE [4] and SAVi [5], and qualitative results on the project website; see Table 1 in [Reviewer 5K7r](https://openreview.net/forum?id=vIotI25gJz&noteId=3Er3w2kH0w). 3D-DLP-D substantially outperforms both SAVI and SLATE across all metrics.
>
> **K-Means and scalability**:  We acknowledge that voxel grids and 3D convolutions are less scalable than 2D inputs due to their higher compute cost, and note this explicitly as a limitation. Our current focus is on training per environment, where 2D inputs are indeed more scalable, as also shown in LPWM [6]. Moreover, scalability and generalization remain challenging even for 2D self-supervised object-centric methods on complex real-world scenes. A promising direction is to leverage pre-trained features [7], e.g., DINO [8], though this is beyond the scope of the present work.
>
>
> **Metrics**: We agree that standard object-centric metrics such as ARI would provide additional evidence. However, these are difficult to compute in our setting because ground-truth masks are unavailable. While SAM-3D-style masks [10] could be used, they are not fully automatic and may not align with the decompositions learned by a self-supervised model. More importantly, the “objects” discovered by our method need not match human semantic masks. We therefore rely primarily on qualitative visualizations and downstream learning performance.
>
>
> [1] [Sajjadi, Mehdi SM, et al. "Object scene representation transformer."](https://arxiv.org/abs/2206.06922)
>
> [2] [Daniel, Tal, and Aviv Tamar. "DDLP: Unsupervised object-centric video prediction with deep dynamic latent particles."](https://arxiv.org/abs/2306.05957)
>
> [3] [Haramati, Dan, et al.. "Entity-centric reinforcement learning for object manipulation from pixels.](https://arxiv.org/abs/2404.01220)
>
> [4] [Kipf, Thomas, et al. "Conditional object-centric learning from video."](https://arxiv.org/abs/2111.12594)
>
> [5] [Singh, Gautam, Fei Deng, and Sungjin Ahn. "Illiterate dall-e learns to compose."](https://arxiv.org/abs/2110.11405)
>
> [6] [Daniel, Tal, et al. "Latent Particle World Models: Self-supervised Object-centric Stochastic Dynamics Modeling."](https://arxiv.org/abs/2603.04553)
>
> [7] [Seitzer, Maximilian, et al. "Bridging the gap to real-world object-centric learning."](https://arxiv.org/abs/2209.14860)
>
> [8] [Oquab, Maxime, et al. "Dinov2: Learning robust visual features without supervision."](https://arxiv.org/abs/2304.07193)
>
> [9] [Wang, Jianyuan, et al. "Vggt: Visual geometry grounded transformer."](https://arxiv.org/abs/2503.11651)
>
> [10] [Carion, Nicolas, et al. "Sam 3: Segment anything with concepts."](https://arxiv.org/abs/2511.16719)
>
> [11] [Luo, Rundong, Hong-Xing Yu, and Jiajun Wu. "Unsupervised discovery of object-centric neural fields."](https://arxiv.org/abs/2402.07376)
>
> [12] [Zhao, Yanpeng, et al. "Dynamic scene understanding through object-centric voxelization and neural rendering."](https://arxiv.org/abs/2407.20908)
>
> [13] [Zhang, Zihui, et al. "GrabS: Generative Embodied Agent for 3D Object Segmentation without Scene Supervision."](https://arxiv.org/abs/2504.11754)
>
> [14] [Chen, Kehua, et al. "Equidiff: A conditional equivariant diffusion model for trajectory prediction."](https://arxiv.org/abs/2308.06564)
>
> [15] [Daniel, Tal, and Aviv Tamar. "Unsupervised image representation learning with deep latent particles."](https://arxiv.org/abs/2205.15821)
>
> [16] [Wang, Tianyu, Miaomiao Liu, and Kee Siong Ng. "Spatially invariant unsupervised 3d object-centric learning and scene decomposition."](https://arxiv.org/abs/2106.05607)
>
> [17] [Crawford, E., & Pineau, J. (2019). Spatially Invariant Unsupervised Object Detection with Convolutional Neural Networks.](https://ojs.aaai.org/index.php/AAAI/article/view/4216)
>
> [18] [Lin, Zhixuan, et al. "Space: Unsupervised object-oriented scene representation via spatial attention and decomposition."](https://arxiv.org/abs/2001.02407)
>
> [19] [Shridhar, Mohit, Lucas Manuelli, and Dieter Fox. "Perceiver-actor: A multi-task transformer for robotic manipulation."](https://arxiv.org/abs/2209.05451)
>
> [20] [Goyal, Ankit, et al. "Rvt: Robotic view transformer for 3d object manipulation."](https://arxiv.org/abs/2306.14896)

---

> > ### Author Rebuttal · Reviewer_PpcE · 2026-04-03
> >
> > I thank the authors for their reply and for engaging with my feedback!
> >
> > My main remaining concern pertains to the "object-centricness" of the proposed approach. Specifically, given the emphasis on object-centric representations in the title and storyline of the paper, I would expect a more direct evaluation of how object-centric the models learned representations are, or at least evidence, that the object-centric structure is the driver of downstream performance in the task considered.
> >
> > As it stands, however, it seems that the reported metrics are geared more towards evaluating reconstruction quality (PSNR, SSIM, LPIPS). Furthermore, while I appreciate the added comparisons between the authors method and object-centric approaches such as Slate, these comparisons also focus on reconstruction based metrics.
> >
> > Thus, in summary, I find it a bit challenging to assess the extent to which object-centric representations are present in the model and whether they are responsible for increased downstream performance.
> >
> > Can the authors comment on why their evaluation is largely focused on reconstruction metrics?

---

> > > ### Author Response · Authors · 2026-04-03
> > >
> > > Thank you for the thoughtful follow-up. We agree that reconstruction metrics such as PSNR/SSIM/LPIPS do not by themselves directly quantify object-centricness. In the paper, these metrics are primarily used to evaluate overall scene reconstruction quality, since reconstruction is the self-supervised training signal for our representation. They are therefore intended to measure whether the learned latent captures the scene faithfully, not to serve as the sole evidence of object-centric structure.
> > >
> > > To evaluate object-centricness more directly, we provide several complementary forms of evidence. First, the decomposition visualizations in Fig. 2 show that the learned particles separate foreground objects from background and yield meaningful foreground masks and bounding boxes. Second, the particle manipulation results in Fig. 4 act as a controllability-based probe: moving or scaling an individual particle induces a localized change only in the corresponding object region, which is evidence that the particle has captured object-level structure rather than a diffuse scene feature. Third, we evaluate the usefulness of this structure in downstream decision-making through robotic policy learning on MimicGen. As discussed in our response to [Reviewer Kdez](https://openreview.net/forum?id=vIotI25gJz&noteId=snJordAtMH), we compare against both EquiDiff[1] with voxel-only observations and 2D multiview RGB DLP, and show that our particle representation is competitive or better across the task suite.
> > >
> > > More broadly, fully quantitative evaluation of object-centricness is difficult in this setting because these datasets do not provide ground-truth object decompositions or instance-level labels that would enable a clean direct metric. For this reason, we evaluate object-centricness through the combination of decomposition quality, controllable localized edits, and downstream policy utility.
> > >
> > > [1] [Chen, Kehua, et al. "Equidiff: A conditional equivariant diffusion model for trajectory prediction."](https://arxiv.org/abs/2308.06564)

---

### Decision · Program_Chairs · 2026-04-30

**Decision:**

Accept (regular)

**Comment:**

The work proposes a novel object-centric 3D representation for scenes based on a self supervised learning paradigm. Initially the paper received very mixed ratings (4, 3, 4, 2). Some of the main concerns from the reviewers related to a better discussion and comparison against other object-centric representations, in particular slot attentions, as well as missing ablations and experimental validation. The rebuttal seems to have address some of these concerns, and reviewers landed on a more positive overall assessment of the work (3, 4, 4, 4). Albeit not on a complete consensus, the signal collected so far from the reviews, rebuttal and discussion was deemed sufficient by the AC to recommend acceptance for this work.